# Molecular basis of positional memory in limb regeneration

L. Otsuki[1,2 ✉], S. A. Plattner[1,2], Y. Taniguchi-Sugiura[1,2], F. Falcon[1,2] & E. M. Tanaka[1,2 ✉]

The amputation of a salamander limb triggers anterior and posterior connective tissue cells to form distinct signalling centres that together fuel regeneration[1]. Anterior and posterior identities are established during development and are thought to persist for the whole life in the form of positional memory[2]. However, the molecular basis of positional memory and whether positional memory can be altered remain unknown. Here, we identify a positive-feedback loop that is responsible for posterior identity in the limb of an axolotl (*Ambystoma mexicanum*). Posterior cells express residual Hand2 transcription factor from development, and this primes them to form a Shh signalling centre after limb amputation. During regeneration, Shh signalling is also upstream of Hand2 expression. After regeneration, Shh is shut down but Hand2 is sustained, safeguarding posterior memory. We used this regeneration circuitry to convert anterior cells to a posterior-cell memory state. Transient exposure of anterior cells to Shh during regeneration kick-started an ectopic Hand2–Shh loop, leading to stable *Hand2* expression and lasting competence to express *Shh*. Our results implicate positive-feedback in the stability of positional memory and reveal that positional memory is reprogrammed more easily in one direction (anterior to posterior) than in the other. Modifying positional memory in regenerative cells changes their signalling outputs, which has implications for tissue engineering.

Many adult cells retain positional information from embryogenesis in the form of spatially organized differences in gene expression and chromatin[3,4], a property that could potentially be used to engineer regenerating tissues. A valuable system to dissect the mechanism of positional information and its contribution to regeneration is the salamander limb[2], in which positional information is functionally encoded in connective tissue cells[5,6]. These cells display differential gene expression and chromatin modification along the limb axes: proximal–distal, dorsal–ventral and anterior–posterior[7–11]. After limb amputation, cells migrate and converge at the amputation surface to form the regenerative blastema. Interactions between cells originating from different limb regions are required for regenerative growth[12,13]. Such positional memories allow cells to create patterning that seamlessly integrates the regenerated part with the remaining stump tissue[2]. Here we present the molecular analysis of a positional memory system that fuels regeneration.

The anterior–posterior axis is central to launching and sustaining limb regeneration. After amputation, Fgf8 secreted from anterior blastema cells interacts with Shh secreted from posterior blastema cells to induce outgrowth in an evolutionarily conserved positive-feedback loop[1,14,15]. Manipulating anterior–posterior interactions generates predictable outcomes. Surgically assembled anterior-only or posterior-only limbs fail to regenerate[16,17]. Conversely, an accessory limb (extra limb) is induced when transplanting posterior-limb skin to an innervated anterior wound (or vice versa) in an assay that generates anterior–posterior discontinuity[18]. Interestingly, Fgf ligands are expressed in the distal, apical ectodermal ridge in most vertebrates, instead of anteriorly as in salamanders[1,19–21]. Thus, the crucial role of anterior–posterior interactions in limb regeneration arose with the spatial rewiring of Fgf and Shh. Nevertheless, downstream mechanisms seem to be similar, because *Shh* inhibition or misexpression during limb regeneration in axolotls yields digit reduction or expansion phenotypes similar to those in chick (*Gallus gallus*) and mouse (*Mus musculus*) limb development[22,23]. Importantly, *Fgf8* and *Shh* are not expressed in uninjured salamander limbs. How cells retain anterior–posterior positional memory to appropriately launch *Fgf8* and *Shh* expression is not known.

To investigate anterior–posterior positional information, we studied living axolotls by using fluorescent reporters, lineage tracing and genetic or pharmacological perturbations. We discovered a Hand2–Shh positive-feedback loop that was responsible for posterior identity. By forcing this loop in regenerating anterior cells, we posteriorized their positional memory, enabling them to express *Shh* after subsequent amputation. We have leveraged positional memory mechanisms to change the signalling outputs from regenerative cells.

## Non-*Shh* lineage cells can express *Shh*

We investigated the origin of posterior, *Shh*-expressing cells during regeneration[1,24] (Fig. 1a,b). To explore whether cells that had expressed *Shh* during development persist in the limb and serve as the source of *Shh*-expressing cells after injury, we performed genetic fate mapping of embryonic *Shh* cells during regeneration in a *Shh* transgenic

[1]Institute of Molecular Biotechnology of the Austrian Academy of Sciences (IMBA), Vienna BioCenter (VBC), Vienna, Austria. [2]Research Institute of Molecular Pathology (IMP), Vienna BioCenter (VBC), Vienna, Austria. ✉e-mail: leo.otsuki@imba.oeaw.ac.at; elly.tanaka@imba.oeaw.ac.at

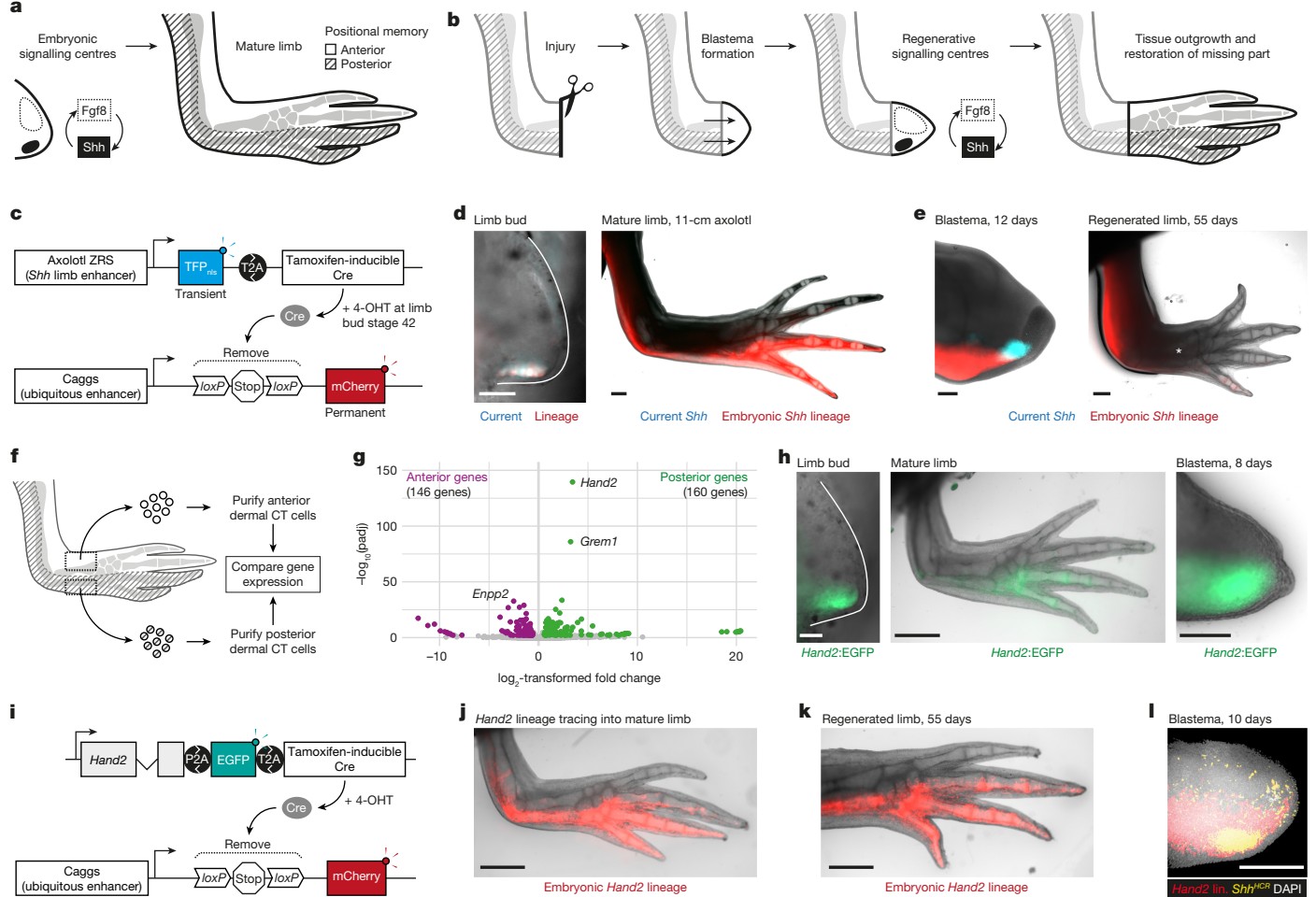

## Fig. 1 | *Shh* cells can arise from outside the embryonic *Shh* lineage.

**a**,**b**, Schematics of axolotl limb development (**a**) and regeneration (**b**). Anterior Fgf8 and posterior Shh interact to fuel limb outgrowth. **c**, Schematic of ZRS>TFP axolotls using a *Shh* limb enhancer (ZRS) to express nuclear-localized TFP (TFP$_{nls}$) and ER$^{T2}$-Cre-ER$^{T2}$, with the aim of lineage-tracing embryonic *Shh* cells through limb regeneration. When crossed to a fate-mapping axolotl and treated with 4-OHT, embryonic *Shh* cells can be permanently labelled with mCherry for lineage tracing. **d**,**e**, Spatial contributions of active *Shh*-expressing cells (blue) and embryonic *Shh* cell lineage (red) to the stage-42 limb bud (**d**, left), mature limb (**d**, right), regenerating blastema (**e**, left) and regenerated limb (**e**, right). The asterisk indicates the regenerated part of the limb. **f**, Transcriptional profiling of dermal connective tissue (CT) cells purified

from anterior or posterior limb. **g**, Volcano plot depicting differentially expressed genes in anterior and posterior cells, with the most statistically significant genes labelled. Differential expression analysis was done using DESeq2 (two-tailed Wald test with Benjamini–Hochberg adjustment for multiple testing), with a false-discovery rate cut-off of *P* < 0.01. **h**, Expression of *Hand2*:EGFP reporter in posterior limb bud, mature limb and blastema. **i**, Lineage tracing of *Hand2* cells using a similar strategy to that in **c**. **j**,**k**, Spatial contributions of embryonic *Hand2* cells (red) to the mature limb (**j**) and regenerated limb (**k**). **l**, 3D reconstruction of a *Hand2* lineage (lin.)-traced blastema stained for *Shh* mRNA (yellow) and DAPI (white). Data are representative of ten limbs (**d**, **e**, **h**, **j** and **k**) or four blastemas (**l**). Scale bars: 100 µm (limb buds), 500 µm (blastemas) or 1 mm (limbs).

reporter animal. Using the conserved *Shh* limb enhancer ZRS (zone of polarizing activity regulatory sequence, also known as MFCS1)[25,26] from the axolotl genome, we co-expressed TFP (teal fluorescent protein) and the tamoxifen-inducible Cre recombinase in transgenic axolotls that we refer to as ZRS>TFP (Fig. 1c); the transgenics are summarized in Extended Data Fig. 1a–f. As predicted, ZRS>TFP labelled *Shh*-expressing cells during limb development and regeneration (Extended Data Fig. 2a–f). To persistently label embryonic *Shh* cells, we crossed the ZRS>TFP axolotl with a *loxP*–mCherry fate-mapping axolotl[9] (Fig. 1c) and treated stage-42 progeny with 4-hydroxytamoxifen (4-OHT), yielding a labelling efficiency of 72.7 ± 18.3% (*n* = 9 limbs; Extended Data Fig. 2g), whereas untreated controls expressed negligible mCherry (Extended Data Fig. 2h).

Embryonic *Shh* cells contributed to the posterior 20% (approximately) of the upper and lower arm, and the posterior 1.5 digits of the hand (Fig. 1d and Extended Data Fig. 3a). Pulsing with 4-OHT after stage 42 labelled the distal subset of the lower arm and hand (Extended Data

Fig. 3b). We amputated labelled forelimbs and tracked regeneration. Most regenerated *Shh* cells (TFP-positive) were mCherry-negative (23.1 ± 22.1% of TFP signal overlapped with mCherry, *n* = 10; Fig. 1e), indicating that cells outside the embryonic *Shh* lineage switch on *Shh* during regeneration. Indeed, embryonic *Shh* cells were depleted from the regenerated limb (Fig. 1e and Extended Data Fig. 3c). To test the requirement for embryonic *Shh* cells, we surgically removed them before amputation (88.7 ± 6.1% depletion; *n* = 6 limbs from the 9 days after amputation (d.p.a.) of blastema) (Extended Data Fig. 3d,e). Depleted limbs expressed ZRS>TFP in the blastema and regenerated with similar timing to controls (Extended Data Fig. 3f,g). Although we cannot exclude the possibility that the residual 10% of embryonic *Shh* cells compensated for the depleted cells by inducing *Shh* non-autonomously, this result indicates that embryonic *Shh* cells are dispensable for expressing *Shh* during regeneration. We infer that the posterior information necessary to express *Shh* is not limited to embryonic *Shh* cells.

## Spatial priming by transcription factors

To identify genes that prime posterior cells to express *Shh*, we transcriptionally compared anterior and posterior limb cells (Fig. 1f). We purified transgenically labelled *Prrx1*+ dermal connective tissue cells, which are strong carriers of positional memory[18,27,28]. We expected to detect molecular differences, because anterior and posterior cells behave differently when transplanted. Anterior skin transplanted posteriorly induces normal-looking or mildly hypomorphic regenerated limbs, whereas posterior skin transplanted anteriorly induces ectopic anterior Shh and extra digits (polydactyly)[28–30].

Anterior and posterior cells differentially expressed around 300 genes (DESeq2, $\alpha < 0.01$) (Fig. 1g). Of these, *Hand2* dominated the posterior cell signature as ordered by statistical significance (Fig. 1g and Supplementary Table 1). *Hand2* encodes a bHLH transcription factor that in limb buds is expressed posteriorly and induces *Shh* in mouse, chick and zebrafish (*Danio rerio*)[31–34]. Hand2 has not been implicated in post-embryonic positional memory, but its intracellular function would be consistent with the persistence of positional information following the enzymatic removal of cell-surface molecules[29]. Several other transcription factors with anterior–posterior-stratified expression in developing mouse limbs were expressed in corresponding domains of the axolotl limb (Supplementary Tables 1 and 2). For example, posterior cells expressed *Hoxd13* and *Tbx2*, whereas anterior cells expressed *Alx1*, *Lhx2* and *Lhx9* (Extended Data Fig. 4a–c). The general forelimb genes *Tbx5* and *Prrx1* were found in both anterior and posterior cells (Extended Data Fig. 4d). Thus, axolotl limb cells continuously express a subset of transcription factors in development-like spatial domains, and this was also reported in zebrafish pectoral fins[35]. To assess whether, and if so, how, anterior and posterior domains can communicate, we analysed Gene Ontology terms, which showed differentially expressed genes enriched in the 'extracellular matrix' and 'cell adhesion' categories (Extended Data Fig. 4e–h and Supplementary Tables 3 and 4). These molecules, including collagens, might generate distinct signalling environments in the anterior and posterior limb.

## *Hand2* cells express *Shh* after injury

We proposed that Hand2 primes posterior cells to express *Shh* after injury, given that Hand2 is necessary for *Shh* expression in mouse limb buds and directly binds the ZRS enhancer[34,36]. To track Hand2, we generated a *Hand2*:EGFP knock-in axolotl co-expressing endogenous *Hand2* with EGFP (enhanced green fluorescent protein) (Extended Data Fig. 5a). *Hand2*:EGFP was expressed continuously in posterior cells: in limb bud, uninjured limb and, as described previously, blastema[1] (Fig. 1h). We expect *Hand2* to be translated, because EGFP expression requires *Hand2* translation and ribosome skipping at the T2A sequence[37,38]. In uninjured limbs, dermal and interstitial connective tissue cells weakly expressed *Hand2*:EGFP (Extended Data Fig. 5b; further characterization can be found in Extended Data Fig. 8a–f). *Hand2*:EGFP fluorescence increased $5.9 \pm 0.4$-fold during regeneration before returning to baseline (Extended Data Fig. 5c). Flow cytometry revealed a similar fivefold increase in *Hand2*:EGFP fluorescence per cell (Extended Data Fig. 5d). *Hand2*:EGFP increased $2.3 \pm 0.2$-fold before ZRS > TFP onset at 7 d.p.a. (Extended Data Fig. 5c,e). Note that these measurements are semi-quantitative because we used a non-labile EGFP.

We next wondered whether *Hand2* cells express *Shh* during regeneration. *Hand2* cells had not been lineage-traced in the limb in any organism. We generated a *Hand2* knock-in axolotl for lineage tracing (Fig. 1i and Extended Data Fig. 5f). We found that embryonic *Hand2* cells contribute to the posterior half of the axolotl forelimb and the posterior 2.5 digits of the hand, resembling active *Hand2* expression (Fig. 1h,j). Nonspecific mCherry labelling was negligible (Extended Data Fig. 5g). After amputation, *Hand2*-lineage cells regenerated a comparable domain (Fig. 1k and Extended Data Fig. 5h, i). Thus, embryonic

*Hand2* cells are retained during adulthood and in regeneration. We also converted the lineage reporter in 7-cm axolotls with fully formed limbs, and this yielded a similar expression pattern to that in animals that had been converted at embryonic stage. However, a higher 4-OHT dose and multiple treatments were required to convert 7-cm axolotls, and this is probably due to weaker *Hand2* and Cre expression than during development (Extended Data Fig. 5j,k). Importantly, 3D imaging confirmed that *Hand2* cells give rise to *Shh* cells during regeneration (Fig. 1l and Extended Data Fig. 5l,m).

## *Hand2* is required for posterior identity

In mouse and zebrafish limb and fin buds, respectively, *Hand2* is necessary for *Shh* expression[31,33,34]. We tested whether this function is conserved in axolotls. We mutated *Hand2* by co-injecting axolotl eggs with Cas9 protein and two efficient single guide RNAs (sgRNAs) targeting the *Hand2* translational start codon (Fig. 2a and Extended Data Fig. 6a,b). Knowing that homozygous *Hand2*−/− mutant mice die embryonically as the result of heart defects[39], we analysed mosaically mutated $F_0$ axolotls (CRISPants). *Hand2* CRISPants had higher lethality than controls, probably reflecting widespread gene deletion: 52% lethal before digit patterning ($n = 60$ of 116 injected eggs) versus 14% ($n = 4$ of 28) (Fig. 2b). When analysing 'escaper' CRISPants that were hypomorphic for *Hand2*, we found that 45% had digit-number or outgrowth defects ($n = 50$ of 112 limbs). The range in severity was consistent with mosaic *Hand2* inactivation (Extended Data Fig. 6c,d). The most severely affected hypomorphs had no limb outgrowth beyond the shoulder girdle ($n = 9$; Fig. 2c), similar to *Shh* CRISPants[40]. When amputated, almost all *Hand2* CRISPants regenerated fewer digits, including those that originally had the correct digit number (Fig. 2d and Extended Data Fig. 6e,f). During this work, we attempted to derive homozygous $F_2$ mutants with a 64-bp deletion encompassing the *Hand2* translational start (*Hand2*Δ64; Extended Data Fig. 6g–i). However, these axolotls had 96.7% lethality before the end of limb development ($n = 29$ of 30 animals), supporting our use of CRISPants. Conditional deletion of endogenous genes is not readily feasible in axolotls.

We validated that *Hand2* was responsible for these phenotypes in two ways. First, we generated *Hand2* CRISPants in a *Hand2*:EGFP background. *Hand2* mutant cells should not express EGFP. We injected *Hand2* sgRNAs into one side of cleaved *Hand2*:EGFP eggs, waited for development then compared EGFP fluorescence in the two limb buds (Extended Data Fig. 6j); 46% of sgRNA-injected axolotls displayed a mutant phenotype ($n = 13$ of 28), as before (Fig. 2b). The defective limb bud had notably less EGFP than the other, consistent with *Hand2* deletion (Extended Data Fig. 6k,l). Second, we targeted *Hand2* with further sgRNAs. Hand2 has a second methionine (M146). We individually injected three sgRNAs targeting sequences at, or downstream of, M146 to generate *Hand2*M146 CRISPants (Extended Data Fig. 6m). The *Hand2*M146 CRISPants had similar phenotypes to the original *Hand2* CRISPants, at similar frequencies (Extended Data Fig. 6n). We conclude that *Hand2* is required for axolotl limb development.

Next, we investigated whether *Hand2* is necessary to express *Shh*. We mutated *Hand2* in the ZRS>TFP background (Fig. 2e). Live imaging showed that *Hand2* CRISPants had fewer ZRS>TFP+ cells and/or expressed weaker TFP than did controls, indicating that *Hand2* is necessary to express *Shh* (Fig. 2f,g). Limb buds with the weakest TFP developed into limbs with 0–3 digits (Extended Data Fig. 6o). TFP intensity during development and regeneration was directly correlated (Spearman's rank test; $r = 0.74$, $P = 2.40 \times 10^{-3}$, $n = 14$ limbs) (Extended Data Fig. 6p). Thus, *Hand2* is necessary for *Shh* expression during axolotl limb outgrowth.

We then explored whether mutation of *Hand2* resulted in loss of positional information by doing a functional transplantation assay. *Hand2* CRISPants lost *Shh* expression, so we could address this question in axolotls using the accessory limb model (ALM). In the ALM, an

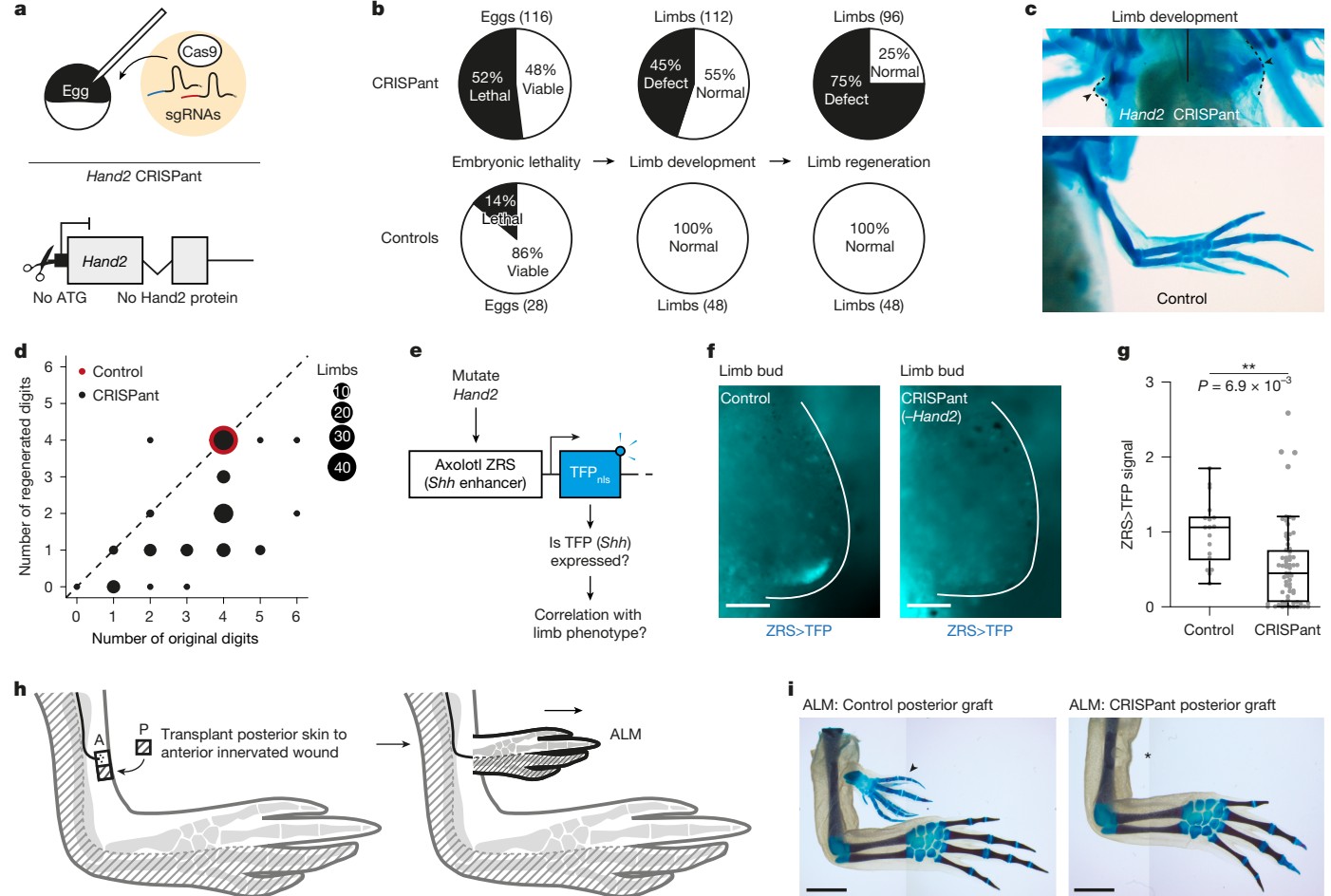

**Fig. 2 | *Hand2* is necessary for *Shh* expression and posterior limb identity.**
**a**, CRISPR–Cas9 strategy to mutate *Hand2*. **b**, Defects in *Hand2* CRISPants (top) compared with controls (bottom). **c**, Severe *Hand2* CRISPant phenotype in which no limb skeleton (blue) formed beyond the shoulder girdle (arrowed; phenotype observed in 9 of 112 limbs). A control limb is shown for comparison. **d**, Dot plot comparing the number of digits at the end of development (*x* axis) with the number of digits regenerated after amputation (*y* axis) in *Hand2* CRISPants and controls. Dot size corresponds to the number of limbs. The diagonal line indicates the occasion in which the number of limbs regenerated was the same number of original digits; *n* = 96 (CRISPant) or *n* = 48 (control) limbs. **e**, Strategy to assess whether *Hand2* is necessary for *Shh* expression by mutating *Hand2* in ZRS>TFP animals. **f**, ZRS>TFP reporter expression in limb buds of control and *Hand2* CRISPant axolotls at stage 42. **g**, Quantification of **f**, normalized to the mean of the control cohort. Boxplots depict median, interquartile range and 1.5× interquartile range. Each dot represents one limb bud; *n* = 68 (CRISPant) or *n* = 18 (control) limb buds. We used a two-tailed Kolmogorov–Smirnov test. **h**, The anterior ALM. **i**, The ALM using control grafts (3 of 6, arrowed; left) or *Hand2* CRISPant grafts (0 of 6, asterisk; right). The difference was statistically significant (assessed by the number of digits formed; two-tailed Wilcoxon signed rank test; *P* = 0.03, *W* = −4.00, *n* = 6 per group). Limbs were stained using Alcian blue (cartilage) and Alizarin red (bone) for contrast. Scale bars: 100 μm (limb buds) or 5 mm (ALMs).

ectopic limb grows out from an innervated anterior wound if grafted with posterior skin (or vice versa), because this generates anterior–posterior discontinuity[18] (*n* = 3 limbs from six surgeries; Fig. 2h,i). If *Hand2* specifies posterior identity, *Hand2*-mutant posterior skin should not induce accessory limbs when transplanted anteriorly. This was the case (*n* = 0 limbs from 6 surgeries with *Hand2* CRISPant skin, Fig. 2i). *Hand2*[M146] CRISPant posterior skin gave the same result (*n* = 0 limbs from 6 surgeries using *Hand2*[M146] CRISPant skin, and *n* = 5 limbs from 6 surgeries using control skin) (Extended Data Fig. 6q). Next, we tested whether *Hand2* mutant cells acquire anterior identity. Unlike anterior skin, *Hand2* CRISPant posterior skin did not induce limbs at innervated posterior wounds (*n* = 0 limbs from 6 surgeries) (Extended Data Fig. 6r,s). Thus, *Hand2* is necessary for posterior identity, at least insofar as inducing *Shh*, but mutant cells do not default to anterior.

## *Hand2* expression posteriorizes identity

We investigated whether *Hand2* misexpression is sufficient for *Shh* expression in axolotls, as was found for mouse limb buds and chick wing buds[32,33]. We generated transgenic axolotls in which the mouse *Prrx1* limb enhancer controlled an mCherry-tagged axolotl *Hand2* sequence, resulting in expression throughout the limb bud and blastema mesenchyme[41] (Fig. 3a and Extended Data Fig. 7a). Each F₀ axolotl expressed mCherry–Hand2 at different levels and in different spatial domains, presumably depending on copy number, genomic insertion site and extent of mosaicism. *Hand2* misexpression induced ectopic ZRS>TFP (*n* = 7 of 9 limb buds carrying TFP) and polydactyly (*n* = 7 of 16 limbs) (Fig. 3b and Extended Data Fig. 7b–d). The polydactyly was similar to that induced by *Shh* misexpression[23]. In two cases, anterior *Hand2* induced an ectopic limb, a phenotype that has not been reported in other species but is consistent with the ability of the axolotl to generate limbs at sites of anterior–posterior discontinuity (Fig. 3c). These phenotypes occurred in animals with stronger mCherry–Hand2 expression (Extended Data Fig. 7e). Misexpression of axolotl *Hand2* is therefore sufficient to trigger *Shh*, polydactyly and, in extreme cases, ectopic limbs.

We predicted that uniform *Hand2* expression might eliminate the anterior–posterior differences required for limb outgrowth (Fig. 3d).

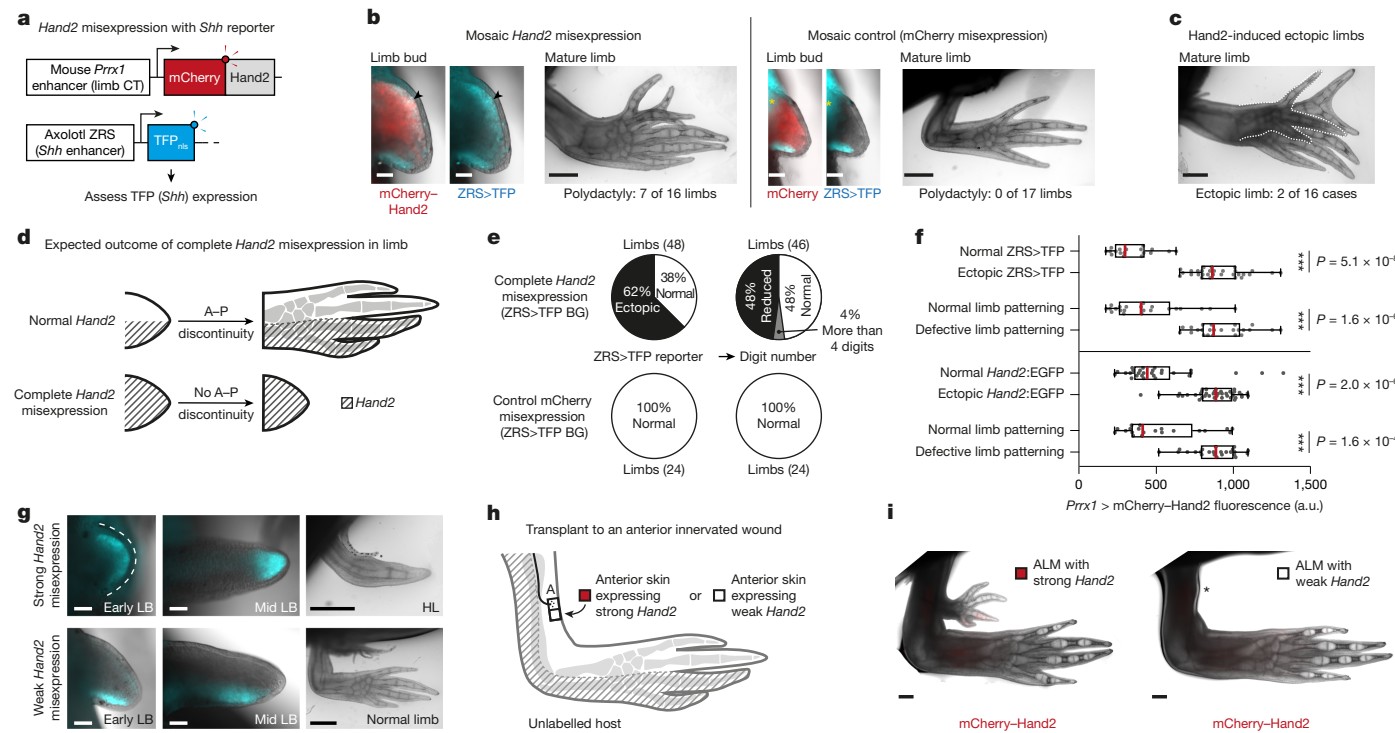

**Fig. 3 | Hand2 is sufficient for Shh expression and posterior limb identity. a**, Strategy to misexpress mCherry–Hand2 fusion protein, with ZRS>TFP reporter in the genetic background. **b**, Stage-42 limb bud mosaically misexpressing mCherry–Hand2 (left) or mCherry alone (right); mCherry–Hand2 induced ectopic anterior ZRS>TFP (arrowed; n = 7 of 9 limbs) and polydactyly (n = 7 of 16 limbs), whereas mCherry alone did not (n = 17 limbs). Asterisks indicate autofluorescence in the blue channel. **c**, An ectopic limb induced by mosaic mCherry–Hand2 expression (dotted outline; 2 of 16 cases). **d**, Predicted experimental outcome when mCherry–Hand2 is expressed uniformly in CT cells; A–P, anterior–posterior. **e**, ZRS>TFP expression and digit number after uniform misexpression of mCherry–Hand2 (top) or mCherry alone (bottom). BG, background. **f**, The mCherry–Hand2 expression level

(x axis) and the resulting limb phenotype (y axis). Boxplots depict median (red line), interquartile range and 1.5× interquartile range. Each dot represents one limb. Summary of two experiments: one in ZRS>TFP axolotls (top, n = 48 limbs) and one in Hand2:EGFP axolotls (bottom, n = 68 limbs). We used two-tailed Kruskal–Wallis tests with Dunn's multiple comparisons testing; a.u., arbitrary units. **g**, ZRS>TFP expression (blue) in axolotls misexpressing strong mCherry–Hand2 (top) or weak mCherry–Hand2 (bottom). LB, limb bud; HL, hypomorphic limb. **h**, Anterior ALM to assess the positional identity of cells misexpressing strong or weak mCherry–Hand2. **i**, All 8 strong mCherry–Hand2 grafts induced an accessory limb (left), whereas none of the 8 weak mCherry–Hand2 grafts did (right) at 34 days after grafting. The asterisk indicates the lack of an accessory limb. Scale bars: 100 μm (limb buds) or 1 mm (limbs and ALMs).

Conceptually, this would be similar to double-posterior salamander limbs that do not regenerate[16,17]. To test this prediction, we used F₁ Prrx1>mCherry–Hand2 animals that uniformly misexpress mCherry–Hand2 in connective tissue cells. These animals also had a reporter for posterior gene expression (ZRS>TFP reporting Shh or Hand2:EGFP reporting endogenous Hand2). F₁ siblings expressed different levels of mCherry–Hand2. Axolotls strongly misexpressing mCherry–Hand2 expressed ZRS>TFP and Hand2:EGFP throughout the anterior–posterior axis and generated hypomorphic spikes or no limbs (Fig. 3e–g and Extended Data Fig. 7f–j). Thus, Hand2 impeded outgrowth, apparently by posteriorizing the entire limb field. By contrast, siblings with weak mCherry–Hand2 exhibited normal-looking ZRS>TFP, Hand2:EGFP and limb patterning (Fig. 3f,g and Extended Data Fig. 7j). Interestingly, the two-fold expression difference between 'strong' and 'weak' mCherry–Hand2 (Fig. 3f) is similar to the 2.39-fold rise in Hand2 that precedes Shh expression during regeneration (Extended Data Fig. 5c). This finding hints at a role for Hand2 levels in inducing Shh.

To characterize Hand2-induced posteriorization, we performed RNA sequencing. We amputated Prrx1>mCherry–Hand2 limbs and Prrx1>mCherry controls, collected anterior blastemas at 14 d.p.a. and purified mCherry⁺ cells to identify Hand2-induced genes. We also compared bona fide anterior blastema cells labelled with Alx4:mCherry (we will discuss this further later). Because the Hand2 misexpression cassette is codon altered, we could quantify endogenous Hand2 transcripts. Hand2-misexpressing cells upregulated posterior transcription

factors (Hand2, Hoxd13 and Klf8) and downregulated anterior factors (Lhx2, Lhx9, Barx1, Zfhx4 and Hoxc10) (Supplementary Tables 5 and 6). Therefore, Hand2 induces expression changes that are consistent with posteriorization (Supplementary Tables 7–10).

To test for functional posteriorization, we used the ALM. Anterior skin from double-transgenic limbs expressing ZRS>TFP, together with either 'weak' Hand2 or 'strong' Hand2, was grafted to innervated anterior wounds. Strong-Hand2 skin grafts upregulated the ZRS>TFP reporter and induced accessory limbs, whereas weak-Hand2 skin did neither (n = 8 grafts per condition) (Fig. 3h,i and Extended Data Fig. 7k). We conclude that strong Hand2 misexpression in anterior skin is sufficient to induce its posteriorization.

## Plasticity of memory during regeneration

Whether anterior–posterior identity is irreversibly fixed after embryonic development remains an open question. We tested whether cells change identity following transplantation of anterior or posterior cells to the opposite side of an unlabelled host limb and subsequent limb amputation. To visualize anterior versus posterior identity, we generated Alx4:mCherry_Hand2:EGFP double-reporter axolotls[35,42] (Fig. 4a,b and Extended Data Fig. 8a,b). Alx4:mCherry and Hand2:EGFP in uninjured limbs were predominantly expressed in loose connective tissue, joints, skeletal and peri-skeletal elements (Extended Data Fig. 8c–e). Each reporter labelled similar cell populations

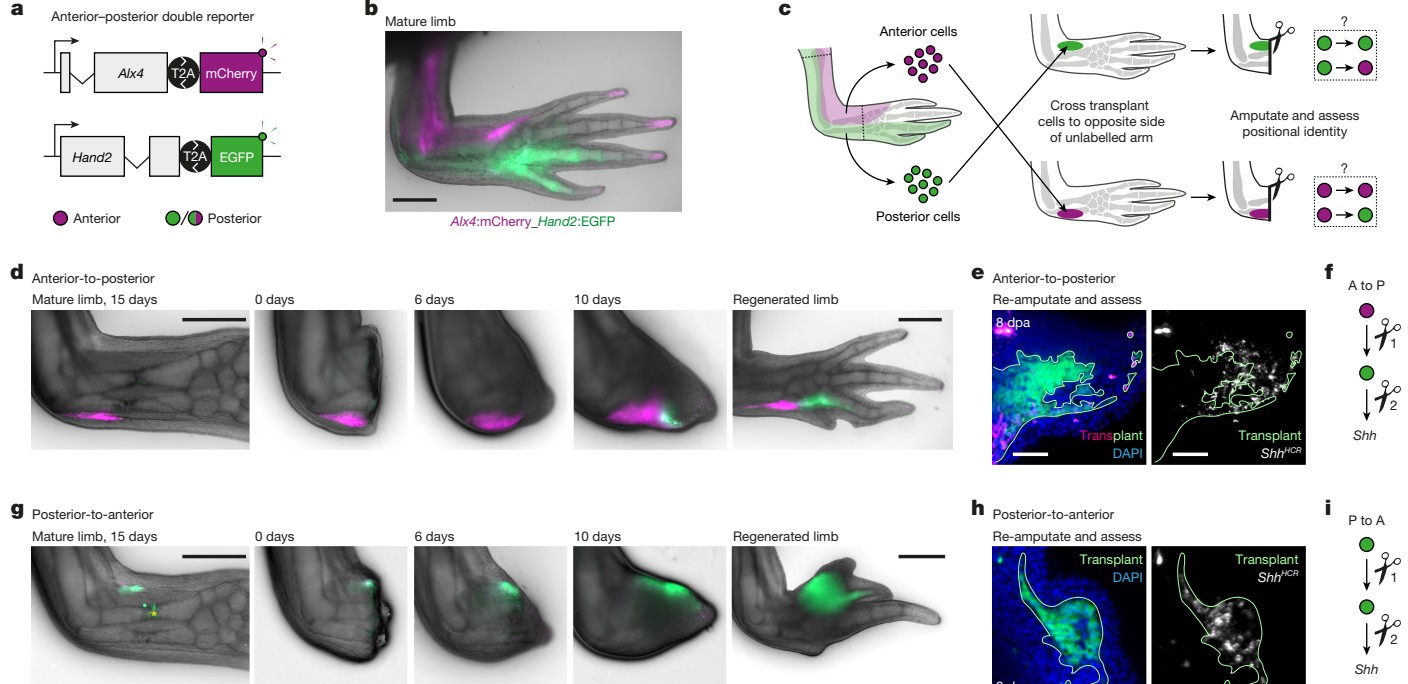

**Fig. 4 | Anterior–posterior positional memory is reprogrammable during regeneration. a**, Schematic of anterior–posterior double-reporter constructs. **b**, Mature limb of an *Alx4*:mCherry_*Hand2*:EGFP double-transgenic axolotl. **c**, Experimental design to explore whether positional identity is fixed or plastic. **d**, Anterior-to-posterior transplantations of *Alx4*:mCherry_*Hand2*:EGFP double-transgenic cells into an unlabelled host (colours as in **b**). **e**, Assay to test whether newly arising EGFP⁺ cells can express *Shh* (white) (*n* = 3 of 3 blastemas).

**f**, Schematized result of the anterior (A) to posterior (P) transplantations. **g**, Posterior-to-anterior transplantations of *Alx4*:mCherry_*Hand2*:EGFP double-transgenic cells into an unlabelled host (colours as in **b**). The asterisk indicates autofluorescence. **h**, Assay to show that EGFP⁺ cells can still express *Shh* (white) (*n* = 3 of 3 blastemas). **i**, Schematized result of the posterior to anterior transplantations. Data are representative of three transplantations each (**d**, **e**, **g** and **h**). Scale bars: 1 mm (**b**, **d** and **g**) or 100 μm (**e** and **h**).

(Extended Data Fig. 8f), so we could purify equivalent anterior or posterior cells for transplantation.

Anterior cells purified by fluorescence-activated cell sorting (mCherry⁺/EGFP⁻) were injected into the posterior side of an unlabelled host limb (Fig. 4c). Two weeks later, the cells remained mCherry⁺, which is consistent with sustained anterior identity (Fig. 4d). After limb amputation, however, transplanted cells that had entered the blastema expressed EGFP from 8 d.p.a., indicating a posteriorized identity (Fig. 4d and Extended Data Fig. 9a), whereas stump cells remained mCherry⁺. These divergent behaviours (mCherry⁺ in the stump and EGFP⁺ in the regenerated limb) indicate that anterior–posterior identity is stable at steady state but flexible during regeneration.

If transplanted anterior cells were stably posteriorized, they should express *Shh* following regeneration and a subsequent amputation. Indeed, a second amputation triggered some of the transplanted cells to express *Shh* (Fig. 4e,f). In this successive amputation experiment, posterior identity persisted through a regeneration cycle, indicating that positional memory was stably posteriorized.

In the reciprocal experiment, purified EGFP⁺ posterior cells were injected anteriorly. Transplanted posterior cells stably retained EGFP expression, not only in the uninjured limb (Fig. 4g), but also during regeneration, resulting in ectopically patterned regenerates (*n* = 3 of 3 limbs; Fig. 4g and Extended Data Fig. 9b). After a second amputation, marked cells expressed *Shh*, further reflecting the maintenance of a posterior memory state (Fig. 4h,i and Extended Data Fig. 9c). We did not detect any anteriorized (mCherry⁺ only) cells (Extended Data Fig. 9b), although we cannot exclude the possibility that some transplanted cells became untraceable by losing both EGFP and mCherry expression. Taken together, transplanted anterior cells switch to a posterior memory state during regeneration, whereas transplanted posterior blastema cells retain their original posterior memory.

To characterize the anterior-to-posterior memory switch, we compared anterior cells transplanted posteriorly (A→P) or anteriorly (A→A) with non-transplanted controls (A or P) at the transcriptional level. Principal component analysis (PCA) revealed that the first principal component, PC1, discriminated samples on the basis of presence or absence of transplantation (Extended Data Fig. 9d). PC2 discriminated A and A→A cells from P and A→P cells, explaining 25% of the total variance. PC2 was driven heavily by Hand2 and EGFP, which is expected given that P and A→P cells were purified by *Hand2*:EGFP expression (Extended Data Fig. 9e and Supplementary Table 11). However, other anterior–posterior factors also contributed to PC2, including Hoxd13 and Lhx9 (Extended Data Fig. 9e and Supplementary Table 11). Crucially, A→P cells were more similar to P cells than to A cells or A→A cells in the PCA (Extended Data Fig. 9d). This finding supports a global shift of A→P transplanted cells towards a posterior transcriptional state.

We also analysed the expression status of anterior factors during this identity change (Extended Data Fig. 4a,b). A→P cells downregulated anterior transcription factors (Alx1, Lhx9, Dmrt2, Pbx3, Hoxc10, Tbx22 and Zfhx4) (Extended Data Fig. 9f,g and Supplementary Table 12). Overall, A→P transplants downregulated 60.1% (578 of 961) of anterior blastema-specific genes and upregulated 22.5% (78 of 346) of posterior blastema-specific genes (Supplementary Table 13). Thus, transplanted cells lost their original anterior identity and gained posterior identity (Extended Data Fig. 9f,g and Supplementary Table 12). Importantly, the loss of anterior markers was not a transplantation artefact. Anterior genes (*Pbx3*, *Dmrt2* and *Hoxc10*) were lost specifically in A→P transplantations and not in A→A controls (Supplementary Table 14). For differentially expressed genes in A→P cells, see Supplementary Tables 15–18.

Anterior-to-posterior transplantation caused substantial shifts in mRNA towards a posteriorized signature. The data support a

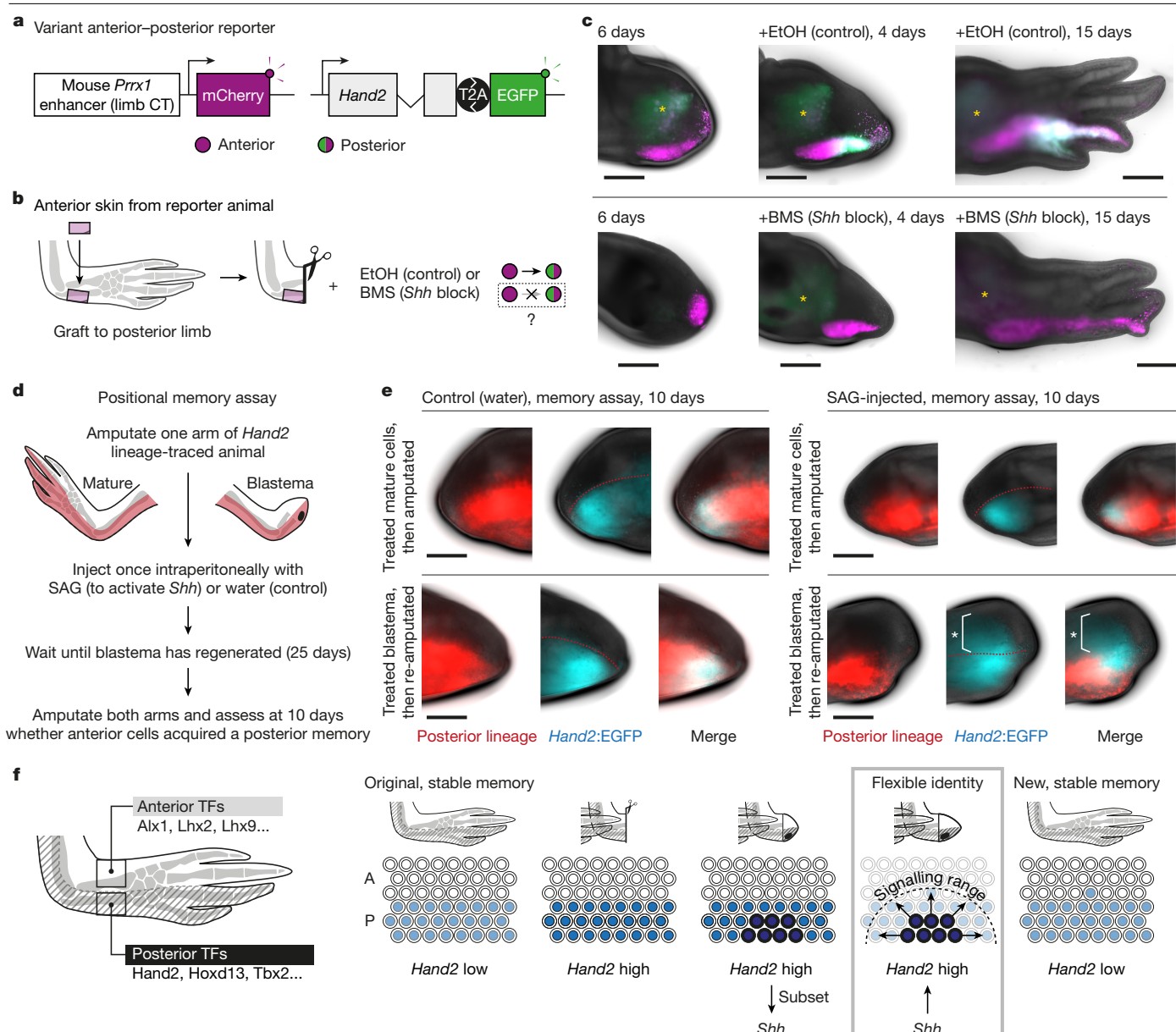

**Fig. 5 | *Shh* signalling posteriorizes anterior cell positional memory.**
**a**, Double reporter in which *Prrx1*>mCherry (magenta) labels connective tissue (CT) cells and *Hand2*:EGFP (green) labels posterior cells. **b**, Strategy to test whether *Shh* signalling is necessary to posteriorize anterior cells (colours as in **a**). **c**, Anterior to posterior transplantations of *Prrx1*>mCherry_*Hand2*:EGFP double-reporter skin to unlabelled hosts, treated with ethanol or BMS-833923 six days after amputation. Asterisks indicate autofluorescence. Images are representative of *n* = 4 of 4 limbs per assay. **d**, Strategy to test whether *Shh* signalling is sufficient to posteriorize anterior cell memory at steady state and/or during regeneration. **e**, *Hand2*:EGFP expression (blue) in blastemas during the positional memory assay in **d**. Top row, experiments in which mature cells were treated with water (control) or SAG, then amputated to form a blastema. Bottom row, experiments in which blastema cells were treated with water or SAG, allowed to regenerate fully and then re-amputated to form a blastema. Embryonic *Hand2* cells are lineage-traced in red. Asterisks indicate *Hand2*:EGFP expression outside the embryonic *Hand2* lineage. Representative of *n* = 5 of 5 limbs (SAG) or 4 of 4 limbs (control) per condition. **f**, Model for transmitting anterior–posterior positional memory through regeneration. Positional memory is stable at steady state but is transiently flexible and vulnerable to posteriorization in the blastema. A, anterior; P, posterior; TF, transcription factor. Scale bars, 1 mm.

reprogramming event with lasting changes to positional memory and signalling potential.

## *Shh* requirement for posteriorization

We proposed that *Shh* posteriorized the A→P transplants because transplanted anterior cells would be exposed to Shh secreted by endogenous posterior cells. We transplanted *Prrx1*>mCherry_*Hand2*:EGFP transgenic cells and exposed animals during regeneration to an

inhibitor of Shh signalling (Fig. 5a,b). Strikingly, BMS-833923 (a Shh pathway inhibitor[40]) prevented posteriorization (Fig. 5c and Extended Data Fig. 9h). Thus, endogenous Shh was responsible for transplant posteriorization.

Considering these findings, we investigated whether Shh signalling would be sufficient to rewrite positional memory in anterior cells. When we delivered a Shh-encoding baculovirus anteriorly, infected cells induced *Hand2*:EGFP locally (Extended Data Fig. 10a,b). Thus, anterior blastema cells are responsive to Shh. During regeneration, the strongest

Hand2 expression occurs next to the Shh signalling centre, indicating that posterior blastema cells also upregulate Hand2 in response to Shh (Fig. 1h,l). Indeed, continuously blocking Shh signalling prevented posterior blastema cells from fully upregulating Hand2 during regeneration, resulting in hypomorphic limbs (Extended Data Fig. 10c). Thus, both anterior and posterior blastema cells are responsive to Shh.

Next, we performed a lineage-tracing experiment to test whether transient Shh signalling can convert anterior cells into posterior memory cells in vivo (Fig. 5d). To distinguish anteriorly and posteriorly specified cells, we converted the Hand2 lineage-tracing axolotls (Fig. 1i) with tamoxifen at the limb-bud stage, resulting in stable expression of mCherry in posterior cells. After unilateral limb amputation, we injected SAG intraperitoneally at the cone stage of regeneration. This approach allowed us to compare mature cells (left limb) and blastema cells (right limb) in one experiment. We found that mCherry⁻ cells in the anterior blastema acquired Hand2:EGFP expression after SAG injection, but those in the unamputated limb did not (Extended Data Fig. 10d,e). The blastema descendants continued to express Hand2:EGFP when we allowed the limb to fully regenerate and then re-amputated it (Fig. 5e and Extended Data Fig. 10f). In a second set of experiments, SAG treatment of blastema cells in the second round of amputation resulted in ectopic Shh expression (Extended Data Fig. 10g). In all, we conclude that transient Shh signalling is sufficient to posteriorize positional memory in anterior blastema cells.

## Discussion

How cells recall positional information to regenerate missing tissue is a long-standing question. Few studies have investigated changes to positional memory, in which positional information is rewritten in a way that that affects successive rounds of regeneration[43]. Here, we identified a genetic circuit that maintains posterior identity in the axolotl limb and alters positional memory when triggered in anterior blastema cells. Positive feedback between Hand2 and Shh, which sustains itself after a trigger, provides a molecular explanation for stable positional memory. It also explains why it is easy to posteriorize anterior cells but difficult to anteriorize posterior cells: anterior cells readily activate the loop following a Shh trigger, whereas posterior cells cannot disengage. This posterior dominance bears parallels with the posterior prevalence of Hox genes during limb development, in which posterior Hox proteins exert more-dominant effects than do anterior Hox proteins. The 5′ Hox genes probably interact with Hand2 to regulate Shh during regeneration, similar to during development. Indeed, Hox genes are differentially expressed in our RNA-seq data, and we found that Hand2 induces Hoxd13 (Extended Data Fig. 4a,b and Supplementary Table 5). Shh activation during limb development involves multiple transcriptional inputs. Genetic analyses in mouse and chick uncovered numerous transcription factors (including PBX1/PBX2, GLI3 and TBX3) acting with Hand2 and 5′ Hox required for posterior Shh expression[42,44–46], although precisely how such factors place the SHH domain remains elusive[47]. Moreover, the BMP inhibitor Grem1 acts as a relay to balance posterior SHH and BMP activity with FGF expression, outgrowth and termination of limb development[48–51]. Understanding how robust positional memory is achieved will require analysis of anterior–posterior chromatin accessibility, transcription factor binding and modelling of multiple feedback loops between signalling pathways beyond Hand2 and Shh.

We showed that Shh can posteriorize positional memory in blastema cells, but not in uninjured cells. Similar conclusions have been drawn for proximalization using retinoic acid[52]. Explaining why blastema cells can alter their positional memory, whereas uninjured cells are fixed, is an important next step. Our conclusion that Shh can change positional memory differs from that of a previous study that used iterative ALM and morphogenetic read-outs[1]. That study lacked transgenic axolotls for tracking anterior–posterior identity, making it difficult

to locate reprogrammed cells. It has been suggested that ventral cells can dorsalize during regeneration, although specific lineage tracing is lacking[10]. It will be interesting to test whether dorsal dominance exists in the dorsal–ventral axis.

Our results provide evolutionary insights into limb ontogenesis. We found that the Shh lineage is more widespread in axolotl limbs than in mouse[53] (Extended Data Fig. 3a). This probably stems from the more-proximal location of Shh cells during axolotl development and could explain why the mutant limb phenotype is more severe in axolotls than in mice[24,40,54,55]. We showed that uniformly expressing Hand2 ablated positional discontinuity and prevented limb outgrowth, possibly owing to the unique configuration of anterior–posterior discontinuity in salamanders. Interestingly, Hand2 was previously implicated in posterior identity in the zebrafish pectoral fin, although Hand2 was not coupled to Shh in this system and memory effects were not assayed[35]. If Hand2 memory cells exist in humans, it would be exciting to trigger these to express Shh. A Hox memory code has been described in the mouse limb[56–58] and in adult human body tissues[4,59].

We propose a model for propagating positional memory through limb regeneration (Fig. 5f). At steady state, anterior and posterior cells have fixed positional memories primed by low-level expression of spatial transcription factors including Hand2. Hand2 levels rise after amputation and induce Shh in a subset of posterior cells. Shh signalling stimulates nearby blastema cells, which have flexible positional identity, to acquire a posterior memory (including Hand2). Anterior blastema cells experiencing Shh are reprogrammed into posterior memory cells (Extended Data Fig. 10h–j). As regeneration finishes, Hand2 expression declines but is retained posteriorly, whereas Shh is extinguished. An unexpected outcome of this model is posterior dominance. It will be important to determine how excessive posteriorization is avoided during regeneration.

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

# Methods

## Ethics oversight

All animal experiments were approved by the Magistrate of Vienna (Genetically Modified Organism Office and MA58, City of Vienna, Austria), under licences GZ: 51072/2019/16, GZ: MA58-1432587-2022-12 and GZ: MA58-1516101-2023-21.

## Axolotl husbandry

Axolotls (*A. mexicanum*) were raised in individual aquaria in Vienna tap water. Axolotl matings were performed by the animal-care team at the IMP. Axolotl surgery, live imaging and tissue collection were performed under anaesthesia in 0.015% benzocaine (Merck, E-1501) diluted with Vienna tap water, using the benzocaine preparation described previously[60]. All limb amputations were performed through the middle of the lower arm (zeugopod), unless indicated otherwise. Axolotl sizes are reported in centimetres, measured from snout to tail. Axolotls up to 16 cm in length were used in experiments (an age of approximately 8 months). Axolotls were randomly allocated into experimental or control groups and housed under identical conditions, except in experiments in which the control assay was performed on one limb and the experimental perturbation on the other limb of the same axolotl. Experiments were not blinded, except to the sex of the animals.

## Axolotl genome and transcriptome reference

We used genome assembly AmexG_v6.0-DD and transcriptome assembly AmexT_v47 (ref. 61).

## Isolation of anterior and posterior dermal cells

Axolotl embryos of genotype tgSceI(Mmu.*Prrx1:TFPnls-T2A-ERT2-Cre-ERT2; Caggs:loxP-GFP-loxP-mCherry*)[Etnka] were treated with 4-OHT (Merck, H7904) as described[62] to permanently label connective tissue cells with mCherry, then raised individually until 12 cm long. Skin (containing dermal connective tissue cells) was removed from the lower arms (zeugopods) and then dissected into anterior and posterior halves, leaving a gap between them. We prepared two replicates for anterior and three for posterior, with each replicate deriving from eight axolotls (16 lower arms). Anterior and posterior samples were dissociated into single-cell suspensions using Liberase TM enzyme (Merck, 5401119001) as described[63], with the following modifications: enzymatic digestion was done for 50 min and the cells were filtered through a 50-μm Filcon filter (BD Biosciences, 340630). The mCherry-positive cells were purified from each replicate by FACS (FACSAria III Cell Sorter, BD Biosciences) using a 100-μm low-pressure nozzle and collected into separate tubes of cold amphibian culture medium[64]. Each replicate was pelleted at $300 \times g$ for 4 min at 4 °C, then resuspended in 500 μl TRIzol reagent (Thermo Fisher Scientific, 15596026). RNA was extracted according to the manufacturer's protocol and stored at −70 °C until required.

## QuantSeq library preparation and RNA sequencing

Libraries for dermal-cell RNA sequencing were prepared using QuantSeq 3′ mRNA-Seq Library Prep Kit FWD (Lexogen) with 20.25 ng of input RNA per sample. Input was 4.5 μl of input RNA plus 0.5 μl of ERCC RNA Spike-In Mix (Thermo Fisher Scientific, 4456740) pre-diluted 1:10,000 in water. Samples were multiplexed for sequencing, using i7 indices 7023 (CACACT, anterior replicate 1), 7025 (TTTATG, anterior replicate 2), 7022 (GGAGGT, posterior replicate 1), 7024 (CCGCAA, posterior replicate 2) and 7026 (AACGCC, posterior replicate 3). Each replicate was sequenced to a depth of 120 M reads, in SE 100 mode, distributed over 3 lanes of a HiSeq 2500 with v4 chemistry (Illumina). Sequencing was performed by the Next Generation Sequencing Facility at the Vienna BioCenter Core Facilities (VBCF), a member of the Vienna BioCenter (VBC), Austria. Dermal-cell RNA sequencing data have been deposited with the Gene Expression Omnibus (GEO) with accession number GSE243137.

## Gene expression analysis

Adaptor sequences were trimmed from the raw sequencing reads using Trimmomatic (v.0.39)[65], with parameters ILLUMINACLIP:Adapters.fa:2:30:7 SLIDINGWINDOW:4:20 MINLEN:40 in single-end mode. Trimmed sequenced reads were mapped to axolotl genome AmexG_v6.0-DD with HISAT2 (ref. 66), with parameters −no-unal −summary-file Output.log -k 5 −very-sensitive -x DBGenome -U Reads.fq.gz > Alignment.sam. We used featureCounts[67] to generate a read counts table. Differential expression analysis was performed on two anterior replicates and three posterior replicates using R v.4.1.2 and DESeq2 (ref. 68) v.1.34.0 with an FDR cutoff of $P < 0.01$. Volcano plots were generated using ggplot2 v.3.3.6 (ref. 69). Heatmaps were generated using the pheatmap package v.1.0.12 (R. Kolde). Gene Ontology analysis was done using the topGO package v.2.46.0 (Alexa A, Rahnenfuhrer J) using parameters ontology = "BP", geneSelectionFun = topDiffGenes, annot = annFUN.org, mapping = "org.Hs.eg.db". To calculate significant Gene Ontology terms, Fisher's exact test was used with the "elim" algorithm. To enable interpretation of the differential expression results, we generated a custom gene nomenclature derived from the AmexT_v47 transcriptome. We concatenated each axolotl gene identifier with the gene symbol for the direct human homologue where available or, if not available, the closest homologue from the NCBI non-redundant database.

## Axolotl transgenesis

Plasmids for axolotl transgenesis were assembled by Gibson Assembly, amplified using Plasmid Maxi Kits (Qiagen, 12163) and verified by Sanger sequencing before egg injection. One-cell-stage axolotl eggs were surface sterilized twice for 5 min with about 0.004% sodium hypochlorite solution (Honeywell, 71696) diluted with Vienna tap water, then washed well with fresh tap water. The following steps were performed as described in ref. 60. Eggs were de-jellied using sharp forceps in 20% Ficoll (Merck, GE17-300-05)/1X MMR/Pen-Strep (Merck, P0781) solution, then held in 10% Ficoll/1X MMR/Pen-Strep solution until microinjection. For microinjections, borosilicate glass capillary needles with filament (Harvard Apparatus, GC100F-15) were pulled using a Flaming/Brown Micropipette Puller P-97 (Sutter Instrument) with settings $P = 500$, heat = 530 (Ramp test + 30), pull = 100, velocity = 120, time = 150. Then 5 nl of the appropriate injection mix was injected into each de-jellied egg, delivered in two 2.5-nl shots. Egg injections were performed using an Olympus SZX10 microscope using a PV830 pneumatic Picopump (World Precision Instruments) with settings vacuum eject, regulator 25, range 100 ms, timed, duration 10-0. Injected eggs were transferred to 5% Ficoll/0.1× MMR/Pen-Strep solution overnight. The next morning, healthy eggs were transferred to individual wells of a 24-well multiwell plate (Thermo Fisher Scientific, 142475) filled with 0.1× MMR/Pen-Strep solution. Embryos were screened for fluorescent transgene expression at embryo stage 42 using an AXIOzoom V16 widefield microscope (Zeiss). Axolotl lines are named according to the convention established in ref. 70.

The following axolotl lines were generated by random insertion I-SceI meganuclease-mediated transgenesis: ZRS>TFP (tgSceI(*ZRS:TFPnls-T2A-ERT2-Cre-ERT2*)[Etnka]); *Prrx1*>mCherry (tgSceI(*Mmu.Prrx1:mCherry*)[Etnka]); and *Prrx1*>mCherry−Hand2 (tgSceI(*Mmu.Prrx1:mCherry-Hand2*)[Etnka]), which expresses an mCherry−Hand2 fusion protein. The injection mix was prepared according to ref. 71: transgene plasmid 1 μg, I-SceI enzyme (NEB R0694) 5 units, CutSmart buffer (NEB) 1×, water to 10 μl. In *Prrx1*>mCherry−Hand2, we fused mCherry to the amino terminus of Hand2, connected by a glycine−serine-rich linker of the sequence SGGGGSGGGGS. In ZRS>TFP, the following CMV minimal promoter was used:

>CMV minimal promoter (56 bp)

GGCGTGTACGGTGGGAGGTCTATATAAGCAGAGCTGGTTTAGTGAAC
CGTCAGATC

The following axolotl lines were generated by NHEJ-mediated CRISPR/
Cas9 knock-in: *Hand2*:EGFP (tm(*Hand2^{t/+}:Hand2-T2A-EGFP*)^{Etnka}); *Hand2*
lineage tracer (tm(*Hand2^{t/+}:Hand2-P2A-EGFP-T2A-ERT2-Cre-ERT2*)^{Etnka});
and *Alx4*:mCherry (tm(*Alx4^{t/+}:Alx4-T2A-mCherry*)^{Etnka}). We followed the
protocol in ref. 72, using the following injection mix: Cas9-NLS protein,
5 µg; gRNA, 4 µg; targeting construct, 0.5 µg; Cas9 buffer 1×; water to
10 µl. Cas9-NLS protein and buffer were synthesized by the Vienna
Biocenter Core Facilities.

The following transgenic axolotls were published previously:
tgSceI(*Caggs:loxP-GFP-dead(Stop)-loxP-mCherry*)^{Etnka} (ref. 9), tgSceI
(*Caggs:loxP-GFP-loxP-mCherry*)^{Etnka} (ref. 62), tgSceI(Mmu.*Prrx1:TFPnls-
T2A-ERT2-Cre-ERT2*)^{Etnka} (ref. 62).

### Further details on generating 3′ knock-in axolotls
We generated and characterized efficient sgRNAs targeting the last
intron of *Hand2* or *Alx4* following the protocol in (ref. 72). The sgRNAs
were produced by assembly and in vitro transcription of a synthesized
DNA template. The following forward oligos, harbouring the sgRNA
target sequence in the *Hand2/Alx4* intron (lower case) plus T7 promoter
(underlined), were at 100 µM concentration (Merck). They were PCR
amplified with the universal oligo_reverse, also at 100 µM, then purified
to generate DNA templates for in vitro transcription:

>*Hand2* sgRNA oligo_forward

GAAAT<u>TAATACGACTCACTATAGG</u>atgctgtcctctaaaccgGTTTTAGAGCT
AGAAATAGC

>*Alx4* sgRNA oligo_forward

GAAAT<u>TAATACGACTCACTATAGG</u>ttcactactggtaaatacGTTTTAGAGCT
AGAAATAGC

>Universal oligo_reverse

AAAAGCACCGACTCGGTGCCACTTTTTCAAGTTGATAACGGACTAGC
CTTATTTTAACTTGCTATTTCTAGCTCTA AAAC

In vitro transcription was performed using a MEGAscript T7 tran-
scription kit (Thermo Fisher Scientific, AM1334) and 500 ng of purified
template in a 20-µl reaction. The transcription reaction was done over-
night at 37 °C. The DNA template was removed by adding 1 µl TURBO
DNase (Thermo Fisher Scientific, AM2238) for 15 min. RNA was purified
by LiCl precipitation. Purified sgRNA was re-suspended in water at a
concentration of 1 µg µl⁻¹ and stored at −70 °C.

Knock-in constructs were generated by PCR amplification and Gibson
assembly of the following components: around 400 bp of the last intron
(harbouring the sgRNA target sequence) plus the complete last exon
of *Hand2/Alx4* minus the stop codon; the transgenes to be knocked in
(for example, T2A-EGFP) plus a stop codon; and poly(A) sequence from
SV40 (*Hand2* knock-ins) or rabbit β-globin (*Alx4* knock-in).

>*Hand2* last intron plus <u>last exon minus stop codon</u> (650 bp)

GGCCGCGGACATTAGGCGACGTAAAGAAAGGCCCATCGCAGCCGCG
GCCTGTATTtTCGCGGATAATGCCTGCGCCGCGTCTGGAGGGGCAGAT
ATAATCCCCAGCTCCACGGCAGCCCTTCAGATGTGGCGATTGCCTCGG
TTTAGAGGACAGCATTTACATAGCTTTCAGGTGAACTTGAGTATGAAT
CGCAATCACTCGTGTTGTCTTTCTCTCTCTCTGTGTATCCCCCTCCCC
CTCTCTCTTTTTATATATATATATATATATATTGCaGTTTCGCCTACAAC
TGTGGCCCTGTCTGTCTGCTAAAAAGGGGGGAATTGGCAAGTGCGTG
TTGCTGAAGGCTGTAGTGCGGTGTGTGTGCGTGTATATATGTTACGTA
GAGATACATAGATACATATCCGTGTTACGTGTTACGAATTCGTGCGTGT
GTGTGTGCGCGATTATCCGCGTTGGTTGTGAACACATGTTTGGGTCT
GCAGCAAATCAACATTCAATTGTGAGATATTGAGTTCTCTTTGCTTTT
GTCTCCCTTCCCGCTCTCTTGCCAG<u>AATGAACTCTTGAAAAGTACCG
TCGGCAGCAACGACAAGAAGAGCAAGGGCAGGACTGGCTGGCCTCAG
CACGTCTGGGCCTTGGAGCTCAAGCAG</u>

>*Alx4* last intron plus <u>last exon minus stop codon</u> (679 bp)

GACATGTAGGGGCAATCTGAAGTCCCACTCAAAGCCCACCTAGAACC
GTCCCTGCTCAGCTGGGGGAAGGCAGAATCAAATTTTGTGGAAGGCA
GTCCTGTAACTCGCACCCAGAACTCTACAGCCTGTACACTGAAATATAA

TCAAATGGTGTTGATAATTCACAATGTGATTCACTACTGGTAAATACCG
GTAACACTGAACCGCTCGAGCGACATCATCAACATATTTCAAATTGGT
ATTAATTGTATAATGTTCCTATACTCGTCTCTTGCTGTAAATCTTATTTAT
TGCCTCCAGCCCTCCAAATAGTGCCACTTTCTCATTCCTTGTCTACT
TTTGTCTTCTCCTGTTACAG<u>ATCCAGAACCCAACATGGATTGGAAACA
ACAGCGGGGGCTCTCCGGTGGCAGCCTGTGTGGTCCCCTGTGACACC
GTCCCATCCTGCATGTCTCCTCATGGCCACCCCCATGCAAGTGGAGGT
GTTTCTGAATTCCTGAGCGTGCCTAGCTCAGGAAGCCACATGGGTCAG
GCACACATGGGTAACCTCTTTGGCACTGCTGGGCTCAGCACAGGCAT
CAATGGCTACGACCTCAACGTGGAGCCAGACCGCAAGACCTCCAGCA
TCGCAGCCCTGCGGATGAAGGCCAAGGAGCACAGTGCCGCCATCTCC
TGGGCCACA</u>

### Codon alteration
The following sequences were codon-altered to enable them to be
distinguished from endogenously expressed mRNA: axolotl *Hand2*
in *Prrx1*>mCherry–Hand2 and axolotl *Shh* in BV-*Shh* baculovirus. An
axolotl codon usage table was generated using the first transcript iso-
form from each gene annotated in axolotl transcriptome assembly
AmexT_v47. Optimizer (http://genomes.urv.es/OPTIMIZER/)[73] was used
with the method 'guided random' to alter codons while still reflecting
axolotl codon usage.

>Codon-altered axolotl *Hand2* ORF (645 bp)

ATGAGCCTGGTGGGCGGTTTCCCACACCACCATCCTGGCGTGCAT
CACCACCATGAGGGCTACCCCTTTTCGGCCGCCGCAGCAACGGGGA
GATGCCACGAAGACTCGCCATACTTTCATGGTTGGCTTATCGGTCATC
CGGAGCTCTCGCCTCCCGATTATGGTCCAGGAGCACCCTACAGTCCTG
AATATGGAGGGGGGGGGCGGCCTTGAACTATGCGGGCCTGGGGGCGCGC
CAGGGGGGAGGAGCCGGAGCGCTTCTCTCAACTAGACCTGTGAAGCG
GCGAGGCACCGCTAATAGGAAGGAGCGGCGGAGAACCCAAAGCATCA
ACAGTGCTTTCGCTGAGCTCCGGGAATGTATCCCGAATGTGCCAGCC
GACACGAAGTTGTCAAAGATCAAAACTTTGCGTCTAGCCACTTCTTA
TATCGCCTACCTGATGGATTTGCTTGCCAAGGATGAGCAGTCTGAAGC
CGAAGCTTTCCGGGCAGATCTGAAACAGAGGGGAGGGGGTGGGGCT
GAGTGTAAGGAAGATAAAAGAAAGAAGGAGTTGAATGAATTGCTGAA
GTCCACAGTCGGGAGTAACGACAAGAAATCCAAGGGTCGCACCGGTTG
GCCACAGCATGTGTGGGCGCTAGAGCTCAAGCAG

>Codon-altered axolotl *Shh* ORF (1,269 bp)

ATGCGTCTCCTCCTTCGCCGGCTACTGCTGGGTACCTTGGTTTGGGC
ACTGCTAGTGCCCAGCGGCCTGACTTGCGGCCCGGGGCGTGGTATC
GGTAAAAGGAGACAGCCTAAAAAACTGACACCCCTCGCGTACAAGCA
GTTTATCCCCAACGTCGCGGAGAAGACACTGGGAGCATCTGGACGTT
ATGAGGGGAAGATCACTAGGAACTCTGACCGTTTCAAGGAGCTCACTC
CTAATTACAACCCCGACATCATTTTTAAGGACGAGGAGAATACAGGAGC
TGACCGACTGATGACTCAGAGGTGCAAAGACAAACTGAATGCCCTGGC
TATTAGCGTAATGAATCAGTGGCCGGGCGTGAAACTGCGGGTGACGGA
AGGCTGGGATGAAGATGGTCATCACAGTGAGGAGAGTCTGCATTACG
AGGGCCGAGCCGTGGATATCACAACCTCTGACCGTGACAGGTCTAAG
TATGGAATGCTGGCACGTCTGGCCGTGGAGGCAGGCTTTGATTGGGTC
TACTTCGAGTCCAAGGCCCACATACATTGCAGCGTGAAGGCGGAGAA
CAGTGTGGCAGCCAAGTCGGGAGGATGTTTTCCGGCCAGTGCTAAGGT
TACACTGGAACATGGCGTTACGAGACCAGTGAAGGATCTGCGACCCGG
AGACCGTGTGCTAGCAGCAGATGGACAAGGTCGACTGGTTTATAGCGAC
TTTCTTATGTTTCTCGACAAAGAAGAGGCAGTGACAAAGGTCTTTTAC
GTCATTGAGACGGAGAGACCAAGGCAGAGGCTAAGGTTGACAGCAGC
CCACCTCCTGTTCGCCGCAAGGCATCCCGCAAACTCATCTAGCTCCA
CCGGGGTTCCAAAGTATCTTCGCATCAAGGGTTCGACCTGGGCACCGGG
TGCTTACTGTCGACCAGGAAGGACGGGGGCTTCAGGAGGCTACTGTCA
CTCGCGTGTACCTGGAGGAGGGTGCCGGAGCCTACGCCCCCGTTACC
AGTCATGGAACCGTTGTGATTGACAAGGTACTCGCCAGTTGCTACGC
AGTGATCGAGGAGCATTCCTGGGCCCACTGGGCTTTTGCCCCTCTGCG
ACTTGGCTACGGCATACTGAGCATCTTTTCCCCTCAAGATTACAGCCC
ACATAGTCCCCCCGCGCCTAGCCAGAAAGAAGGCGTGCATTGGTACTC
AGAAATCCTGTATCATATAGGGACATGGGTGCTGCATAGCGACACTATTC
ACCCCTGGGGCATGGCCGCCAAGTCGAGT

## Generation of *Hand2* CRISPant axolotls

To generate *Hand2* CRISPants lacking a translational start, the following mix was microinjected into fertilized one-cell-stage eggs, following the protocol in ref. 72: Cas9-NLS protein, 5 μg; sgRNA1, 2 μg; sgRNA2, 2 μg; Cas9 buffer 1×; water to 10 μl. The target sequences for sgRNA1 and sgRNA2 flank the *Hand2* translational start. The control mix contained all these components except the sgRNAs. The sgRNA1 and sgRNA2 were produced by in vitro transcription using the PCR-based strategy described above.

>*Hand2* CRISPant sgRNA1 target
GCGGCCCCTGGGAGGCCC
>*Hand2* CRISPant sgRNA 2 target
ACCCCAGGGTGGTGGTGA

To assess whether *Hand2* CRISPants lose *Hand2* expression, an sgRNA1/sgRNA2 mix was injected into the left or right blastomeres of four-cell-stage *Hand2*:EGFP eggs. In this manner, half of the animal acts as an internal control (EGFP is not affected) and the other half acts as the test (to test whether EGFP is reduced). We chose to inject four-cell-stage eggs rather than two-cell-stage eggs because the latter have incomplete cell cleavage, which leads to the risk of sgRNAs leaking into the opposite half of the egg. Each injected blastomere received half the dose of the regular egg injection. EGFP fluorescence was measured when the larger of the two limb buds reached stage 46/47. Control animals were injected with the control mix lacking the sgRNAs. *Hand2*:EGFP intensity in the brighter limb bud was divided by the intensity in the dimmer limb bud to yield a fluorescence ratio.

To generate *Hand2* M146 CRISPants targeting sequences close to M146, the following sgRNAs were injected individually instead. M146 sgRNAs were produced by in vitro transcription using the PCR-based strategy described above.

>*Hand2* M146 sgRNA1 target
CCTACCTCATGGACCTGC
>*Hand2* M146 sgRNA2 target
TCATGGACCTGCTGGCCA
>*Hand2* M146 sgRNA3 target
CCAAGGACGAGCAGAGCG

## Estimation of indel frequency in *Hand2* CRISPants

Genomic DNA was individually extracted from *Hand2* CRISPant or control sibling limbs. Amputations were performed through the middle of the lower arm and each off-cut was placed in 50 μl of 50 mM NaOH. Collected tissue was heated to 95 °C for 12 min, then cooled to 4 °C in a thermocycler. Next, 5 μl of 1 M Tris, pH8 was added, then the extracted DNA was stored at 4 °C until genotyping. For genotyping, 1 μl of extracted DNA was PCR amplified using KAPA HiFi HotStart ReadyMix (Roche 07958927001) and primers that generate a roughly 750 bp amplicon surrounding the *Hand2* translational start. Then 30 ng of PCR-amplified and purified DNA was Sanger sequenced using *Hand2*_sequencing primer at the IMP/IMBA Molecular Biology Service. The indel frequency was estimated from the Sanger-sequencing results using the ICE Analysis Tool (Synthego, https://ice.synthego.com/).

>*Hand2*_genotyping_F
GAAGTAGCAGGGATGGACGAG
>*Hand2*_genotyping_R
AAGGCGCTGTTGATGCTCT
>*Hand2*_sequencing
CACAGGCCAGGACTTCAAGAA

## 3-Primer genotyping strategy to identify *Hand2*[Δ64] mutants

Genomic DNA was extracted individually from tail clips as described above in the section above. PCR amplification was performed with the following 3-primer mix and KAPA2G Fast HotStart Genotyping mix (Roche 2GFHSGKB). The PCR reaction produces a roughly 752 bp amplicon as a positive control and also a 431 bp amplicon if the *Hand*

translational start is intact. Homozygous *Hand2*[Δ64] mutants fail to amplify a 431 bp amplicon but amplify the larger amplicon.

>*Hand2*_genotyping_F
GAAGTAGCAGGGATGGACGAG
>*Hand2*_genotyping_R
AAGGCGCTGTTGATGCTCT
>*Hand2*_genotyping_R2
CCCCCCACCAAGCTCATG

## Determination of axolotl ZRS enhancer

The axolotl ZRS enhancer was determined by multiple species alignment of the following genome sequences using mVISTA[74] and Pip-Maker[75]:

axolotl (*A. mexicanum*) assembly, AmexG_v6.0-DD chr2p:694366863-694689506;
human (*Homo sapiens*) assembly, hg38 chr7:156769228-156790956;
mouse (*M. musculus*) assembly, mm10 chr5:29292950-29323801;
chick (*G. gallus*) assembly, Gal6 chr2:8538956-8559114;
fugu (*Takifugu rubripes*) assembly, fr3 chr10:5739579-5747090.
>Axolotl ZRS enhancer, AmexG_v6.0-DD chr2p:694613160-694613980 (821 bp)

ACCTTAATATCCATCTTTGCATTTGAAGTTGTTGCATAAAATGTACCA
CGAGCGACAGCAACATCCTGACTAATTAGCCAAATTACCCAGACATCC
CTCCAAAAAAGCCGCGAAACAGAGAGCATGTCTGTCGGATTAAAAGGT
TGTAACTCCTAAAACATCAAACGGAGCGCCAGATAATAAAAGCCAATC
GTACAGAAATTTGAGGTAACTTCCTTGCTTAATTAATTAGCTAGGCCA
GTTGGAGCGAGGAGGCCAACGCGGGCGCGTAGAACGCCCATAAAGCT
GAACAACTCGACAGCACAAAAGTGGAGAAACAAAGATTTTTTAATATGC
GTCTATCCTGTGTCACAGTTTGAAATTGTCCTGGTTTATGTCCCTTTTGG
CAAAGTTACAATAAAAGTGACCCTGTACTGTATTTTATGGCCAGACGACT
TTTCGTTTTGTTCCCGGTGACTAATTTGACTCAGGCCCCCATCTTGAAT
AGACACAGAAAGGGGCCGGGGGAATGAGGCTGTCTGTCTCGCTTGGG
TTTCATTGCATTTTTTCATTATTCGGGCTCGTTTTTCGCCACAGATCAT
CCATAAATTGTTGGAAATGAGTGATTAAGGAAGTGCTGCTTAATGTTAG
TAGCACACATTCTTTGTGCGTTTCACCCTCCCGCCCCCTCCATTTTGT
GGGTGAGAGGAAATCAAGTAATGCAGAAACAATAAGGAAGCCTCCTGC
TGGGAACCTTTCAAGGAAATGTAACCTGCATACTGTTTTGATCTCGGTG
TTCCTTTCAGAGTATGCCGCGATGTTTCAACAGCTATTTTCATGTG

## Genetic lineage tracing (ZRS/*Hand2*)

For lineage tracing, either ZRS>TFP axolotls or *Hand2* lineage-tracing axolotls were mated with *loxP*–mCherry fate-mapping axolotl of genotype tgSceI(*Caggs:loxP-GFP-dead(Stop)-loxP-mCherry*)[Etnka] (ref. 9). To induce Cre–*loxP* recombination during development, stage-42 ZRS or *Hand2* lineage-tracing embryos were bathed overnight in the dark in 500 ml of 2 μM 4-OHT, as described in the water-based method of ref. 60. We calculated the overlap between embryonic *Shh* cells (mCherry⁺) and regeneration *Shh* expression (TFP⁺) using wide-field microscopy. To induce Cre–*loxP* recombination in *Hand2* cells of the mature limb, 7-cm axolotls were bathed individually and overnight in the dark in 100 ml of 5 μM 4-OHT. After treatment, the axolotls were transferred to tap water and allowed to recover for one week. The same 5 μM 4-OHT treatment and one week recovery was repeated twice more, for a total of three treatments. Animals were screened for Cre–*loxP* recombination and mCherry expression using an AXIO-zoom V16 wide-field microscope (Zeiss). Regeneration experiments on ZRS>TFP lineage tracings were performed on 11-cm axolotls. Regeneration experiments on *Hand2* lineage tracings were performed on 6-cm axolotls.

## Quantification of ZRS>TFP fluorescence

An integrated fluorescence score was calculated for each *Hand2* CRISPant limb bud (mean TFP intensity × area), then normalized to the mean of the control cohort (set to 1). Quantifications were performed on stage-42 limb buds imaged using wide-field microscopy.

## Assessment of leaky mCherry expression

ZRS/*Hand2* lineage tracing animals were generated as described above and genotyped individually to ensure that they carried both Cre and *loxP-Stop-loxP*-mCherry genetic cassettes. Primers used for genotyping were:

>Cre_Fw
ATCCGAAAAGAAAACGTTGA
>Cre_Rv
ATCCAGGTTACGGATATAGT
>Cherry_Fw
GGATAACATGGCCATCATCAAGGAGTTC
>Cherry_Rv
GTCTTGACCTCAGCGTCGTAGTG

Half of the animals were treated at limb-bud stage 42 with 2 μM 4-OHT overnight to induce recombination and mCherry labelling. The other half were left untreated, to assess for nonspecific mCherry expression (leakiness). When the animals reached 6 cm, their limbs were removed, dissociated with Liberase TM enzyme (see the section 'Quantifying *Hand2*:EGFP expression during regeneration' below) and analysed for mCherry fluorescence using a flow cytometer, a 100-μm low-pressure nozzle and FLOWJO software (BD Biosciences). Untreated and treated samples were analysed in the same session. At least 130,000 events were recorded for each sample.

## Surgical depletion of embryonic *Shh* cells

ZRS lineage-traced axolotls were prepared by treating stage-42 embryos with 2 μM 4-OHT, as described in the section 'Genetic lineage tracing (ZRS/*Hand2*)', above. At an axolotl size of 6 cm, the left arm of each axolotl was depleted for ZRS-lineage cells by using microscissors to excise tissue posterior to the ulna in the lower arm. Successful depletion was confirmed by imaging loss of mCherry fluorescence using an AXIOzoom V16 wide-field microscope (Zeiss). The right arm of each animal was treated as a control and depleted for an equivalent amount of tissue anterior to the radius instead. Two days after surgery, each arm was amputated through the distal part of the lower arm, distal to the depleted region. Images were acquired every few days after amputation to assess the onset of ZRS>TFP expression in depleted versus control limbs. The mCherry depletion efficiency was estimated by comparing the area of mCherry-positive tissue in the blastema in wide-field images acquired from control and depleted animals.

## Tissue preparation, staining and imaging

Samples were fixed overnight in 4% paraformaldehyde, pH 7.4, at 6 °C. The next morning, samples were washed well with cold PBS then equilibrated with the following solutions at 6 °C for one overnight each: 20% sucrose/PBS; 30% sucrose/PBS; and a 1:1 mix of 30% sucrose/PBS and Tissue-Tek O.C.T. compound (Sakura). Samples were mounted in Tissue-Tek O.C.T. compound, frozen on dry ice and stored at −70 C until sectioning. Cryosections of 16 μm thickness were prepared using a Cryostar NX70 (Thermo Fisher Scientific) and collected on Superfrost Plus adhesion microscope slides (Epredia, J1800AMNZ). Slides were stored at −20 °C until required. For staining, slides were brought to room temperature then washed well with PBS to remove O.C.T. compound before proceeding to the following steps. For DAPI staining only, slides were incubated with DAPI 1:1,000 in PBS + 0.2% Triton X-100 for 1 h at room temperature, then washed well with PBS + 0.2% Triton X-100 before mounting. For staining with anti-Prrx1 or anti-Col1A1 antibody, slides were blocked for 30 min at room temperature with PBS + 0.2% Triton X-100 + 1% normal goat serum (NGS), then incubated overnight at 6 °C with rabbit anti-Prrx1 antibody[62] diluted 1:500 in PBS + 0.2% Triton X-100 + 0.1% NGS or mouse anti-Col1A1 antibody (SP1.D8, DSHB) diluted 1:50 in PBS + 0.2% Triton X-100 + 0.1% NGS. The following day, slides were washed well with PBS + 0.2% Triton X-100 then incubated for 2 h at room temperature with Alexa 647-conjugated anti-rabbit

(Invitrogen, A-21244) or anti-mouse (Invitrogen, A-21240) secondary antibody diluted 1:500 in PBS + 0.2% Triton X-100. Slides were washed well with PBS + 0.2% Triton X-100 before mounting. For HCR (hybridization chain reaction) in situ hybridization, slides were stained according to the HCR RNA-FISH protocol for fixed frozen tissue sections (Molecular Instruments), omitting post-fixation and Proteinase K treatment. Probe hybridization buffer, wash buffer and amplification buffer were from Molecular Instruments. Samples were mounted in Abberior Mount liquid antifade mounting media (Abberior) for imaging. Images were acquired with an LSM980 AxioObserver inverted confocal microscope with ZEN software (Zeiss), plus AiryScan 2 for HCR experiments only.

## Whole-mount sample preparation, staining and imaging

Samples were fixed overnight in 4% paraformaldehyde, pH 7.4, at 6 °C. The next morning, samples were washed well with cold PBS then dehydrated progressively through ice-cold 25% methanol/PBS, 50% methanol/PBS, 75% methanol/PBS and 100% methanol (30 min each). Samples were kept for one night in 100% methanol at −20 °C. The next day, samples were rehydrated progressively through ice-cold 75% methanol/PBS, 50% methanol/PBS, 25% methanol/PBS and PBS (30 min each). Samples were washed twice more with cold PBS then stained for *Shh* transcripts using the HCR RNA-FISH protocol for whole-mount zebrafish embryos and larvae (Molecular Instruments), starting at the section 'Detection stage'. After completion of the HCR protocol, samples were stained overnight at 6 °C in DAPI 5 mg ml⁻¹ (Sigma) diluted 1:1,000 in 5× SSC + 0.1% Tween 20. The next day, samples were washed well with 5× SSC + 0.1% Tween 20 then optically cleared overnight at room temperature on an aerial rotator in clearing-enhanced 3D (Ce3D) solution, refractive index 1.50, prepared as described previously[76]. Images were acquired in Ce3D solution using a LightSheet.Z1 microscope with ZEN software (Zeiss) and custom imaging chamber, as described[77].

## HCR probe design and detection

HCR probes targeting axolotl *Shh* mRNA, probe hybridization buffer, wash buffer, detection hairpins and amplification buffer were from Molecular Instruments. Sequences unique to *Shh* mRNA were identified by BLAST alignment against axolotl transcriptome assembly Amex.T_v47 (ref. 61). Sequences were considered not unique if they exhibited homology to non-target transcripts at more than 36 of 50 nucleotides. *Shh* HCR signal was detected using B5 hairpins conjugated to Alexa-647 fluorophore. HCR probes targeting TFP mRNA were purchased at the 50-pmol scale from IDT (Integrated DNA Technologies, oPools), suspended in water and stored at −20 °C. TFP HCR signal was detected using B1 hairpins conjugated to Alexa-546 fluorophore.

## Quantifying *Hand2*:EGFP expression during regeneration

**For whole-tissue measurements.** Axolotls (4.5 cm) harbouring *Hand2*:EGFP were amputated through the middle of the lower arm and imaged throughout regeneration with identical acquisition settings using an AXIOzoom V16 wide-field microscope (Zeiss). Longitudinal imaging of 6 limbs was done on days 0, 2, 4, 7, 10, 15, 21, 28 and 39 after amputation. Mean EGFP fluorescence intensity was measured in Fiji software[78] by manually drawing a region of interest in the EGFP-positive region of the blastema. At 0, 2 and 4 d.p.a., no or little blastema had formed, so measurements were instead taken at 500 μm behind the amputation plane. The 500-μm source zone for lower-arm regeneration was established in ref. 79.

**For single-cell measurements.** The *Hand2*:EGFP intensity of mature arm cells, 7 d.p.a. blastema cells and 14 d.p.a. blastema cells were compared by flow cytometry. Lower-arm tissue was removed from 6-cm *Hand2*:EGFP axolotls. The entire lower arm was removed for mature measurements. Blastemas were generated by amputating through the middle of the lower arm 7 or 14 days before flow cytometry. Tissues removed were then dissociated into single-cell suspensions

using Liberase TM enzyme (Merck, 05401127001) as described previously[63], with the following modifications: dissociation was done for 55 min (mature sample) or 45 min (blastema samples), and the cells were filtered through a 70-μm MACS SmartStrainer (Miltenyi Biotec, 130-098-462). Cells were analysed by FACS (using a FACSAria III Cell Sorter, BD Biosciences) with a 100-μm low-pressure nozzle. Mean fluorescence intensities were quantified using FLOWJO software (BD Biosciences).

## ALM

ALM experiments were performed on the upper arm of the axolotl, as described[80]. For *Hand2* CRISPant ALMs, donor axolotls (F$_0$ *Hand2* CRISPant) were 9–10 cm in size and host axolotls (GFP-expressing controls) were 13–14 cm. Donor skin grafts (1 mm × 1 mm) were transplanted distal to the deviated nerve on host animals. Donor grafts were removed from *Hand2* CRISPants deemed to have a high mutation rate, as judged by regeneration of a hypomorphic spike after a previous lower-arm amputation. As controls, skin grafts were removed from sibling axolotls injected with Cas9 but no sgRNA. *Hand2* CRISPant ALM and control grafts were done on opposite arms of each host axolotl. Then, 58 days after surgery, ALMs were removed and fixed for skeletal staining using Alcian blue and Alizarin red. For *Hand2* M146 CRISPant ALMs, the procedures were similar, except that donor axolotls (F$_0$ *Hand2* M146 CRISPants generated using sgRNAs 1–3, or sibling controls injected with control mix) were 7 cm in size and host axolotls (d/d) were also 7 cm. ALMs were analysed 27 days after grafting. *Hand2* misexpression ALMs were performed similarly. Donor axolotls (strong *Prrx1*>*Hand2*) were 5 cm and host axolotls (unlabelled controls) were 8 cm. As controls, skin grafts were removed from weak *Prrx1*>*Hand2* siblings. ALM formation was deemed to have been completed by 34 days after surgery.

## Alcian blue and Alizarin red staining

Skeletal staining of fixed accessory limbs was performed as described previously[81]. Stained limbs were imaged in 70% ethanol/water using an AxioCam ERc 5s colour camera (Carl Zeiss Microimaging) mounted on an Olympus SZX10 microscope. Alcian blue (A3157), Alizarin red (A5533) and Trypsin (85450C) were from Merck.

## Cell transplantation by injection of FACS-sorted cells

Upper and lower arms were removed from 4-cm double-reporter axolotls (*Alx4*:mCherry_*Hand2*:EGFP) and dissociated into single-cell suspensions using Liberase TM enzyme (Merck, 05401127001) as described[63], with the following modifications: dissociation was done for 55 min and the cells were filtered through a 70-μm MACS SmartStrainer (Miltenyi Biotec, 130-098-462). Anterior cells (mCherry-positive plus EGFP-negative) or posterior cells (EGFP-positive) were purified by FACS (FACSAria III Cell Sorter, BD Biosciences) using a 100-μm low-pressure nozzle and collected into separate tubes of amphibian culture medium[64]. Pelleting and injection of FACS-sorted cells into the arms of 4-cm unlabelled sibling axolotls was done as described[63], using a Nanoject II injector (Drummond Scientific Company). Anterior cells were injected into the posterior lower arm, whereas posterior cells were injected into the anterior lower arm. We injected 9,000 cells per experiment. Host arms were imaged 2 days after injection to confirm successful transplantation using an AXIOzoom V16 wide-field microscope (Zeiss). At 15 days, host arms were amputated distal to the transplant. Regenerating blastemas were imaged at 6, 8, 10, 15 and 26 d.p.a. until the limb was considered fully regenerated. At 26 d.p.a., a second amputation was done through the regenerated part of the limb to generate a second blastema and test whether cells had altered their positional memory. This second blastema was removed at 8 d.p.a. and fixed and processed for whole-mount HCR staining against *Shh* transcripts.

## Isolation of cells for RNA sequencing

**For *Hand2*-misexpression experiments.** We amputated 16-cm *Prrx1*>mCherry–Hand2 (test) or *Prrx1*>mCherry (control) axolotls

through the mid-zeugopod. The animals also had ZRS>TFP in the genetic background, but this was not used in this experiment. We analysed *Prrx1*>mCherry–Hand2 limbs with weaker phenotypes, because strong misexpression resulted in no limb. At 14 d.p.a., the anterior part of each blastema was removed and dissociated into single cells using Liberase TM enzyme, as described above. We FACS-purified mCherry-positive cells into amphibian culture medium[64] using one animal (two blastemas) per replicate. There were three replicates for *Hand2* misexpression blastemas and two replicates for mCherry controls.

**For A→A and A→P cells.** We took upper and lower arms from uninjured 7.5-cm double-reporter axolotls (*Alx4*:mCherry_*Hand2*:EGFP) and dissociated them into single cells using Liberase TM enzyme, as described above. We purified mCherry-positive, EGFP-negative anterior cells for injection. We did cell injections as described above. About 25,000 sorted events were injected into the anterior or posterior zeugopod of unlabelled, uninjured sibling animals. Two days after injection, limbs were amputated through the distal part of the graft to induce regeneration. After 47 days, the regenerated limb was re-amputated through the distal part of the graft to induce a second blastema. This second blastema was removed at 12 d.p.a. Blastemas were dissociated into single cells as described above and prepared for FACS. In A→A experiments, we purified mCherry-positive anterior cells regardless of EGFP expression. In A→P experiments, we purified EGFP-positive posterior cells regardless of mCherry expression. Cells were sorted into amphibian culture medium[64]. One injected arm was used per replicate. By the time of removal, animals were approximately the same size as those used for anterior and posterior cell isolation (below).

**For anterior and posterior cells.** We amputated 11-cm double-reporter axolotls (*Alx4*:mCherry_*Hand2*:EGFP) at the mid-zeugopod. At 15 d.p.a., whole blastemas were removed and dissociated into single cells using Liberase TM enzyme, as described above, and prepared for FACS. To isolate anterior blastema cells, we purified mCherry-positive, EGFP-negative cells. To isolate posterior blastema cells, we purified EGFP-positive cells, regardless of mCherry expression. Anterior and posterior blastema cells in the same replicate were isolated from the same blastema preparation (paired samples). We used seven animals to generate four replicates each of anterior and posterior blastema cells. Cells were sorted into amphibian culture medium[64].

## RNA library preparation, sequencing and analysis

We proceeded immediately to RNA isolation after cell sorting. RNA was purified into 20 μl nuclease-free water using an in-house magnetic bead-based isolation kit. The following RNA inputs were used to construct libraries: *Hand2* overexpression and mCherry control (0.7 ng each); anterior/posterior high input (45 ng each); anterior/posterior low input (4.5 ng each); and A→A and A→P (4.5 ng each). Sequencing libraries were constructed using QuantSeq 3′ mRNA-Seq V2 REV kit with unique dual indices (Lexogen). The low-input protocol was used for samples with less than 10 ng input, and RNA removal was reduced to 5 min for the 0.7 ng libraries, as suggested by the manufacturer. Sequencing libraries were purified into 18 μl nuclease-free water. Each replicate was sequenced to a depth of around 50 M reads, in PE 150 mode (read 2 + CSP-read1), distributed over two lanes of a NovaSeq X 1.5 B flowcell. Sequencing was done by the Next Generation Sequencing Facility at Vienna BioCenter Core Facilities (VBCF), part of the Vienna BioCenter (VBC), Austria. These sequencing data have been deposited at the Gene Expression Omnibus (GEO) with accession number GSE284768.

Gene expression analysis was done using DESeq2, similarly to the dermal cell samples (see the 'Gene expression analysis' section, above). A threshold of 100 counts was used. PCA was performed using anterior/

posterior low-input libraries (to match the inputs of the A→A and A→P samples), and the other gene expression comparisons were done against the anterior/posterior high-input libraries. We used a threshold of *padj* < 0.05 for significant differential expression.

## Dilution and storage of BMS-833923 and SAG

BMS-833923 (Cayman Chemical, 16240) was dissolved to 10 mM in ethanol and stored as single-use aliquots at −70 °C. InSolution Smoothened Agonist (SAG, Merck 566661) for intraperitoneal injections was obtained at 10 mM in water and stored as single-use aliquots at −20 °C. SAG for bathing experiments was purchased from Merck (566660), diluted to 40 μM and stored as single-use aliquots at −20 °C.

## Skin transplantation plus intraperitoneal delivery of BMS-833923

Several modifications were made to the assay relative to the anterior-to-posterior and posterior-to-anterior transplantations by cell injection. One limitation of the *Alx4*:mCherry_*Hand2*:EGFP double reporter was that cells would lose fluorescence if they lost anterior and posterior identity. To avoid this, we substituted *Alx4*:mCherry for *Prrx1*>mCherry, which expresses mCherry regardless of positional identity. This enabled continuous monitoring of the transplant, and *Hand2*:EGFP labelled posterior identity. Owing to limited animal availability, we used hindlimbs as donors instead of forelimbs (positional memory in forelimb and hindlimb are compatible[82]). To reduce the number of donors required, we transplanted cells by skin grafting instead of cell injection.

Skin areas (about 1 mm × 1 mm) were transplanted from the anterior lower leg of 7-cm double-reporter axolotls (*Prrx1*>mCherry_*Hand2*:EGFP) to the posterior lower arm of 8-cm unlabelled *d/d* hosts, maintaining dorsal–ventral and proximal–distal directionality. Four days after transplantation, the host arm was amputated through the distal third of the transplant, and blastema outgrowth was monitored every 1–2 days using an AXIOzoom V16 wide-field microscope (Zeiss). At 6 d.p.a. (the conical blastema stage), test axolotls were injected intraperitoneally with 25 μl of BMS-833923 diluted to 1 mM in water. Control axolotls instead received the appropriate dilution of ethanol in water injection. Injection mix contained Fast Green dye (Thermo Fisher Scientific) for visual contrast. Blastemas were further imaged 4 and 15 days after injection to assess for changes in positional identity.

## Assessing blocking *Shh* signalling on *Hand2*:EGFP expression

*Hand2*:EGFP axolotls (7 cm) were amputated at the top of the lower arm. Every 3 days from 0 d.p.a. until 21 d.p.a., test axolotls were injected intraperitoneally with 20 μl of BMS-833923 diluted to 1 mM in water, whereas control axolotls were instead injected with 20 μl of water. Injection mix contained Fast Green dye (Thermo Fisher Scientific) for visual contrast. Blastemas were imaged every three days using an AXIOzoom V16 wide-field microscope (Zeiss). Mean *Hand2*:EGFP fluorescence was quantified from manually defined regions of interest in the posterior blastema.

## SAG positional-memory experiment

*Hand2* lineage-traced axolotls were prepared by treating stage-42 embryos with 2 μM 4-OHT, as described in the section 'Genetic lineage tracing (ZRS/*Hand2*)', above. At a size of 8 cm, each axolotl had the right arm amputated through the middle of the lower arm (blastema assay). The left arm was left intact (mature assay). At 8 d.p.a., test axolotls were injected intraperitoneally with 20 μl of SAG diluted to 1.5 mM in water. Control axolotls instead received a water injection. Injection mix contained Fast Green dye (Merck F7258) for visual contrast. Both the blastema limb and the mature limb were imaged every few days until 25 days after injection using an AXIOzoom V16 wide-field microscope (Zeiss). On day 25, both limbs were amputated through the hand-plate

region to assay for effects on positional memory from the expression of *Hand2*:EGFP.

## SAG positional-memory experiment (*Shh* HCR assay)

*Hand2*:EGFP axolotls (5 cm) were amputated through both lower limbs, then bathed in water (control) or 10 nM SAG (test) for the first 21 days of regeneration. Bathing volume was 40 ml, and solution was prepared and exchanged daily, following the protocol of ref. 1. Regeneration was deemed to be complete at 30 d.p.a. Axolotls were raised for a further 30 days in water, to ensure complete washout of SAG from test animals. Subsequently, axolotls were re-amputated through the hand-plate region to generate a new blastema in the reprogrammed part of the limb. At 9 d.p.a., the new blastemas were removed and fixed for whole-mount HCR staining against *Shh* mRNA, tissue clearing and light sheet imaging.

## Baculovirus production and injection

Pseudotyped baculovirus was produced as described in ref. 83. BV-mCherry, a control baculovirus to misexpress mCherry, was published previously as *ch*BV[83]. The cytomegalovirus immediate-early promoter (pCMV) drives expression of mCherry in infected cells. BV-*Shh*, to misexpress axolotl *Shh*, was generated for this study. pCMV drives the expression of nuclear-localized mCherry T2A axolotl *Shh*. Co-translational cleavage in the T2A sequence releases full-length axolotl *Shh* protein. Axolotl *Shh* was codon-altered to enable the distinction of virally expressed mRNA from endogenous axolotl *Shh* mRNA.

Either BV-mCherry or BV-*Shh* was injected into the anterior lower arm of 4-cm *Hand2*:EGFP axolotls. The injection mix contained Fast Green dye (Merck, F7258) for visual contrast. Then, 18 days after infection, limbs were amputated through the middle of the lower arm. The regenerating blastema was imaged every few days using an AXIOzoom V16 wide-field microscope (Zeiss). At 11 d.p.a., blastemas were removed for fixation, whole-mount tissue clearing and imaging using a LightSheet. Z1 microscope (Zeiss).

## Image analysis

Microscope images were analysed using ZEN software (Zeiss) or Fiji software[78].

## Statistical analysis and data representation

Statistical analyses and graph plotting were done using Prism software (GraphPad). Data were tested for assumptions of normality and equality of variance to determine the appropriate statistical tests to perform. Measurements were taken from distinct samples unless indicated otherwise. No data were excluded. Mean values are reported ± s.d. Statistical significance was defined as $P < 0.05$. All figures were made using Adobe Illustrator.

## Reporting summary

Further information on research design is available in the Nature Portfolio Reporting Summary linked to this article.

## Data availability

Genome assembly AmexG_v6.0-DD (https://genome.axolotl-omics.org/index.html) and transcriptome assembly AmexT_v47 (https://www.axolotl-omics.org/assemblies) were used[61]. All RNA-sequencing data have been deposited at the Gene Expression Omnibus (GEO) under accessions GSE243137 (dermal cell data) and GSE284768 (all other data). Source data are provided with this paper.

## Code availability

The analyses were performed using previously described computational tools, as described in the section 'Gene expression analysis'.

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

**Acknowledgements** We thank members of the Tanaka laboratory for discussions (P. Tardivo and A. Polikarpova), assistance with RNA-seq library preparation (F. Becerril Perez and M. Leyva), bioinformatics (S. Nowoshilow) and baculovirus production (H. Okulski). We thank C. Lilliehook, O. Chara, K. Lust, E. Bassat and W. Masselink for comments on the manuscript; the animal care team (M. Blaschek, E. Silic, V. Szilagyi, A. Lentz-Koblenc, J. König, V. Vojnicsek, D. Bastian, E. Kiligan, E. Zöllner, T. Torrecilla Lobos, S. Faltin and W. Auer) for axolotl care; staff at the BioOptics facility and the Molecular Biology Service at the IMP/IMBA Core Facilities for support; and staff at the Next Generation Sequencing facility at the Vienna Biocenter Core Facilities for RNA sequencing. This work was funded by Human Frontier Science Program (HFSP) fellowship LT000785/2019-L to L.O.; an Österreichische Forschungsförderungsgesellschaft (FFG) FEMtech Praktika studentship to S.A.P.; and a European Research Council (ERC) advanced grant 742046 (RegGeneMems) to E.M.T. For the purpose of open access, the author has applied a CC BY public copyright licence to any Author Accepted Manuscript version arising from this submission.

**Author contributions** L.O. and E.M.T conceived the project and secured funding. L.O. performed and analysed all experiments, with support from S.A.P. for Extended Data Fig. 5c–e. L.O. and Y.T.-S. microinjected axolotl eggs to generate transgenic axolotls. F.F. aligned RNA sequencing data to the axolotl genome and generated gene-count matrices. L.O. and E.M.T. wrote and revised the manuscript. All authors approved the manuscript.

**Funding** Open access funding provided by Research Institute of Molecular Pathology (IMP) / Institute of Molecular Biotechnology (IMBA)/ Gregor Mendel Institute of Molecular Plant Biology (GMI).

**Competing interests** The authors declare no competing interests.

**Additional information**
**Correspondence and requests for materials** should be addressed to L. Otsuki or E. M. Tanaka.

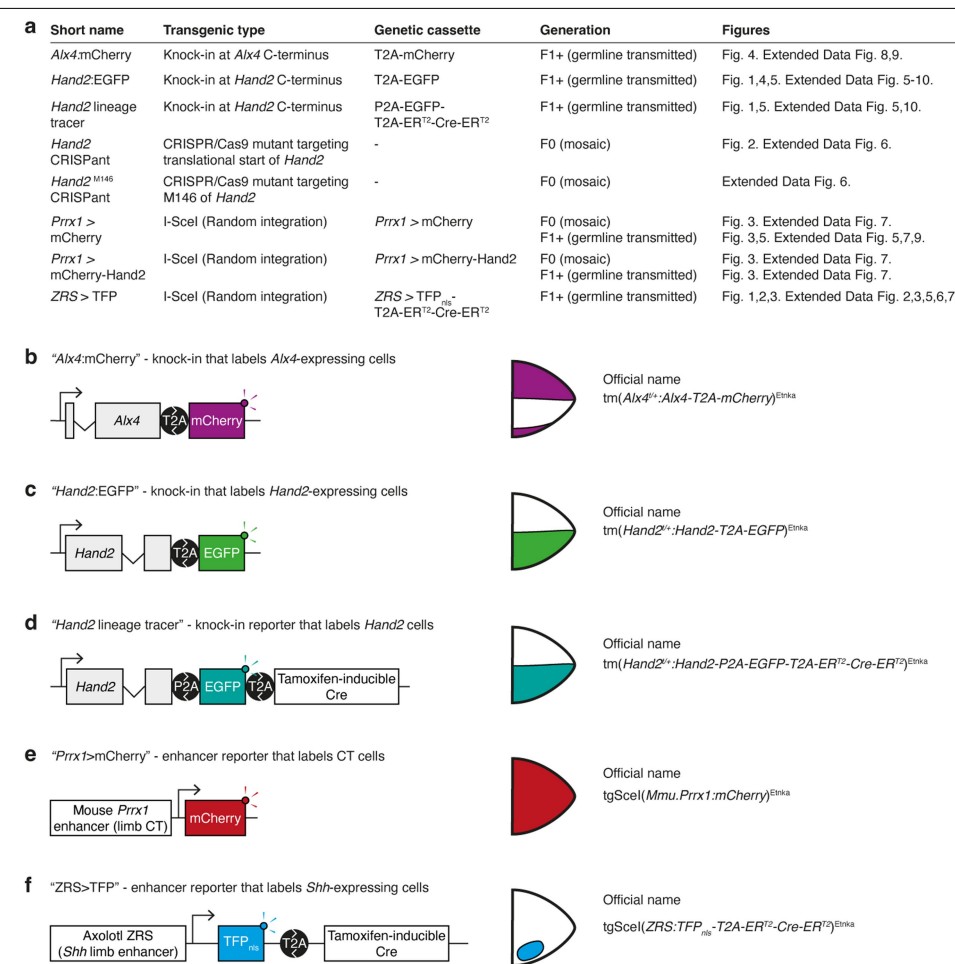

**Extended Data Fig. 1 | Summary of the transgenic axolotls generated in this study. a**, Transgenic axolotls generated in this study, and the figures that they appear in. **b-f**, The genetic constructs used to generate axolotls for this study (*left*), their expression domains in the limb bud/blastema (*centre*) and official nomenclature (*right*). In the expression domain schematics, only mesenchyme is depicted and anterior is up, posterior is down.

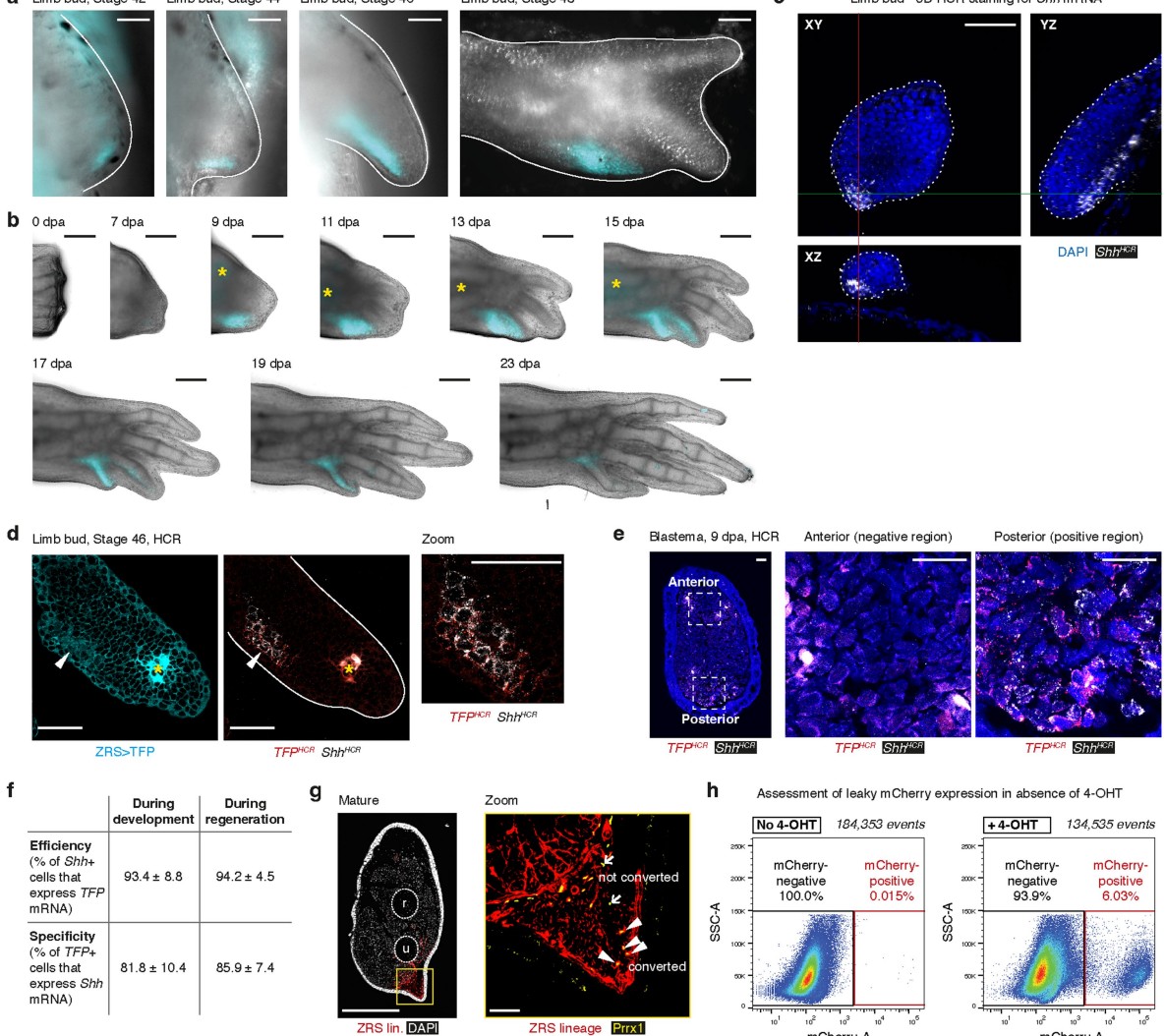

**Extended Data Fig. 2 | Characterisation of the ZRS > TFP transgenic axolotl.**
**a,b**, ZRS > TFP expression (cyan) during limb development (**a**) and regeneration
(**b**). TFP perdures beyond the *Shh* expression window. Asterisks indicate
autofluorescence in the cyan channel. **c**, Orthogonal views of a limb bud stained
for *Shh* mRNA (white) and DAPI (blue), then 3D imaged to show specificity of the
*Shh* probe. **d**, Confocal section of a ZRS > TFP limb bud, stained for TFP mRNA
(red) and *Shh* mRNA (white). Endogenous TFP fluorescence (cyan) was
preserved poorly. Arrowhead indicates zoomed area. Asterisk indicates
autofluorescence. **e**, Confocal cross-section of a ZRS > TFP blastema (4 cm
animal), stained for TFP mRNA (red) and *Shh* mRNA (white). **f**, Quantification of
ZRS > TFP reporter efficiency and specificity. Data are mean ± SD. Development

data (**d**): *n* = 8 limb buds, 182 TFP+ cells and 154 *Shh*+ cells quantified.
Regeneration data (**e**): *n* = 17 blastemas, 1,318 TFP+ cells and 1,199 *Shh*+ cells
quantified. **g**, Estimation of Cre/loxP recombination efficiency. Confocal
cross-section of a lower forelimb depicting lineage traced cells in red and nuclei
in white. The boxed region is magnified to the right. Prrx1 antibody (yellow)
stains connective tissue cells. Recombination efficiency was defined as the
percentage of Prrx1+ cells posterior to the ulna bone that were mCherry+.
r: radius. u: ulna. **h**, Estimation of mCherry leakiness in ZRS lineage tracing
experiments using flow cytometry. *n* = 12 4-OHT-treated animals and *n* = 12
untreated animals. Images are representative of 8 (**a,b,d**), 6 (**c**), 12 (**e**) or 9 (**g**)
specimens. Scale bars: 100 μm (**a,c,d,e,g**) or 500 μm (**b**).

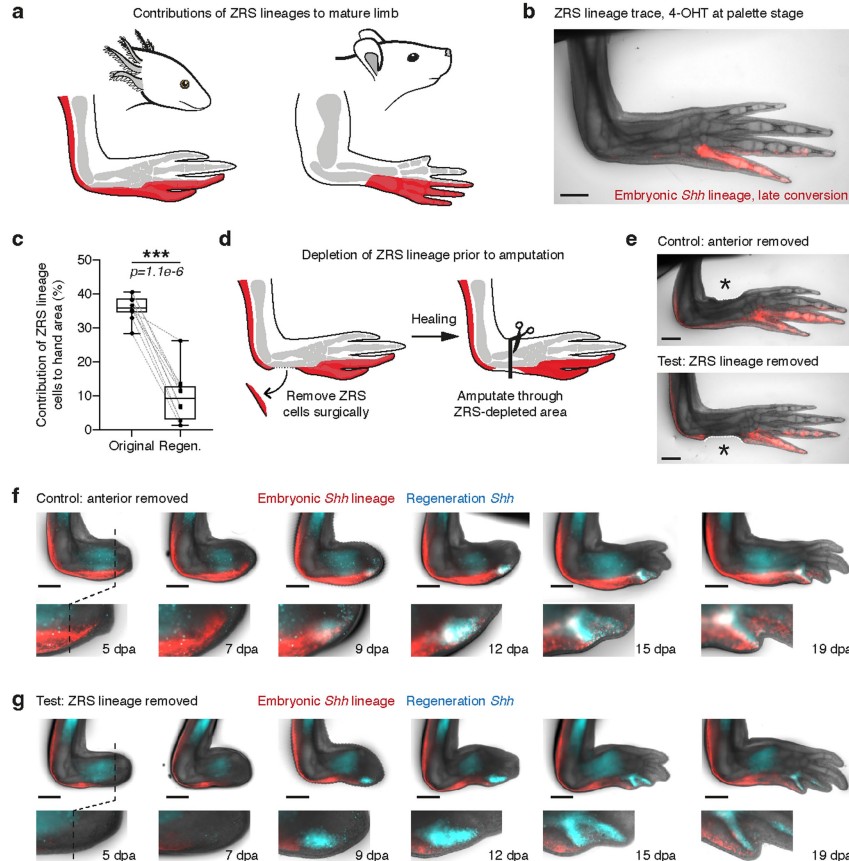

**a**, Contributions of ZRS lineages to mature limb

**b**, ZRS lineage trace, 4-OHT at palette stage

Embryonic *Shh* lineage, late conversion

**c**

***
p=1.1e-6

Contribution of ZRS lineage cells to hand area (%)

Original    Regen.

**d**, Depletion of ZRS lineage prior to amputation

Remove ZRS cells surgically — Healing — Amputate through ZRS-depleted area

**e**, Control: anterior removed

Test: ZRS lineage removed

**f**, Control: anterior removed — Embryonic *Shh* lineage — Regeneration *Shh*

5 dpa    7 dpa    9 dpa    12 dpa    15 dpa    19 dpa

**g**, Test: ZRS lineage removed — Embryonic *Shh* lineage — Regeneration *Shh*

5 dpa    7 dpa    9 dpa    12 dpa    15 dpa    19 dpa

**Extended Data Fig. 3 | Characterisation of the axolotl ZRS lineage.**
**a**, Schematic summarising the lineage tracing of embryonic *Shh* cells into the forelimb in axolotls (*left*, this study) or in mice (*right*, Harfe et al. [53]). **b**, Lineage tracing of embryonic *Shh* cells converted at limb bud stage 47 with 4-OHT (*n* = 8 limbs). **c**, Contribution of embryonic *Shh* cells to the hand area following development ("Original") or after amputation and regeneration ("Regen."). Boxplots depict median, interquartile range and 1.5x interquartile range. Each dot is a measurement from 1 limb. Data from the same limb are connected by a dotted line. Paired two-tailed *t*-test, *n* = 10 limbs per condition. **d**, Strategy to deplete embryonic *Shh* cells prior to amputation. For control animals, anterior tissue was removed instead. **e**, Control and *Shh* cell-depleted limbs immediately prior to amputation. Asterisks indicate surgery sites. *n* = 6 limbs per condition. **f**, Regeneration of control limbs. *Top row*: overview. *Bottom row*: blastema zoom. Dotted line indicates amputation plane. 6/6 limbs expressed ZRS > TFP (cyan) at 7 dpa. **g**, Regeneration of *Shh* lineage-depleted limbs. Dotted line indicates amputation plane. 6/6 limbs expressed TFP (cyan) at 7 dpa. Scale bars: 1 mm.

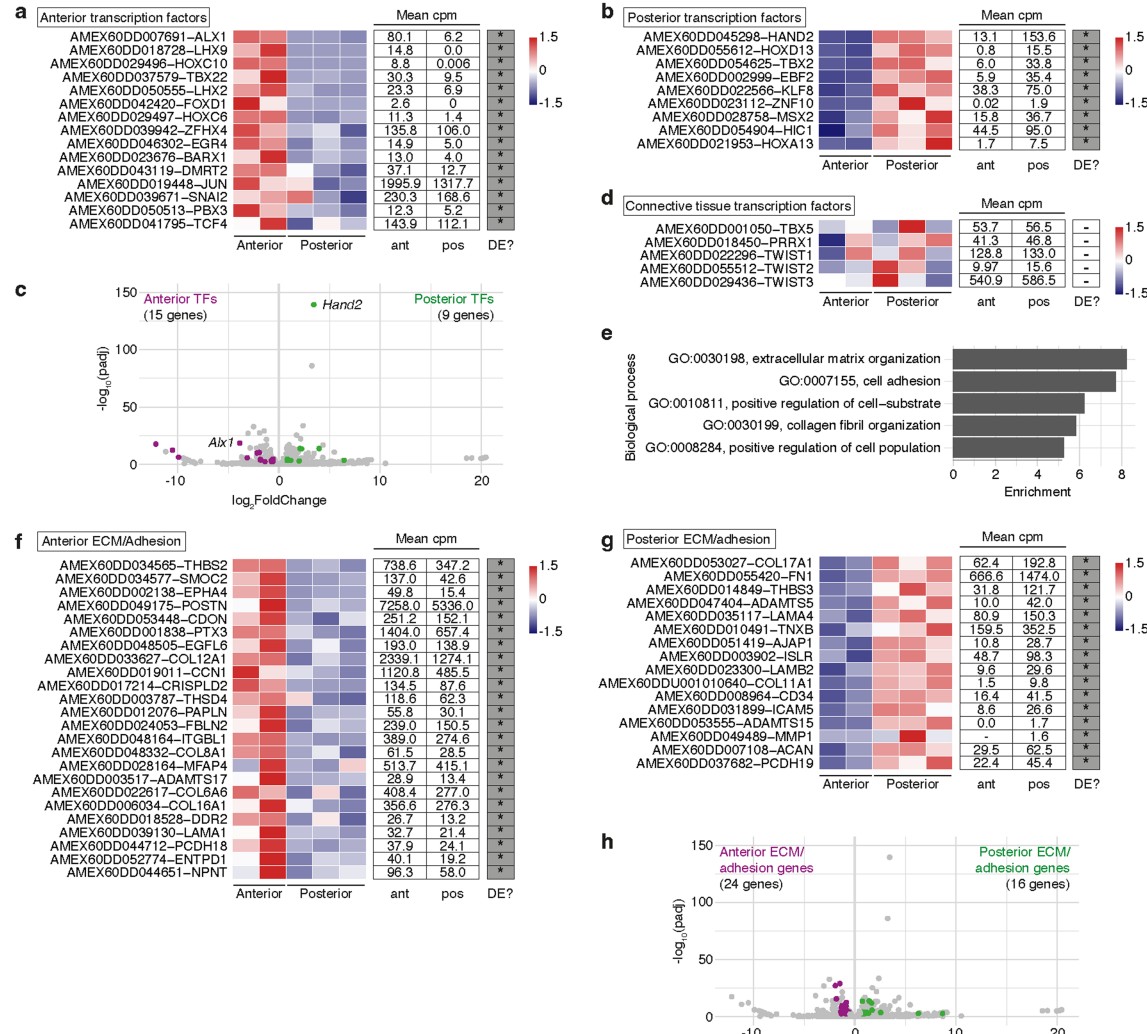

**Extended Data Fig. 4 | Gene expression in anterior and posterior dermal cells. a-b**, Heatmaps depicting the relative expression of transcription factors enriched in anterior cells (**a**) or posterior cells (**b**). Genes are ordered by decreasing statistical significance. Heatmap scores are normalised by row. cpm: counts per million. ant and pos: anterior and posterior samples. An asterisk and dark grey box in the DE? column indicates statistically significant differential expression. **c**, Volcano plot highlighting the transcription factors differentially expressed in anterior cells (magenta) or in posterior cells (green). The most statistically significant transcription factor in each population is labelled. Differential expression analysis was performed with DESeq2 (two-tailed Wald test with Benjamini-Hochberg adjustment for multiple testing), with a FDR cutoff of $p < 0.01$. **d**, Heatmap of general connective tissue and forelimb identity genes. **e**, Top 5 Gene Ontology (GO) terms describing the differentially expressed genes. **f,g**, Heatmaps depicting the relative expression of differentially expressed genes belonging to the GO term categories 'extracellular matrix' or 'cell adhesion' in anterior cells (**f**) or posterior cells (**g**). **h**, Volcano plot highlighting differentially expressed genes belonging to the GO term categories 'extracellular matrix' or 'cell adhesion'. Differential expression was determined statistically using DESeq2 (two-tailed Wald test with Benjamini-Hochberg adjustment for multiple testing), with a FDR cutoff of $p < 0.01$. Gene nomenclature: we concatenated AMEX[....], the axolotl gene identifier, with the gene symbol of the human homologue.

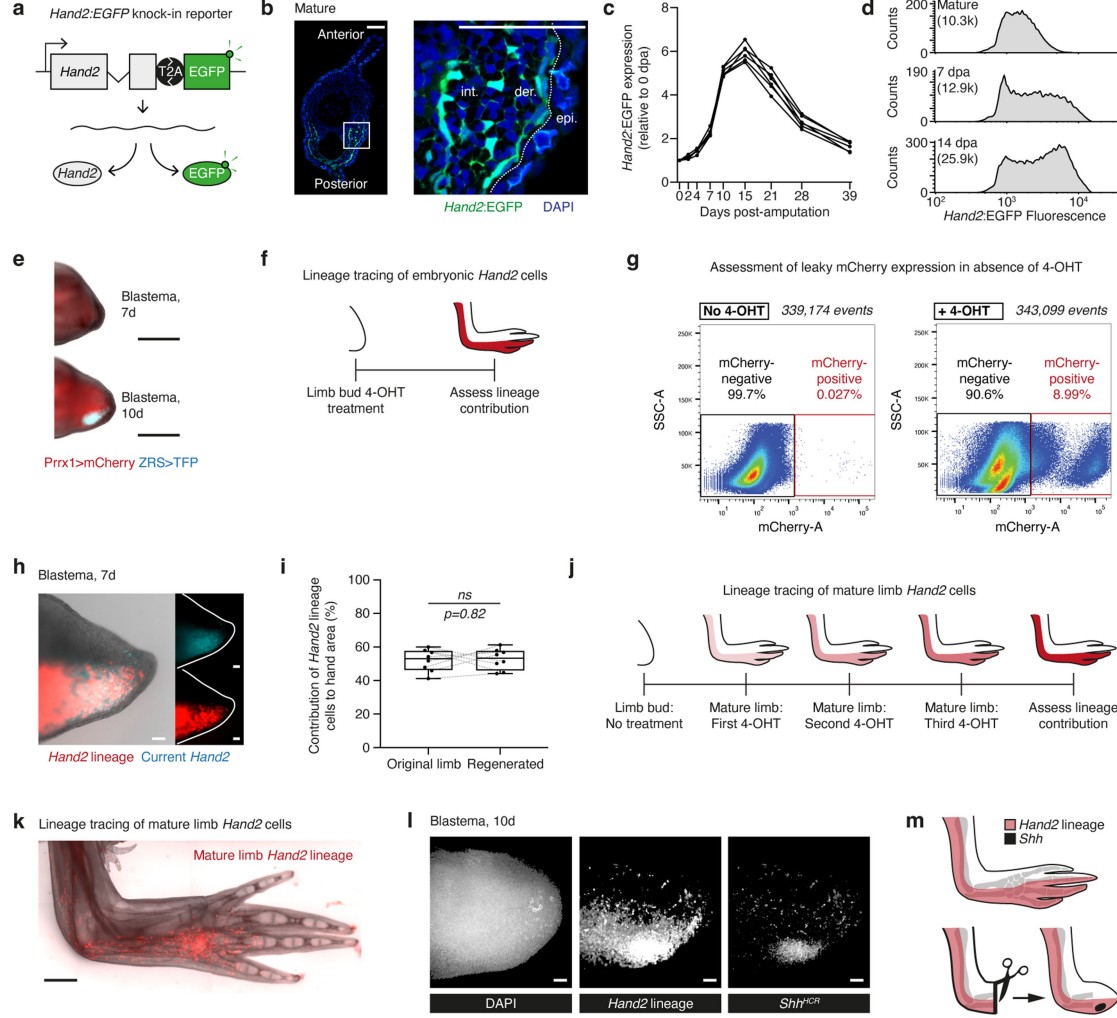

**Extended Data Fig. 5 | *Hand2* lineage tracing in limb bud and mature limb.**
**a**, Schematic of the *Hand2*:EGFP knock-in reporter construct. **b**, Cross-section through the lower forelimb of a *Hand2*:EGFP axolotl. Boxed region is magnified to the right. Dotted line indicates border between epidermis (epi) and dermis (der). Int: interstitial cells. **c**, *Hand2*:EGFP expression after amputation. Each line connects data from 1 limb. *n* = 6 limbs. **d**, *Hand2*:EGFP levels in mature limb and blastema, measured using flow cytometry. Only EGFP+ events are displayed. **e**, Comparison of ZRS > TFP expression at 7 dpa (positive signal in 1/7 axolotls) and 10 dpa (positive signal in 7/7 axolotls). *Prrx1*>mCherry (red) labels connective tissue cells. **f**, Lineage tracing strategy to label embryonic *Hand2* cells. **g**, Estimation of mCherry leakiness in *Hand2* lineage tracing experiments. *n* = 12 4-OHT-treated animals and *n* = 12 untreated animals. **h**, A blastema

lineage traced for embryonic *Hand2* cells (red). Current *Hand2* expression is in cyan. **i**, Contribution of embryonic *Hand2* cells to the hand area following development ("Original limb") or after amputation and regeneration ("Regenerated"). Boxplots depict median, interquartile range and 1.5x interquartile range. Each dot is a measurement from 1 limb. Paired data are connected by a dotted line. Paired two-tailed *t*-test, *n* = 8 limbs. *ns*: not significant. **j**, Lineage tracing strategy to label steady state *Hand2* cells. **k**, *Hand2* cells labelled following the treatment in (**j**). **l**, Split channel images of the *Hand2* lineage-traced blastema in main Fig. 1l. **m**, Schematic illustrating the contributions of embryonic *Hand2* cells to the limb and to *Shh* cells during regeneration. Images are representative of 6 (**b,k**), 7 (**e,h**) or 4 (**l**) specimens. Scale bars: 100 μm (**b,e,h,l**) or 1 mm (**k**).

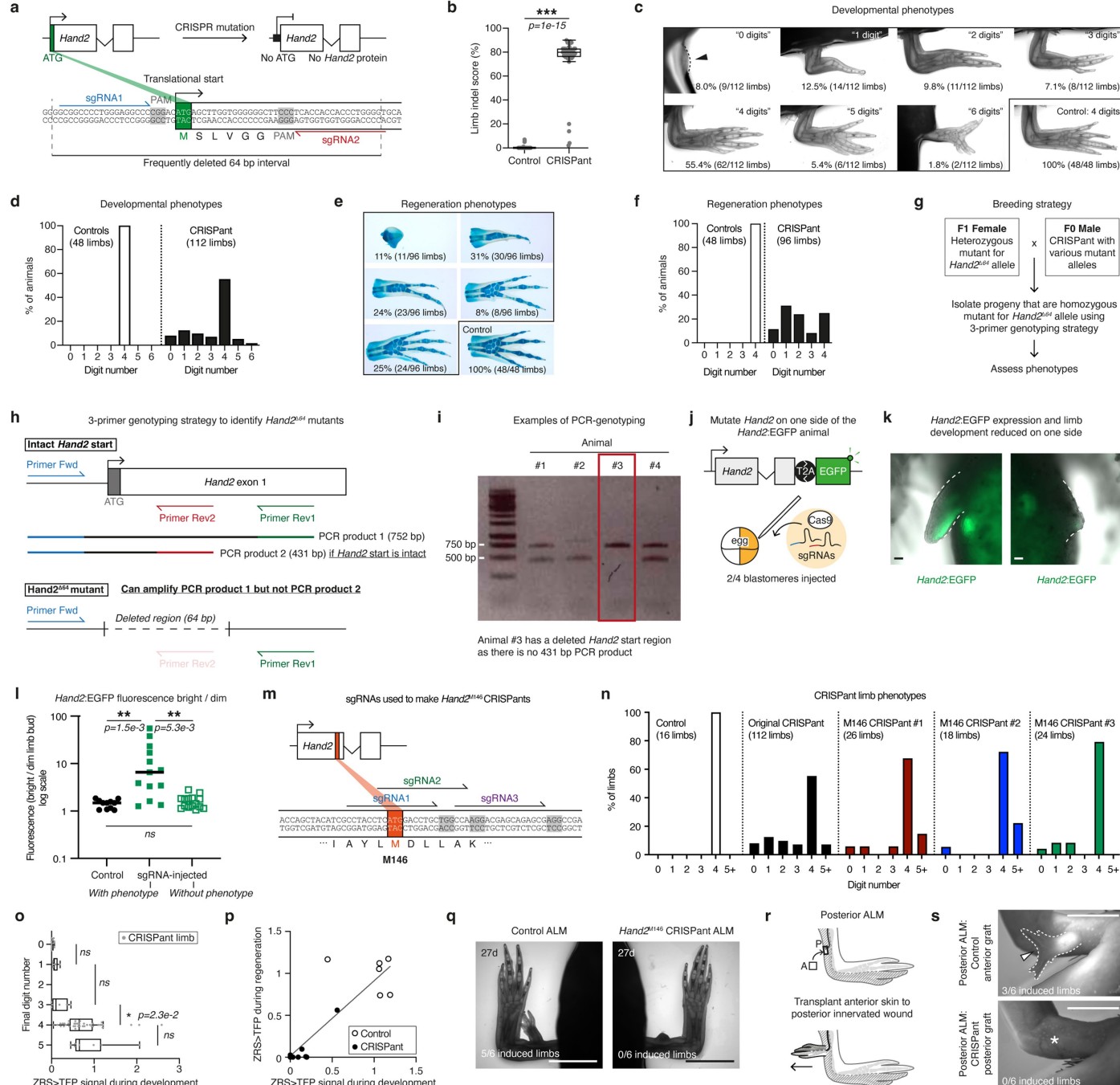

**Extended Data Fig. 6 | Characterisation of *Hand2* CRISPants during development, regeneration and ALM. a**, Strategy to generate *Hand2* CRISPants. **b**, Indel frequency in control (*n* = 46) or *Hand2* CRISPant (*n* = 47) axolotls. Two-tailed Kolmogorov-Smirnov test. **c,d**, Developmental phenotypes (**c**) and quantifications (**d**). **e,f**, Regeneration phenotypes (**e**) and quantifications (**f**). **g,h,i**, Breeding (**g**) and genotyping (**h,i**) strategies to isolate *Hand2^Δ64^* homozygotes. Lethality: 96.7% (29/30, homozygotes) vs 19.4% (12/62, background rate including trans-heterozygotes undetectable with this strategy). **j,k**, Injection of *Hand2* sgRNAs into one side of *Hand2*:EGFP eggs (**j**) and resulting phenotype in 13/28 axolotls (**k**). **l**, Comparison of *Hand2*:EGFP intensity in left and right limb buds. Ratios: 1.34 ± 0.30 (controls, *n* = 11), 13.31 ± 16.50 (*Hand2* CRISPants with defects, *n* = 13), 1.61 ± 0.51 (*Hand2* CRISPants without defects, *n* = 15). Ratios are mean ± SD. Two-tailed Kruskal-Wallis test with Dunn's multiple comparisons. Plots depict medians. **m**, Target

sequences for *Hand2* M146 sgRNAs. **n**, Digit phenotypes in *Hand2^M146^* CRISPants, controls and *Hand2* CRISPants. **o**, Comparison between limb bud ZRS > TFP expression (normalised to controls) and final digit number. Two-tailed Kruskal-Wallis test with Dunn's multiple comparison testing, *n* = 68 axolotls. **p**, Paired comparison between ZRS > TFP intensity in limb bud and blastema (two-tailed Spearman's rank test, *p* = 2.4e-3, *n* = 14 axolotls). **q**, Anterior ALM using control or *Hand2^M146^* CRISPant grafts (*n* = 6 each). There was a significant difference (number of digits formed, two-tailed Wilcoxon signed rank test, *p* = 0.03, W = −21.00). **r**, Schematic depicting the posterior ALM. **s**, Posterior ALM induction by control or *Hand2^M146^* CRISPant grafts (*n* = 6 each). There was a significant difference (number of digits formed, two-tailed Wilcoxon signed rank test, *p* = 0.03, W = −4.00). Boxplots (**b,o**) depict median, interquartile range and 1.5x interquartile range. ns: not significant, *p* > 0.05. Scale bars: 100 μm (**k**) or 5 mm (**q,s**).

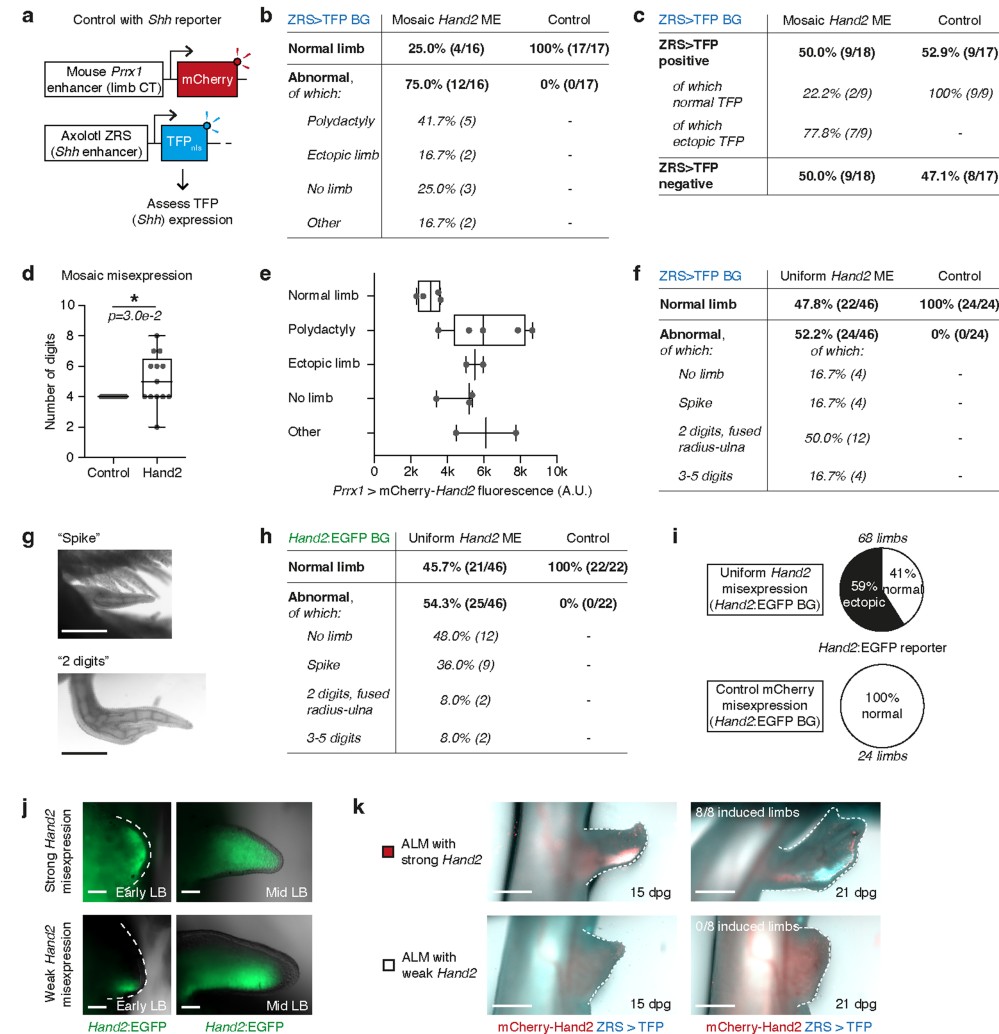

**Extended Data Fig. 7 | Characterisation of limbs misexpressing _Hand2_.**
**a**, Schematic of a control axolotl misexpressing mCherry in a ZRS > TFP background. **b**, Limb phenotypes arising from mosaic _Hand2_ misexpression (ME) in the ZRS > TFP genetic background (BG). **c**, Effects of mosaic _Hand2_ misexpression on ZRS > TFP expression. **d**, Boxplot comparing the number of digits in limbs mosaically misexpressing _Hand2_ compared to controls. Two-tailed Kolmogorov-Smirnov test, _n_ = 17 (controls) or 13 (_Hand2_) limbs. **e**, Comparison of mCherry-_Hand2_ intensity in mosaic experiments with resulting limb phenotypes. _n_ = 16 limbs total. The trend seen was not significant, likely due to low _n_ numbers in some categories (**b**, two-tailed Kruskal-Wallis test, _p_ > 0.05). **f**, Effects of uniform _Hand2_ misexpression (ME) on digit number in the ZRS > TFP genetic background (BG). **g**, Examples of "spike" and "2 digits, fused

radius-ulna" phenotypes quantified in (**f**). **h**, Effects of uniform _Hand2_ misexpression (ME) on digit number in the _Hand2_:EGFP genetic background (BG). **i**, Effects of uniform _Hand2_ misexpression on _Hand2_:EGFP expression. **j**, Comparison of _Hand2_:EGFP expression (green) in axolotls uniformly misexpressing strong mCherry-_Hand2_ (_top_) or weak mCherry-_Hand2_ (_bottom_). Images are representative of 40 and 28 limbs respectively. **k**, Anterior ALM performed with anterior skin grafts misexpressing strong _Hand2_ (_top_) or weak _Hand2_ (_bottom_). 8/8 strong _Hand2_ ALMs induced ZRS > TFP at 21dpg, compared to 0/8 weak _Hand2_ grafts. Boxplots (**d,e**) depict median, interquartile range and 1.5x interquartile range. Each dot is a measurement from one limb. Scale bars: 1 mm (**g,k**) or 100 μm (**j**).

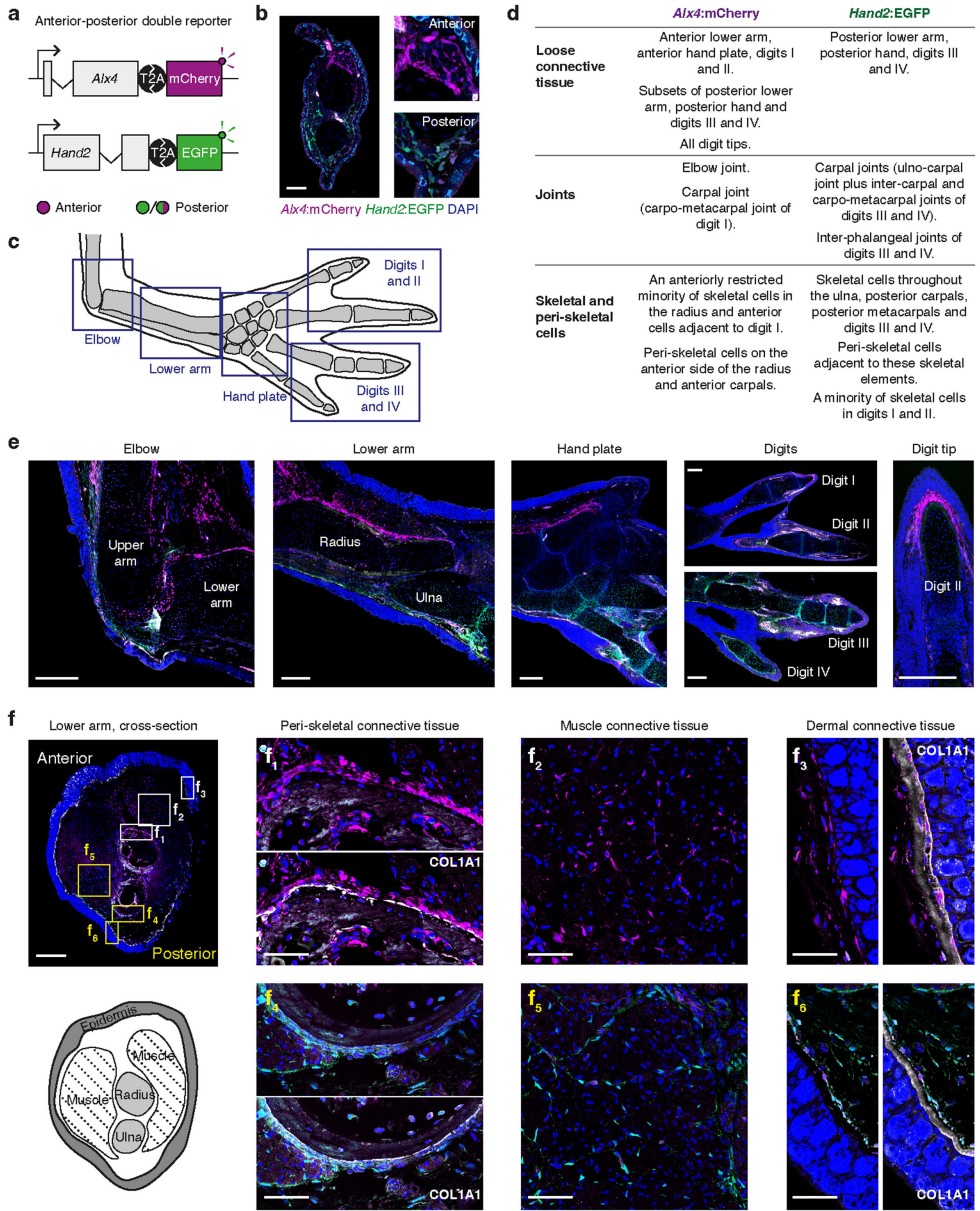

**Extended Data Fig. 8 | Characterisation of *Alx4*:mCherry_*Hand2*:EGFP double reporter expression. a**, Schematic of an *Alx4*:mCherry (magenta)_*Hand2*:EGFP (green) double knock-in axolotl to distinguish anterior and posterior cells. **b**, Cross-section through the lower arm of an *Alx4*:mCherry_*Hand2*:EGFP axolotl, with nuclei stained by DAPI (blue). Insets show magnifications of anterior and posterior tissue. **c**, Schematic of the limb sectioned and imaged in (**e**). **d**, Expression domains of *Alx4*:mCherry and *Hand2*:EGFP reporters in loose connective tissue, joints, skeletal elements and peri-skeletal elements, assessed from (**e**). **e**, Longitudinal confocal sections of a limb from a 15 cm double reporter axolotl. Magenta is endogenous *Alx4*:mCherry fluorescence,

Green is endogenous *Hand2*:EGFP fluorescence and Blue is nuclear staining (DAPI). See (**c**) for anatomy. **f**, Confocal cross-section of a limb from a 15 cm double reporter animal. $f_1$-$f_6$ are magnifications of the boxed regions in the overview panel, and highlight different connective tissue populations. $f_{1\text{-}3}$ are anterior regions (predominantly *Alx4*:mCherry-positive) and $f_{4\text{-}6}$ are posterior regions (*Hand2*:EGFP-positive). Colours are as in (**e**). White signal is antibody staining against COL1A1, which highlights peri-skeletal and dermal cell niches. Images are representative of 6 (**b**) or 3 (**e,f**) limbs. Scale bars: 100 μm (**b**,$f_1$-$f_6$) or 500 μm (**e,f**).

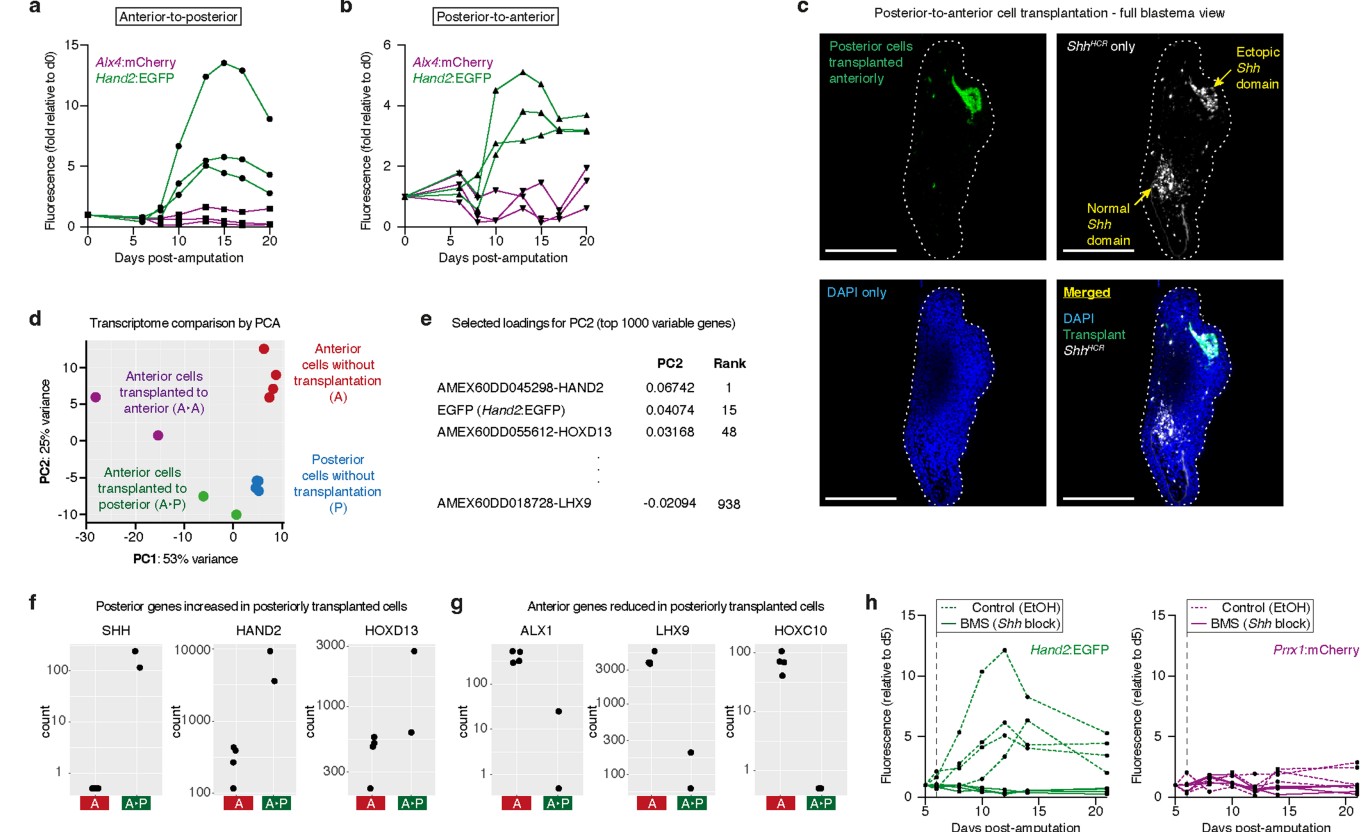

**Extended Data Fig. 9 | Characterisation of the anterior-posterior positional memory switch. a,b**, Quantification of *Alx4*:mCherry and *Hand2*:EGFP fluorescence intensities in anterior-to-posterior transplantations (**a**) or posterior-to-anterior transplantations (**b**), normalised to 0 dpa values. Each line connects data from 1 limb. *n* = 3 limbs per transplantation type. **c**, Low magnification image of the posterior-to-anterior transplanted blastema depicted in main Fig. 4h, representative of 3 transplantations. **d**, Principal component analysis (PCA) of RNA-seq datasets. **e**, Selected gene loadings for principal component 2 (PC2). The top 1000 variable genes were used as an input. **f**, Expression of selected posterior genes in anterior blastema cells (A) or anterior cells transplanted posteriorly (A→P), normalised to sequencing depth. All genes are differentially expressed as assessed by DESeq2, *padj* < 0.05. **g**, Expression of selected anterior genes in anterior blastema cells (A) or anterior cells transplanted posteriorly (A→P), normalised to sequencing depth. All genes are differentially expressed as assessed by DESeq2, *padj* < 0.05. **h**, *Hand2*:EGFP or *Prrx1*>mCherry expression in anterior-to-posterior transplantations treated with BMS-833923 (*Shh* pathway inhibitor) or ethanol (control), normalised to 5 dpa values. Dotted vertical line indicates time of injection. Each line connects data from 1 limb. *n* = 4 limbs per condition. Scale bars: 500 µm.

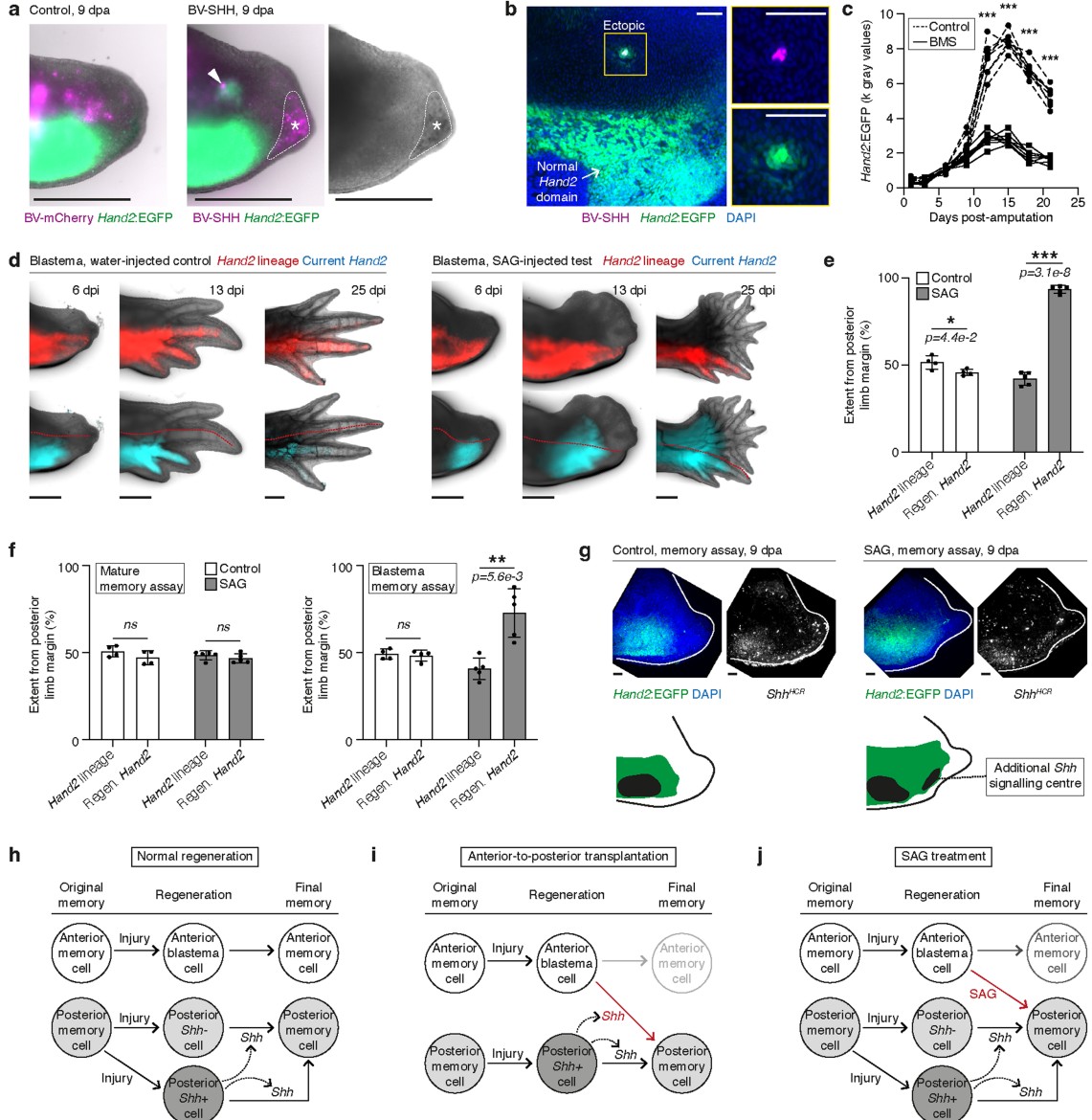

**Extended Data Fig. 10 | *Shh* signalling can induce a posterior memory in anterior blastema cells. a**, *Hand2*:EGFP limbs infected with baculoviruses encoding *Prrx1*>mCherry (BV-mCherry) or *Prrx1*>mCherry-T2A-*Shh* (BV-SHH). 7/10 BV-SHH-injected blastemas expressed anterior *Hand2*:EGFP after amputation (arrowhead), compared to 0/10 controls. Asterisk indicates autofluorescence. **b**, The BV-SHH blastema in (**a**), imaged with lightsheet microscopy. The boxed region is magnified to the right. **c**, Mean *Hand2*:EGFP expression in axolotls injected with BMS-833923 or water (control). Two-way ANOVA with Šídák multiple comparisons, F(7, 98) = 178.0. $n$ = 8 blastemas per condition. *** from left to right: $p$ = 3.0e-6, 3.1e−11, 7.5e-11, 2.1e-7. **d**, *Hand2* lineage-traced axolotls treated with SAG or water. Red labels embryonic *Hand2* cells (posterior cells). Cyan labels active *Hand2* expression. Representative of $n$ = 4 (control) or 5 (SAG) blastemas. **e**, Quantification of (**d**). The anterior extent of mCherry (*Hand2* lineage) and *Hand2*:EGFP (regeneration *Hand2*) were measured from the posterior edge in 25 dpi blastemas. Two-way ANOVA with Šídák multiple comparisons, F(1, 7) = 458.4. $n$ = 4 blastemas (control) or 5 blastemas (SAG). Data are presented as mean values ± SD. **f**, Same quantification as in (**e**), but for the positional memory assay (see main Fig. 5d). Two-tailed paired $t$-tests, $n$ = 4 blastemas (control) or 5 blastemas (SAG). Mean values ± SD. *ns*: not significant, $p$ > 0.05. **g**, Control and SAG-treated blastemas stained for *Shh* mRNA (white). Ectopic *Shh* signalling centres were seen in 4/6 cases (SAG) and 0/6 cases (controls). **h**, Original and regenerated anterior-posterior memory are broadly concordant during normal limb regeneration. **i**, In anterior-to-posterior transplantations, anterior cells come into proximity of *Shh*-expressing posterior cells, which posteriorize cell memory (red arrow). **j**, SAG treatment activates *Shh* signalling in anterior blastema cells, posteriorizing their positional memory (red arrow). Scale bars: 1 mm (**a,d**) or 100 μm (**b,g**).

# Reporting Summary

## Statistics

For all statistical analyses, confirm that the following items are present in the figure legend, table legend, main text, or Methods section.

| n/a | Confirmed | |
|---|---|---|
| ☐ | ☒ | The exact sample size (*n*) for each experimental group/condition, given as a discrete number and unit of measurement |
| ☐ | ☒ | A statement on whether measurements were taken from distinct samples or whether the same sample was measured repeatedly |
| ☐ | ☒ | The statistical test(s) used AND whether they are one- or two-sided *Only common tests should be described solely by name; describe more complex techniques in the Methods section.* |
| ☐ | ☒ | A description of all covariates tested |
| ☐ | ☒ | A description of any assumptions or corrections, such as tests of normality and adjustment for multiple comparisons |
| ☐ | ☒ | A full description of the statistical parameters including central tendency (e.g. means) or other basic estimates (e.g. regression coefficient) AND variation (e.g. standard deviation) or associated estimates of uncertainty (e.g. confidence intervals) |
| ☐ | ☒ | For null hypothesis testing, the test statistic (e.g. *F*, *t*, *r*) with confidence intervals, effect sizes, degrees of freedom and *P* value noted *Give P values as exact values whenever suitable.* |
| ☐ | ☒ | For Bayesian analysis, information on the choice of priors and Markov chain Monte Carlo settings |
| ☐ | ☒ | For hierarchical and complex designs, identification of the appropriate level for tests and full reporting of outcomes |
| ☒ | ☐ | Estimates of effect sizes (e.g. Cohen's *d*, Pearson's *r*), indicating how they were calculated |

*Our web collection on statistics for biologists contains articles on many of the points above.*

## Software and code

Policy information about availability of computer code

| Data collection | No custom software was used for data collection. |
|---|---|
| Data analysis | Differential expression analysis of RNA-sequencing data. Adaptor sequences were trimmed from the raw sequencing reads with Trimmomatic (version 0.39), using parameters: ILLUMINACLIP:Adapters.fa:2:30:7 SLIDINGWINDOW:4:20 MINLEN:40 in single-end mode. Trimmed sequenced reads were mapped to axolotl genome AmexG_v6.0-DD with HISAT2, using parameters: --no-unal-summary-file Output.log -k 5-very-sensitive -x DBGenome -U Reads.fq.gz Alignment.sam. featureCounts was used to generate a read counts table. Differential expression analysis was performed on 2 anterior replicates and 3 posterior replicates using R version 4.1.2 and DESeq2 version 1.34.0 with an FDR cutoff of p < 0.01. Volcano plots were generated using ggplot2 version 3.3.6. Heatmaps were generated with the pheatmap package version 1.0.12 (R. Kolde). Gene Ontology analysis was performed with the topGO package version 2.46.0 (Alexa A, Rahnenfuhrer J) using parameters: ontology= "BP", geneSelectionFun = topDiffGenes, an not= annFUN.org, mapping= "org.Hs.eg.db". To calculate significant GO terms, Fisher's exact test was performed with the "elim" algorithm. To facilitate interpretation of the differential expression results, we generated a custom gene nomenclature derived from the AmexT _ v47 transcriptome. We concatenated each axolotl gene identifier with the gene symbol for the direct human homologue where available or, if not available, the closest homologue from the NCBI non-redundant database. Determination of axolotl ZRS enhancer The axolotl ZRS enhancer was determined by multiple species alignment of the following genome sequences using mVISTA (https://genome.lbl.gov/vista/mvista/submit.shtml) and PipMaker (http://pipmaker.bx.psu.edu/pipmaker/). Axolotl (Ambystoma mexicanum) assembly AmexG_v6.0-DD  chr2p:694366863-694689506 Human (Homo sapiens) assembly hg38 chr7:156769228-156790956 Mouse (Mus musculus) assembly mm10 chr5:29292950-29323801 |

Chick (Gallus gallus) assembly Gal6 chr2:8538956-8559114
Fugu (Takifugu rubripes) assembly fr3 chr10:5739579-5747090

Estimation of indel frequency in Hand2 CRISPants
Indel frequency was estimated from Sanger sequencing results using the ICE Analysis Tool (Synthego, https://ice.synthego.com/).

Image analysis
Microscope images were analysed using ZEN software (Zeiss) or Fiji software 2.14.0/1.54f.

Statistical analysis and data representation
Statistical analyses and graph plotting was performed using Prism software v10 (GraphPad). Data were tested for assumptions of normality and equality of variance to determine the appropriate statistical tests to perform. No data were excluded. Mean values are reported +/- standard deviation (SD). Statistical significance was defined as p < 0.05. All figures were assembled in Adobe Illustrator 2023.

For manuscripts utilizing custom algorithms or software that are central to the research but not yet described in published literature, software must be made available to editors and reviewers. We strongly encourage code deposition in a community repository (e.g. GitHub). See the Nature Portfolio guidelines for submitting code & software for further information.

# Data

Policy information about availability of data

All manuscripts must include a data availability statement. This statement should provide the following information, where applicable:
- Accession codes, unique identifiers, or web links for publicly available datasets
- A description of any restrictions on data availability
- For clinical datasets or third party data, please ensure that the statement adheres to our policy

Genome assembly AmexG_v6.0-DD (https://genome.axolotl-omics.org/index.html) and transcriptome assembly AmexT_v47 (https://www.axolotl-omics.org/assemblies) were used (Schloissnig et al. 2021).

All RNA-sequencing data have been deposited at the Gene Expression Omnibus (GEO), under accessions GSE243137 (dermal cell data) and GSE284768 (all other data).

There are no restrictions on data availability.

# Research involving human participants, their data, or biological material

Policy information about studies with human participants or human data. See also policy information about sex, gender (identity/presentation), and sexual orientation and race, ethnicity and racism.

| | |
|---|---|
| Reporting on sex and gender | Not applicable to this study as no human data were collected. |
| Reporting on race, ethnicity, or other socially relevant groupings | Not applicable to this study as no human data were collected. |
| Population characteristics | Not applicable to this study as no human data were collected. |
| Recruitment | Not applicable to this study as no human data were collected. |
| Ethics oversight | Not applicable to this study as no human data were collected. |

Note that full information on the approval of the study protocol must also be provided in the manuscript.

# Field-specific reporting

Please select the one below that is the best fit for your research. If you are not sure, read the appropriate sections before making your selection.

☒ Life sciences  ☐ Behavioural & social sciences  ☐ Ecological, evolutionary & environmental sciences

For a reference copy of the document with all sections, see nature.com/documents/nr-reporting-summary-flat.pdf

# Life sciences study design

All studies must disclose on these points even when the disclosure is negative.

| | |
|---|---|
| Sample size | No sample size calculation was performed. All sample sizes are indicated in the relevant figure legend, main text or methods. n numbers refer to independent biological replicates. A minimum of 3 biological replicates were used per condition, which we considered a balance between data reliability and animal usage. |
| Data exclusions | No data were excluded. |

| Replication | Each experiment was replicated at least once unless indicated otherwise, with successful replication of conclusions. Conclusions were made based on experiments using independent axolotl cohorts and multiple experimental approaches. |
| --- | --- |
| Randomization | Axolotls from the same cohort were randomly allocated into experimental or control groups and housed under identical conditions. Exceptions are experiments (detailed in the Methods) in which the control was performed on one limb and the experimental perturbation on the other limb of the same axolotl. For ALM experiments, we ensured that there was no bias in the success of ALM formation based on left or right host limb. |
| Blinding | Blinding was not feasible as the work was largely performed by one author and/or phenotypes were much more severe in the experimental cohort than in the control cohort. |

# Reporting for specific materials, systems and methods

We require information from authors about some types of materials, experimental systems and methods used in many studies. Here, indicate whether each material, system or method listed is relevant to your study. If you are not sure if a list item applies to your research, read the appropriate section before selecting a response.

## Materials & experimental systems

| n/a | Involved in the study |
| --- | --- |
| ☐ | ☒ Antibodies |
| ☒ | ☐ Eukaryotic cell lines |
| ☒ | ☐ Palaeontology and archaeology |
| ☐ | ☒ Animals and other organisms |
| ☒ | ☐ Clinical data |
| ☒ | ☐ Dual use research of concern |
| ☒ | ☐ Plants |

## Methods

| n/a | Involved in the study |
| --- | --- |
| ☒ | ☐ ChIP-seq |
| ☐ | ☒ Flow cytometry |
| ☒ | ☐ MRI-based neuroimaging |

## Antibodies

| Antibodies used | Rabbit anti-Prrx1 antibody, generated by Gerber et al. 2018 (DOI: 10.1126/science.aaq0681). Mouse anti-Col1A1 antibody (SP1.D8, DSHB). Alexa 647-conjugated anti-rabbit (Invitrogen A-21244) or anti-mouse (Invitrogen A-21240) secondary antibody. |
| --- | --- |
| Validation | Prrx1 antibody was characterised in axolotl limb tissue by Gerber et al. 2018 (DOI: 10.1126/science.aaq0681). Col1A1 antibody was characterised in axolotl tissue by Gerber et al. 2018 (DOI: 10.1126/science.aaq0681). |

## Animals and other research organisms

Policy information about studies involving animals; ARRIVE guidelines recommended for reporting animal research, and Sex and Gender in Research

| Laboratory animals | Axolotl (Ambystoma mexicanum), age up to ~8 months. |
| --- | --- |
| Wild animals | The study did not involve wild animals. |
| Reporting on sex | Sex data was not collected in this study. Sex is not readily distinguishable based on morphology in the axolotl sizes used in this study. Thus, data collection was randomised with respect to sex. Sex determination in these animals requires a genotyping PCR, and the injury from tissue harvesting could induce an unwanted tissue response, as suggested in DOI: 10.1016/j.ydbio.2017.07.010. |
| Field-collected samples | The study did not involve animals collected from the field. |
| Ethics oversight | All animal experiments were approved by the Magistrate of Vienna (Genetically Modified Organism Office and MA58, City of Vienna, Austria), under licenses GZ:51072/2019/16, GZ: MA58-1432587-2022-12 and GZ: MA58-1516101-2023-21. |

Note that full information on the approval of the study protocol must also be provided in the manuscript.

# Plants

| | |
|---|---|
| Seed stocks | Not applicable. |
| Novel plant genotypes | Not applicable. |
| Authentication | Not applicable. |

# Flow Cytometry

## Plots

Confirm that:

☒ The axis labels state the marker and fluorochrome used (e.g. CD4-FITC).

☒ The axis scales are clearly visible. Include numbers along axes only for bottom left plot of group (a 'group' is an analysis of identical markers).

☒ All plots are contour plots with outliers or pseudocolor plots.

☒ A numerical value for number of cells or percentage (with statistics) is provided.

## Methodology

| | |
|---|---|
| Sample preparation | The Hand2:EGFP intensity of mature arm cells, 7 dpa blastema cells and 14 dpa blastema cells were compared by flow cytometry. Lower arm tissue was harvested from 6 cm Hand2:EGFP axolotls. The entire lower arm was taken for mature measurements. Blastemas were generated by amputating through the middle of the lower arm 7 or 14 days prior to flow cytometry. Harvested tissues were dissociated into single cell suspensions using Liberase TM enzyme (Merck 05401127001) as described in Lin et al. 2021 (DOI: 10.1016/j.devcel.2021.04.016), with the following modifications: dissociation was performed for 55 mins (mature sample) or 45 mins (blastema samples) and the cells were filtered through a 70 um MACS SmartStrainer (Miltenyi Biotec 130-098-462). |
| Instrument | Cells were analysed by FACS (FACSAria III Cell Sorter, BD Biosciences) using a 100 um low pressure nozzle. |
| Software | FLOWJO software (BD Biosciences). |
| Cell population abundance | Representative population abundances are depicted in Supplementary Fig. 1. Population abundance generally ranged from ~5-70% depending on genotype and experiment. Purity was not assessed by re-sorting post-sort cells. |
| Gating strategy | The gating strategy is depicted in Supplementary Fig. 1. Cells were identified based on FSC-A and SSC-A gates, and doublet exclusion performed using SSC-H and SSC-W gates. Cell purity was generally > 80% and the proportion of single cells post-dissociation > 90%. The population of interest was identified by comparing against a parallel treated negative control sample lacking the relevant fluorescent protein, as indicated in the manuscript. This allowed determination of positive and negative cells. |

☒ Tick this box to confirm that a figure exemplifying the gating strategy is provided in the Supplementary Information.

