## [Peer Review file · Nature]

Molecular basis of positional memory in limb regeneration

Corresponding Author: Professor Elly Tanaka

This file contains all reviewer reports in order by version, followed by all author rebuttals in order by version. Parts of this Peer Review File have been redacted as indicated to maintain the confidentiality of unpublished data.

Version 0:

Reviewer comments:

Referee #1

(Remarks to the Author)

This paper addresses a component of the broad question within the regenerative biology field of how developmental and regenerative process may overlap or may differ. It also addresses how cells in a mature tissue setting may be recruited to contribute toward the patterning of a regenerate limb and the degree of plasticity of these patterning behaviors. The model used to explore this question is the regenerating axolotl limb, and the tools combine transgenic reporter animals, imaging, transplantation, RNA localization techniques, and some computational analyses.

It is an interesting work whose overall findings will be important to the limb development and regeneration fields. The crux of the work is that while Sonic hedgehog (Shh) expression in the posterior region of the developing limb governs patterning of the anterior-posterior axis, during regeneration its regulation differs from how it is regulated in development. In regeneration, Shh is under the control of Hand2 transcription factor. This is important because how blastema compute positional inputs to organize the various axes for the new limb—which is obviously important for the new limb's form and its ultimate function—is still poorly understood. More published have tackled proximodistal specification, and even there, the whole story is still murky. Anterior-Posterior patterning in salamander limb regeneration has been previously explored to some extent in the axolotl model (see below—mainly work of Stephane Roy), with the main findings that Sonic hedgehog is doing something analogous in axolotl blastema as it does in chick and mouse limb development (and in other species with respect to development). In exploring this role further, these authors leverage the power of genetic reporter animals (which are very labor- and time-intensive to produce for axolotl) to ask which embryonic cells that once expressed these key patterning factors are progenitors for the various cells found in fully-developed limbs, via lineage mapping. Further, the standard reporters (promoters just driving fluorophore expression) revealed that persistent expression of Shh and Hand2 in mature limbs with a posterior expression domain. Through a combination of transplantation, extirpation, and accessory limb model (ALM) experiments, the authors argue that Hand2 is more of the upstream master regulator/organizer of anterior-posterior information input into the regenerating limb, while Shh-expressing cells in the blastema can be sourced from either anterior or posterior domains and they are under control of Hand2 in their competency. They specifically claim that “Hand2 expression enforces positional memory.” It is refreshing to see a paper in 2023 that combines so many physical experiments (with a few computational analyses) in an age when many papers are centered more on analyzing large amounts of computational data and trying to make claims from that with little experimentation.

Whether these differences will be of broad interest outside the limb development and regeneration field is a question to consider. There is an obvious argument to be made that should human limb regeneration ever be a therapeutic reality, these issues will be important for patterning those limbs, and, hence, understanding the biology in a natural limb regeneration model is intrinsically important.

This review focuses more on the technical aspects supporting the authors' claims. The following issues should be addressed by the authors, and some of them require some additional experimentation.

The authors state, “The molecular properties that distinguish anterior and posterior cells prior to injury, which enable them to initiate different signaling centers after amputation, are not known.” Can they further refine or defend this statement? An alternative explanation is that there is no preordained code or at least none that differs between A and P in a direct reflection of future use as signaling centers.

The authors state, “Importantly, *Fgf8* and *Shh* are not expressed in the uninjured salamander limb meaning that . . . “ Can this actually be defended from the RNAseq data (that are presented here and that are available publicly from other studies)?

What is the control for leakiness of the Tamoxifen-Cre experiments? This is important because the data and interpretation they present hinges on these reporters faithfully recapitulating the endogenous expression domains, no more and no less. They did have trouble with one line not efficiently recombining (see below), necessitating three separate tamoxifen injections. This means we should remember that they might still not hit all of the cells capable of turning that promoter on at that time of induction. What they don't address is the flip side—can they convincingly show that there is no ectopic expression of the Cre reporter in the absence of Tamoxifen? We need to see low-mag photos of whole limbs, probably in sections on a slide, to address this concern.

The authors say, “Having confirmed that TFP labelled *Shh* cells during limb development and regeneration . . . “ Are these F1 animals where every cell capable of expressing the reporter has the reporter? There should be more rigorous analysis to demonstrate every TFP cell expresses *Shh* (by co-labelling with the TFP fluorophore and with HCR-FISH).

To rule out a possible requirement for embryonically-derived *Shh* cells, the authors surgically removed these cells and asked what happened following amputation. They note that they only successfully removed 89% of the cells, yet the claims they make are stated as if 100% were removed. This is important because residual cells may provide enough function to execute the role of the larger group. Wording needs to be fixed to address this concern. Ideally, a genetic ablation method for killing relevant cells without surgery would also be employed here to state these kinds of conclusions. This is probably outside the scope of what can be required in a revision because nobody has published a reliable, effective cell ablation method for axolotls (yet).

Relatedly, are there other possible conclusions from the surgical removal of embryonic *Shh* cells, other than that some “wider domain” of posterior cells in mature limb is competent to express *Shh*?

Digit reduction following amputation is only interesting with respect to this subject (Figure 3h-l, extended data 6i-j) if the authors show controls have no digit reduction. Controls are essential for making the conclusions they make in the paper. Yet, the number of controls is very meager, and it is unclear if some of the experimental groups having deviations from the normal number of digits is significant given the number of samples in the experimental groups as well. For example, is N=6 for the ALM model enough to conclude that if accessory limbs fail to form in the experimental group, this failure is attributable to genotype of graft rather than experimental noise? What is the standard ALM success rate in the authors' hands and in the hands of others (published accounts) with and without epidermal tissue grafted? These authors appear to only have a 50% success rate with wild-type grafts, as they point out, and that is only N=3/6.

Lines 285-301: Is this the right experiment? Why ectopically express *Hand2* everywhere rather than only in the anterior domain? The interpretation of the results is hairy given how the experiment was performed. Please discuss.

Some claims are difficult to evaluate because incomplete info is provided; particularly lacking are low-magnification views that would enable reader to assess how specific some of the markers or genes of interest are for the areas of tissue in question. For example, how specific is the probe shown in figure 5h? We can't really evaluate that question because all they present is a very high-magnification view. At the least, some of these need to be shown in full limbs, probably in supplemental material so they can be displayed with high resolution and large, and those images need to be at a reasonable enough exposure time to believe that if positive cells were outside the reported expression domain, they would be detected.

Another spot in question is the high-magnification view of the blastema in figure 2k. It is important to also present the low-magnification view featuring the full limb with the overlay of the two signals?

Claims that a limb has been “completely posteriorized” are weakly supported. The authors have the kinds of data in hand that would guide a reasonable amount of experimentation to more responsibly make this statement—they have the expression data from their sequencing study. They need to do more to characterize these markers to make this claim.

Are the ectopic digits produced really posterior digits in figures 4b and 4c? How do we know? This has been a persistent issue/debate in limb development field over the years, and there are ways that these authors can shore up their claims to support, for example, at a molecular level, that these digits actually are posterior digits.

Is there precedent for CRISPR mutagenesis and genotyping using alterations to codon (making base pair diffs that do not change amino acid sequence)? If so, please cite.

Important question—one of the biggest technological issues with the paper: The authors do not use established, homozygous lines for their claims about loss-of-function. They use F0 animals. Axolotl generation time is about one year, so clearly doing the full-on genetics will take significantly more time. However, it is critical that they make clear that they are using animals that are not clean lines and the caveats that implies. Because they only use this one mutagenesis approach to support their claims about *Hand2*, and no parallel methods, there are important shortcomings that should be addressed. These are crispants. The cells will be mixture of different alleles, and some will be differentially edited at the two different alleles. Therefore, typical validations such as characterizing RNA expression, or even the sequence of the RNA expressed, cannot be performed. There will be too many different alleles/RNA species with potentially different effects. If they had a stable line, they could confirm the loss of the mRNA expression, or at least that the resultant sequence was untranslatable.

This issue is critical for this work, but these particular authors addressing this exact issue is something that is highly critical for the field. The reason is because the Tanaka lab has pioneered these techniques in axolotl, and the work it produces will be taken as a sort of “industry standard” down the line when other papers are considered and peer reviewed. It sets a precedent. Most people working on axolotl would benefit, publication-wise, if F0 data is published as loss-of-function proof in axolotl in Nature. Use of F0 will be more commonly accepted in peer-reviewed publications, which can be good to accelerate sharing results, but it always needs to be coupled with clear explanations of the potential downfalls of this strategy/short-cut. As a counterpoint to what is presented in this work, for their 2022 paper published in Development, Takeuchi et al. went further than crispant-based data in a newt study. They intercrossed F1 animals with their Hoxa13 CRISPR-induced alleles to generate F2 animals, a real allelic series in a stable genetic setting. This appears to have been required to make the loss-of-function claims with respect to patterning along the proximodistal axis. So it is possible. It is also possible to make cohorts of F1 animals that at least all have the same genotype by mating two F0 and selecting those offspring with equivalent alleles for study.

The alleles themselves should be further analyzed with experiments:

The targeted removal of the start codon may be an odd choice. The authors mention they did this and it is schematized in the extended data. However, they never explain why they designed the targeting strategy this way, nor do they provide a citation of anyone else doing this before. There are good reasons why this could be problematic. Hand2 is a 2-exon gene with a single intron. There is a second methionine at amino acid position 146, still within the first exon, but comfortably far away from their double gRNA strategy. Thus, it is quite possible this locus could be translated into an abbreviated version of Hand2 in their mutants (aka crispants). If this other methionine is being used in the animals to make a version of Hand2 protein, it would be an N-terminal truncated version, and all bets are off about what kind of function that truncated protein might have. It could be dominant negative or geomorphic, or it could have wild-type function (or it could be a null, which is probably what they hope, but they have not evaluated nor even discussed these important issues). For context, please see Tuladhar et al., 2019, Nature Communications, “CRISPR-Cas9-based mutagenesis frequently provokes on-target mRNA misregulation” (PMID 31492834). They addressed these issues using a variety of genes with well-established biological roles, and they found that ~50% of the induced alleles (across intended to be loss-of-function null alleles) indeed produce some type of protein and that these can have a variety of functions with respect to what the wild-type protein does and what the true null phenotype would be.

Why not use a single guide right after the M146-encoding codon, but before the intron? This would likely be sufficient. Alternatively, they could use two guides to cut the whole gene out or the whole first exon, or they could take out the transcriptional start site. The results of this study, as presented, may turn out to match the results of these more rigorous methods for creating null alleles, but we don't know. What is presented cannot be claimed to be a true loss-of-function phenotype without more defense. It currently reads as, “We found the phenotype we predicted, so it must have worked as imagined.”

Another approach could be to use the start-codon guide they already have to take out the fluorescence in their hand2 knock-ins. If the protein is silenced, the GFP should also be silenced; if the protein is truncated, then the GFP should persist. However, it's still difficult to interpret this one because there are many possible outcomes. The point stands that if they are going to be permitted to make the claims they want from F0 data, they should at minimum have more validation, and they also need to note the shortcomings of the approach in the text.

Adding a very clear table about the different lines produced, whether they are F0 mosaic transgenics (random insertion), F0 mosaic knock-ins, F0 mosaic crispants, F1 knock-ins, established reporters, etc. would be very helpful. Extended data figure 1 is still confusing.

It may be possible for the authors to use a complementary method to prove their points without having to derive new alleles or breed these to true homozygosity. It is likely that other reviewers outside the axolotl field will be more insistent on true F2 homozygotes versus wt sibs for these studies (for example, zebrafish).

Other issues to address/consider in contextualizing this work:

There are no references to the two key papers by Stéphane Roy, Susan Bryant, David Gardiner. These papers are foundational for this research question and must be presented. In one published in 1999 (PMID 10656763), it was demonstrated (through ectopic expression using Vaccinia viruses) that ectopic Shh expression in axolotl blastemas causes mirror-image duplications of digits along the A/P axis, which resembles what is seen in mouse and chick. In the other (PMID 12115913, published in 2002), it was demonstrated that inhibiting Shh activity (pharmacologically, using cyclopamine) reduces axolotl digits in the manner predicted based on mouse and chick. Both of these papers together underscore that cell behaviors downstream of Shh activity are similar in the contexts of development and regeneration. Overall, they established this to be true using the axolotl model. This work presented in this paper clearly stands on that foundational work, yet it is never mentioned in the text.

Lines 166-167 reference a study in zebrafish that came to similar conclusions as this paper with respect to Shh. Additional context for why the axolotl study is so novel is necessary, which should be beyond the idea that now the appendage under study is a tetrapod limb.

(Remarks to the Author)

Otsuki and colleagues investigated how anterior-posterior CT cells maintain their positional features in the axolotl limb. Their research began by showing that during limb regeneration, posterior Shh expression doesn't necessarily originate from cells that once expressed Shh. The authors revealed the dynamics and lineage of posterior CT-associated Hand2+ cell expression. Through a series of thorough gain- and loss-of-function experiments, a solid link between Hand2 and Shh was established. Using a double transgenic model, the work revealed that anterior to posterior transplants could activate Hand2 upon amputation. Conversely, posterior-to-anterior transplants were unable to stimulate anterior-associated Alx4 expression upon amputation. Instead, these cells maintained Hand2 expression, activated Shh, and resulted in abnormal patterning. Further experiments showed that some cells that once expressed Alx4 in anterior-posterior grafts were able to express some Shh. Shh was shown to mediate the activation of Hand2 in anterior to posterior transplants. The team also used a memory assay, which showed that even brief exposure to Shh can prime anterior cells to express Hand2. Based on these, the authors proposed a model to explain how AP axis info is retained during regeneration, emphasizing the dominant role of Hand2 expression and posterior cells.

The study is methodically planned and excellently implemented by combining sophisticated axolotl transgenic lines with traditional grafting studies, effectively utilizing the axolotl's distinctive features. The high standard of presentation of the research deserves special mention. This study is a commendable extension of the Tanaka Lab's prior work, Nacu et al (2016), which initially identified the Shh-Fgf8 loop as the mechanism controlling anterior-posterior positional information. This manuscript is an important work and would be of interest to developmental and regeneration biologists. I am unsure if its reach can hit the broader-audience, as the topic is rather niche and their earlier work (Nacu et al 2016) has conceptually proposed the molecular basis for the anterior-posterior features, and this work elegantly adds more components. The most innovative experiments are shown in Figures 5 and 6. The established link between Shh and Hand2 (shown in Figures 2, 3, and 4) mirrors the known link in other species, as the authors also mentioned. The post-embryonic posterior expression of Hand2 is another novel finding. The experiments and results are sufficiently detailed in the figures/methods, and the figures are designed to make even the complicated experiments easy to understand. Although it is a very high-quality work, my biggest frustration with this manuscript is that, the text and interpretations contain many terminologies that over-sophisticate/complicate findings. Claims of "memory", "reprogramming", "positional identity" require substantial revisions via editing the text and performing new experiments. The proposed model may not be able to explain some performed experiments or findings in the literature. Also, the results are interesting, but their broader implications are not substantiated, giving the impression of overstating the importance of the findings. Without overly sophisticated terminologies and resulting complications, this is an interesting and solid work.

Comments

1. One of the central issues of the manuscript revolves around the ambiguous description of anterior/posterior "identity," demarcated by Alx4 and Hand2/Shh expression. If a genuine identity component exists, one would anticipate a pronounced transcriptional difference between these populations, especially considering that Alx4 and Hand2 are transcription factors. It would be necessary to explore available scRNA-seq datasets to confirm if Alx4 and Hand2 cells manifest significant differences and subsequently assay additional marker gene expressions to bolster the "identity" element beyond a singular gene observation. However, I remain dubious about the presence of a major transcriptional shift. Previously published limb scRNA-seq datasets, including those from Tanaka Lab, have failed to distinguish between anterior and posterior CT. This suggests that positional information might not represent distinct identities. Instead, CT cells may possess certain plasticity, with Hand2 inducing stable dominant alterations within this framework.
2. The authors employ the term "reprogramming" based on a transition from Alx4 to Hand2 expression. However, this alteration might not necessarily denote reprogramming, as the term "reprogramming" is used to describe perturbations that induce stable cell identity changes. Recent research further shows reprogramming via single-cell transcriptomics-based clustering analysis for validation. The authors could undertake scRNA-Seq on sorted Alx4+ and Hand2+ cells after their grafting experiments depicted in Fig 5 with appropriate controls to validate their reprogramming assertion. Yet, I remain uncertain if the authors would genuinely identify a cell fate alteration qualifying as reprogramming. Based on extensive publicly accessible scRNA-seq data, even cells expressing Shh (potentially inclusive of hand2+) don't segregate into distinct clusters indicative of a new, stable cell type. Similarly, anterior and posterior CT cells lack discernible transcriptome distinctions in such datasets. The manuscript's bulk RNA-seq results further underscore the idea that CT cells are malleable, reacting to external cues, with Shh acting as a potent inducer of stable Hand2. Given these factors, it remains ambiguous whether the observed phenotypes in this study genuinely epitomize positional identity, memory, or reprogramming.
3. In connection with Fig 5, when grafting from anterior to posterior limbs, it's possible that anterior genes, aside from alx4, remain active in the grafts. The authors should perform staining for anterior and posterior markers in these experiments. Ideally, conducting scRNA-seq on these grafted samples would offer definitive data regarding the extent to which anterior or posterior genes have altered, providing support about positional characteristics as distinct identities.
4. In several sections of the text and figures, there's ambiguity regarding how the authors differentiate between "memory" and "positional identity". For instance, in the titles of Fig 5 and 6, substituting "memory" with "positional identity" wouldn't seem to alter the meaning. Could the authors elucidate their criteria for distinguishing between memory and positional identity, aside from the experiment in Fig 6d and e (which appears to be the sole definitive assay for memory)?

5. In Extended Data 4, while RNAseq results consistently highlight Hand2 and a few others as being enriched in the posterior, there seems to be inconsistency in the replicates for genes associated with signaling pathways. It might be prudent to exclude conclusions related to these pathways, given that the findings aren't stemming from reproducible data. Moreover, such information detracts from the central narrative and doesn't offer significant insight.
6. The conclusion made by the authors on lines 230/510, stating that "hand2 encodes posterior identity," lacks adequate support. If Hand2 indeed encoded posterior identity, its overexpression would result in double-posterior limbs, evident through mirrored digit morphologies (PMID: 15192229, 2226209). However, this phenotype isn't observed in the present work. The conclusion on lines 230/510 needs revisiting. Interestingly, other portions of the text seem to address this topic with greater precision.
7. Figure 6d-e shows that brief activation of Shh can prime anterior cells to express Hand2, so one would expect that cells that have received Shh once would be able to readily re-activate Hand2/Shh. However, Figure 1 shows that the Shh lineage does not contribute to the new regenerative Shh cells, which contradicts the proposed priming data and suggests that there may be no so-called "memory". Shh treatment can also prime anterior cells, but, surprisingly, Shh progeny do not necessarily activate Shh. Based on their model, this finding is a bit contradictory. Do the authors have an explanation for this in terms of their model?
8. Fig 6d-e shows that there is a priming induced by Shh activation in anterior tissues. Based on this, one would expect that repeated amputations would result in axolotls with limb growth defects as they would activate Hand2/Shh more broadly. However, the axolotl is known to regrow its limb perfectly even after repeat amputations, although one study suggests that they may even lose their regenerative capacity (PMID: 29302364). So how do the authors resolve this with their proposed model?
9. ZRS is used as a Shh enhancer to label the Shh lineage, which marks distal posterior cells. However, there is another study (PMID: 31538936) that shows an additional proximal anterior Shh expression in developing axolotl limbs during limb bud stages. Some of the images in this manuscript (Extended data 2) also show some fluorescent signal for the proximal anterior region (although the signal in these data could be autofluorescence?). However, their lineage tracing data does not show this anterior proximal signal. Could you clarify this point and check if you see the proximal anterior signal by other means such as HCR? Because if they are seeing Shh in the proximal anterior region, they may need to rethink their interpretations because these populations would be primed for Shh? This situation is confusing within this manuscript and inconsistent with another published paper.
10. The broader implications for the tissue engineering component are very vague and a bit stretched. The mechanism described here reads very specific to the axolotl, even for limbs, as the axolotl has its unique anterior-posterior mesenchymal *fgf8-shh* loop. Is there a concrete example and experimental evidence that the authors can provide for their proposed tissue engineering idea? Softening or modifying such broad implications would be a better presentation of the implications.
11. Fig 2 legend indicates "posterior memory cells". It is not clear how this figure has data related to so-called "memory". Deleting memory and just mentioning "posterior cells increase Hand2 expression..." represents the data accurately.
12. The authors used Alx4 to label anterior cells based on an earlier study. The same study and other work in the field found Gli3R as another anterior marker (at least at limb bud stages), which is also more relevant to the Shh pathway. Is there a reason why the authors did not follow up on Gli3R? Do they see its expression as anterior specific in the axolotl and still expressed at post-embryonic stages as Hand2? If the transplanted anterior cells contain Gli3R, one would expect them to turn on Hand2 and Shh during the regeneration assay, as shown in Fig. 5. This would also support the idea that posterior to anterior transplants induce patterning defects and extra limbs because the cells would have receptors to easily receive Shh.
13. Do the authors have an explanation for why Hand2 crispants regenerate with fewer digits? If these crispants have mosaic Hand2 expression or clones with higher Hand2 expression, these Hand2+ cells would be selected, eventually leading to more digit formation during regeneration. Likewise, if there were a memory or Shh-based priming, one would expect to see more digits after amputations because there would be more Shh? I am not sure that the current model explains these data.
14. Ext data fig 5 shows that hand2 is expressed in dermal and interstitial fibroblasts, but the images in the main figures look like hand2 is expressed in perichondral fibroblasts, and joints. Could the authors provide additional confocal sections to map where Hand2 is expressed precisely?
15. It is not shown at the cellular level where Alx4 is expressed. This is important information because the grafting experiments between the AP axis in fig5 assume that Alx4 and Hand2 are similar cell types with different positions, but no data on this is shown.
16. The results suggest that hand2 cells are responsible for the effect of the posterior skin graft on the anterior regions, possibly due to their priming for shh expression. Do the authors still see the ectopic patterning and shh expression after amputations when they sort and inject hand2- cells (from the hand2+ posterior areas) (perhaps using *prx1:mcherry* background can help with this experiment)? If they don't, then it would further solidify that the positional information refers to this Hand2 gene expression, but not to any other gene expression associated with posterior cells.
17. Hand2 was reported to be posterior specific by the same group, Nacu et al. (ext data fig9. Fig 2 tries to provide the same

information as novel. Please provide accurate information on novelty.

18. In Nacu et al - the authors showed that cyclopamine treatment blocks regeneration when given on day 6. Here they use a different inhibitor for the same purpose (which is also suggested to block regeneration in another study PMID: 35433673), but they see some form of growth/regeneration happening in Fig. 6c. Could the authors comment on this discrepancy?

19. It would be great to indicate autofluorescence. ext Fig. 3 f, Shh could be perceived as it is expressed in the middle bones. A similar situation can be seen in other figures.

20. Skin grafting with AP juxtaposition can induce additional limbs (ALM model). Meanwhile, in this study, anterior or posterior cell "injection" does not. The authors' figures show that the ALM model requires additional wounding and innervation. It might be useful to point out this difference in the text, as it might be perceived as somewhat contradictory to their earlier study.

21. The experiment in Figure 5g is labeled "regenerated limb," but the limb looks abnormal and is noted as ectopically patterned in the text. It might be more accurate to mention that this does not fully suggest a regeneration scenario in the figure.

Referee #3

(Remarks to the Author)

The manuscript by Otsuki et al reports on the molecular nature of cellular memory in axolotl limbs and limb blastemas during regeneration. The authors develop an extensive set of new cell tracing and functional genetic models in the axolotl and combine this with drug treatments and tissue transplantation/amputation to demonstrate that persistent Hand2 expression is responsible for posterior identity during limb regeneration. They also demonstrate that the signalling environment can switch anterior to posterior cell identity but not vice-versa. Furthermore, they uncover a positive feedback loop between Shh and Hand2 required for regeneration and able to reset anterior cell identity to a posterior identity/memory. This work is a major breakthrough in understanding positional memory and its role in regeneration, with implications for tissue engineering. The manuscript is rather mature for publication and I only have minor comments/suggestions:

-In Figure 2e, Hand2 reporting in the mature limb appears evident only in skeletal elements, however in the sections shown in the extended data Figure 5a, the reporter appears exclusively in connective tissue. The authors may consider explaining this apparent discrepancy

- Do the authors have a notion of the proportion of Hand2-deleted cells required to produce the diversity of phenotypes shown in Figure 3 and extended data Figure 6? Given that in some of these mosaic animals Shh is not detected, does this suggest non-autonomous cell effects of Hand2 deletion?

- The authors may think on revising this statement: "Taken together, these results show that Hand2 is necessary for posterior identity and suggest that cells that fail to acquire a posterior identity do not default to an anterior identity". Another view is that cells do not fail to acquire posterior identity (which could only be determined by a wider expression analysis) but rather that they lost their ability to support regeneration through losing their ability to activate Shh.

Referee #4

(Remarks to the Author)

The study by Otsuki et al. identifies the molecular mechanism that provides posterior cells in adult axolotl limbs with a memory of their posterior origin by their low level expression of the Hand2 transcription factor. This primes for regeneration after limb amputation as the observed Hand2 up-regulation reinitiates Shh expression in the blastema and establishes a Hand2-Shh positive feedback loop. After limb regeneration is complete, this posterior memory is re-established by sustained low level Hand2 expression in posterior cells. The authors show that this memory is stable and that Hand2-expressing cells cannot be "anteriorized". Instead they keep their ability to trigger the Hand2-Shh feedback loop repeatedly following re-amputation. Furthermore, anterior cells can be reprogrammed as Hand2 expressing memory cells by exposure to Shh.

This study provides fundamental novel insights into how a long-term molecular memory of posterior identity is maintained and able to trigger complete regeneration by initiating a positive feedback loop between Hand2 and Shh. These results are of very high general importance as they provide insights into processes that to date have remained elusive. In addition, the approaches are state-of-the-art and showcase novel genetic and molecular tools that make axolotl the prime model to study the molecular basis of cellular memory for limb regeneration.

Comments

1. Hand2: based on their transcriptome analysis (Fig. 2a-c), the authors select the transcription factor Hand2 as the prime transcription factor candidate for molecular memory. This is valid but one could have equally well chosen Hoxd13/a13 as

these 5' Hox transcription factors are also required to activate Shh and function as part of a positive feedback with Shh during limb development in vertebrates. However, this reviewer agrees that Hand2 is a good choice and is not suggesting analysis of 5' Hox genes, but would like the authors to discuss the potential involvement/requirement of 5' Hox and other posterior transcription factors. It is almost a given that for long-term and robust molecular memory, more than a two-factor feedback system is needed.

2. The authors should be cautious about quantitative statements when using EGFP reporter construct as this protein is very stable. For quantitative analysis one would have to use one of the available labile GFP versions.

3. Page 7: the authors state that in other vertebrates it is the distal epidermis-posterior mesenchyme that controls limb bud outgrowth, which is too simplistic. Many studies have shown that for properly controlled limb bud development, interactions between anterior and posterior transcription factors and signaling pathways are crucial.

4. In this reviewer's view the authors should discuss the striking parallels that emerge thanks to their study between normal limb development (chicken/mouse) and axolotl limb regeneration. This is really stunning – just one example: the posterior dominance during regeneration seems equivalent to posterior prevalence during normal limb development.

5. As this manuscript is submitted to a very general journal: can the authors briefly discuss/ speculate if such memory cells exist or not in higher vertebrates like mammals and why this trait was lost during evolution of higher vertebrates.

Technical issues

This is a complex and data-rich study of very high quality and excellent design, which fully supports the conclusions by the authors. There is few, but important additional comments:

6. Shh HCRs: there is unfortunately significant background signal -probably from the vascular system. This renders assessment of data difficult.

Fig. 2k: in addition to the overlap, please include the individual panels for the Hand2 lineage and Shh HCR using a greyscale.

Fig. 5e/h: Shh signal and background are difficult to discern.

Fig. S9g: due to the background the normal and ectopic Shh signaling center are hardly discernible and the scheme is not really helpful either. Can the authors please consider to include high mags and or conventional BMPurple whole mount in situ to detect Shh.

7. Fig. 4l: The "ZRS>TFP" lineage is indicated but not visible. Including a high magnification of the relevant region of the ectopic limb in the left panel seems necessary.

Minor but still important issues

8. Line 232: "...Hand2 is necessary to activate Shh expression..." (rather than "to express")

9. Over all the readability of the manuscript could be improved somewhat by removing some non-important details and redundant text (not easy to pinpoint what to cut exactly).

10. Literature reference 27 (Charite et al. 2000) is problematic as citation for proof that Hand2 is required to activate Shh. The main reason is that Hand2-deficient embryos start to die around the stage (due to the severe heart defects) when Shh is normally activated in early limb buds, which poses a cause and consequence problem. This was circumvented by limb bud specific conditional inactivation of Hand2, which indeed disrupts Shh activation but not early limb bud development (Galli et al. 2010, reference 32).

Version 1:

Reviewer comments:

Referee #1

(Remarks to the Author)

REVISION COMMENTS:

General:

The authors have done a very thorough job addressing each of the concerns we raised, many of which were vital for ensuring arguments were well supported, the audience was informed of the considerations and nuances influencing experimental decisions and conclusions, and the work was adequately contextualized with existing literature. Certainly, we can all agree that interesting questions remain for future work that will be inspired by this study, including some questions I could now ask that could be riffs off valid questions the other reviewers raised. However, assuming the other reviewers are satisfied with the authors' efforts and responses to their personal critiques, I do not favor more revision. A second revision would perpetuate the practice of journals asking for even more experimentation and analysis beyond what was reasonably and implicitly agreed upon by all the parties at this table. Scientific progress using the axolotl model has been thwarted by these kinds of practices, resulting in major papers being held up for years.

Specific (minor, but would be an easy addition that I do not need to re-review following inclusion in text):

Their reasoning about the genetics makes sense, and it is important to include these details in the manuscript, which the authors have now done. One final point that I think should be noted in the revised text before it is published is that around lines 246-250, after rationalizing why use the crispants in the cross due to probable homozygous loss-of-function lethality at the Hand2 locus: please add one sentence mentioning that conditional alleles of endogenous genes (for example, floxed alleles of Hand2) is a method not yet available in axolotl. Without mentioning this, I feel readers will next wonder why not knock Hand2 out specifically in limb tissues, in animals that had already fully developed. That strategy is commonplace in

mouse, for example, and I think it would benefit both the paper in hand and our field for this information to be made overt. Beyond it not being an established technology in axolotl (likely due to low efficiency of targeted knock-ins in general?), such an approach would also require something like three years to execute rigorously.

Referee #2

(Remarks to the Author)

The authors have adequately addressed most of my concerns. I remain skeptical about their claims that the observed phenotype represents cellular reprogramming. In the current literature, such claims are demonstrated and well-supported by single-cell sequencing methods or pure isolated cell grafting and tracing experiments. Bulk RNA-seq and heterogeneous population grafts, as performed here, are less rigorous (their methods and conclusions would have been acceptable a decade ago, but this concept and the methods used to investigate it have evolved). While the technical limitations they discuss are somewhat valid (although more sensitive single-cell methods, like SMART-seq variations, exist, albeit impractical in this context), I still recommend they reconsider their use of the term "reprogramming" to align with current standards and to avoid unnecessary critique of an otherwise high-quality and interesting study.

I have one analysis suggestion to partially address my concern. The authors indicated that they identified 146 genes linked to anterior tissue identity and 160 genes to posterior tissue identity in their bulk RNA-seq data. In their new bulk RNA-seq analysis following grafting between anterior and posterior tissues (Extended Data Fig. 9), they should report how many of the anterior and posterior genes from the 146 and 160 suggested, beyond the six selected (Extended Data Fig. 9f), show statistically significant changes following grafting. While they highlight statistically significant genes in Supplementary Tables 8-11, these documents do not provide this specific information. Reporting in-text how many of these anterior and posterior identity genes (e.g., X/146) show significant shifts following grafting would strengthen their claims, even if it does not fully meet the standards of current literature (and they would still not account for other potential explanations such as tissue heterogeneities influencing results..).

This is an excellently implemented and interesting study. I still find the manuscript verbose, and their responses to questions about their model are somewhat incomplete. However, these issues should not prevent the manuscript's publication. I am supportive of its acceptance.

Referee #3

(Remarks to the Author)

The authors have answered all my concerns. The manuscript reveals a fundamental aspect of regenerative biology through extensive and rigorous experiments. It also opens several future questions related to the topic of positional memory: how is hand2 expression maintained? is it functional in mature cells or just sitting there as positional memory? is hand2 alone or just a part of a TF network with various alternative entry points?

Referee #4

(Remarks to the Author)

Otsuki and colleagues have conducted a large number of additional analyses and revised the text extensively in response to the comments of all four reviewers. This has significantly improved the study and added important support to the data and conclusions.

There is one issue that this reviewer already pointed out in the initial review, namely that a 2-component feedback system involving HAND2 and SHH as proposed by the authors is unlikely sufficient for establishing and maintaining a robust but yet reprogrammable memory mechanism. Rather, this study appears to identify two of the (essential) components of a likely larger and more complex molecular mechanism that underlies positional memory and its re-programmability, which as such is a very important discovery. The comment is not focusing on a detail, as data-based simulations of feedback systems have shown that systems consisting of several interacting feedback loops generate both robustness and adaptability - as seems the case for positional memory. Furthermore, HAND2 regulates the spatio-temporal kinetics of Shh expression in concert with several other transcriptional regulators. Genetic analysis in the mouse shows that the essential transcription factors include Hand2, 5'Hox TFs, Pbx, Gli3 and others that function in Shh activation, up-regulation and restriction within the posterior limb mesenchyme. The authors should nuance their discussion and put their findings better into this broader context in light of the general and fundamental importance of studies published by Nature.

Response to reviewer comments

(comments received on 2023-11-17)

Your manuscript, "Molecular basis for positional memory and its reprogrammability in limb regeneration", has now been seen by 4 referees, whose comments are attached below. While they find your work of potential interest, as do we, they have raised important concerns that in our view need to be addressed before we can consider publication in Nature.

We thank the reviewers for their constructive suggestions. We have addressed all of these (in blue text) through additional experiments and discussions and hope that our manuscript is now satisfactory for publication.

Referee #1 (Remarks to the Author):

This paper addresses a component of the broad question within the regenerative biology field of how developmental and regenerative process may overlap or may differ. It also addresses how cells in a mature tissue setting may be recruited to contribute toward the patterning of a regenerate limb and the degree of plasticity of these patterning behaviors. The model used to explore this question is the regenerating axolotl limb, and the tools combine transgenic reporter animals, imaging, transplantation, RNA localization techniques, and some computational analyses.

It is an interesting work whose overall findings will be important to the limb development and regeneration fields. The crux of the work is that while Sonic hedgehog (Shh) expression in the posterior region of the developing limb governs patterning of the anterior-posterior axis, during regeneration its regulation differs from how it is regulated in development. In regeneration, Shh is under the control of Hand2 transcription factor. This is important because how blastema compute positional inputs to organize the various axes for the new limb—which is obviously important for the new limb’s form and its ultimate function—is still poorly understood. More published have tackled proximodistal specification, and even there, the whole story is still murky. Anterior-Posterior patterning in salamander limb regeneration has been previously explored to some extent in the axolotl model (see below—mainly work of Stephane Roy), with the main findings that Sonic hedgehog is doing something analogous in axolotl blastema as it does in chick and mouse limb development (and in other species with respect to development). In exploring this role further, these authors leverage the power of genetic reporter animals (which are very labor- and time-intensive to produce for axolotl) to ask which embryonic cells that once expressed these key patterning factors are progenitors for the various cells found in fully-developed limbs, via lineage mapping. Further, the standard reporters (promoters just driving fluorophore expression) revealed that persistent expression of Shh and Hand2 in mature limbs with a posterior expression domain. Through a combination of transplantation, extirpation, and accessory limb model (ALM) experiments, the authors argue that Hand2 is more of the upstream master regulator/organizer of anterior-posterior information input into the regenerating limb, while Shh-expressing cells in the blastema can be sourced from either anterior or posterior domains and they are under control of Hand2 in their competency. They specifically claim that “Hand2 expression enforces positional memory.” It is refreshing to see a paper in 2023 that combines so many physical experiments (with a few computational analyses) in an age when many papers are centered more on analyzing large amounts of computational data and trying to make claims from that with little experimentation.

Whether these differences will be of broad interest outside the limb development and

regeneration field is a question to consider. There is an obvious argument to be made that should human limb regeneration ever be a therapeutic reality, these issues will be important for patterning those limbs, and, hence, understanding the biology in a natural limb regeneration model is intrinsically important.

This review focuses more on the technical aspects supporting the authors' claims. The following issues should be addressed by the authors, and some of them require some additional experimentation.

The authors state, "The molecular properties that distinguish anterior and posterior cells prior to injury, which enable them to initiate different signaling centers after amputation, are not known." Can they further refine or defend this statement? An alternative explanation is that there is no preordained code or at least none that differs between A and P in a direct reflection of future use as signaling centers.

- Yes, the existence of a preordained code is supported by molecular profiling and morphogenetic assays. Anterior skin transplanted to the posterior side induces normal-looking or mildly hypomorphic regenerates, while posterior skin transplanted to the anterior side results in polydactylous regenerates (PMID: 7003051, PMID: 6754846). Thus, there is a position-based difference that is stable to transplantation. Moreover, posterior cells transplanted anteriorly upregulate *Shh* as they would normally do, despite their ectopic location (PMID: 31386776). Thus, the positional code directly reflects future use during regeneration as signalling centres. These properties are the basis for the ALM assay (PMID: 15136146).
- We have added the following text to the manuscript to clarify this:
 - *"We expected to detect stable molecular differences based on previous experiments that showed an asymmetry in the morphogenetic capacities of anterior and posterior cells, even after transplantation to an ectopic site. Anterior skin transplanted to the posterior side induced normal-looking or mildly hypomorphic regenerates, while posterior skin transplanted anteriorly induced ectopic anterior Shh expression and polydactylous regenerates (Slack 1980, Maden 1982, Iwata 2020)."*

[REDACTED]

The authors state, “Importantly, *Fgf8* and *Shh* are not expressed in the uninjured salamander limb meaning that . . . “ Can this actually be defended from the RNAseq data (that are presented here and that are available publicly from other studies)?

- Yes, this can be defended from our bulk RNA-seq data.
 - For *Fgf8*, we detect an average of 0.7 Counts per million (CPM) across anterior and posterior samples. For *Shh* we detect 0.1 CPM.
 - For reference, we detect *Hand2* at 153.6 CPM in posterior samples and its background level in anterior samples is 13.1 CPM. Thus, the values for *Fgf8* and *Shh* likely reflect background noise.

What is the control for leakiness of the Tamoxifen-Cre experiments? This is important because the data and interpretation they present hinges on these reporters faithfully recapitulating the endogenous expression domains, no more and no less. They did have trouble with one line not efficiently recombining (see below), necessitating three separate tamoxifen injections. This means we should remember that they might still not hit all of the cells capable of turning that promoter on at that time of induction. What they don't address is the flip side—can they convincingly show that there is no ectopic expression of the Cre reporter in the absence of Tamoxifen? We need to see low-mag photos of whole limbs, probably in sections on a slide, to address this concern.

- We have assessed leakiness of our Cre reporters using flow cytometry. We believe this to be a more quantitative readout than imaging tissue sections.
- We found very little expression of the Cre reporter in the absence of 4-OHT:
 - We prepared *ZRS-Cre* and *Hand2-Cre* lineage tracing animals and genotyped them individually to ensure that they carried both Cre and loxP-Stop-loxP-mCherry genetic cassettes.
 - We treated half of the animals with 4-OHT during development to induce mCherry expression (positive control) while keeping the other half untreated to assess non-specific mCherry expression (leakiness).
 - We harvested limbs from 6 cm animals, dissociated them into single cells and analysed mCherry expression using a flow cytometer.
 - In untreated animals, we found:
 - 28 mCherry+ events out of 184,353 measured events for *ZRS-Cre* (0.015%, $n = 12$ animals, **panel a**).
 - 92 mCherry+ events out of 339,174 measured events for *Hand2-Cre* (0.027%, $n = 12$ animals, **panel b**).
 - These percentages of spurious conversion are well below the percentages observed in other, published tracing experiments using Cre-loxP during regeneration, for example PMID: 21571227 (Figure 5E, 0.1-0.2% leakiness).

- We have added these data to Extended Data Fig. 2h and Extended Data Fig. 5f.

The authors say, “Having confirmed that TFP labelled *Shh* cells during limb development and regeneration . . .” Are these F1 animals where every cell capable of expressing the reporter has the reporter? There should be more rigorous analysis to demonstrate every TFP cell expresses *Shh* (by co-labelling with the TFP fluorophore and with HCR-FISH).

- Yes, these animals are F1 generation or later (i.e. every cell carries the reporter).
 - We now clarify this in a table in Extended Data Fig. 1a.
- We quantified the overlap between TFP mRNA and *Shh* mRNA on tissue sections using HCR *in situ* hybridisation (**Panels below**). We obtained similar values for developing and regenerating limbs (**Table below**). We found that the ZRS > TFP reporter is efficient, with only low expression outside of the *Shh* domain.
 - We expect some of the non-specificity to stem from TFP transcripts perduring in cells that previously expressed *Shh* but switched it off. Additionally, we expect to detect TFP mRNA more sensitively than *Shh* mRNA. This is because TFP HCR signal is stronger than that of endogenous *Shh* (more TFP transcripts expressed per cell than *Shh* due to being an enhancer reporter).

- We have added these data to Extended Data Figs. 2e-f.

To rule out a possible requirement for embryonically-derived *Shh* cells, the authors surgically removed these cells and asked what happened following amputation. They note that they only successfully removed 89% of the cells, yet the claims they make are stated as if 100% were removed. This is important because residual cells may provide enough function to execute the role of the larger group. Wording needs to be fixed to address this concern. Ideally, a genetic ablation method for killing relevant cells without surgery would also be employed here to state these kinds of conclusions. This is probably outside the scope of what can be required in a

revision because nobody has published a reliable, effective cell ablation method for axolotls (yet).

- We have fixed the wording and added more explanation to address this:
 - “animals ~~lacking~~ embryonic Shh cells” -> “animals with reduced embryonic Shh cells”.
 - “It cannot be excluded that the residual 10% of Shh lineage cells in this assay fully substituted the function of the removed cells by proliferating and initiating Shh expression. However, we find this unlikely, given that apparently normal Shh expression is achieved by 9 dpa (within ~2 cell division times of an axolotl limb blastema cell). Taken together with the lineage tracing during normal regeneration, our data strongly support that embryonic Shh cells are dispensable for generating a regenerative Shh signalling centre.”
 - “We ~~conclude~~ that” -> “We infer that”.

Relatedly, are there other possible conclusions from the surgical removal of embryonic Shh cells, other than that some “wider domain” of posterior cells in mature limb is competent to express Shh?

- Please see above.
- Additionally, our experiments are consistent with investigations by Slack, Tank and others in the 70’s and 80’s, which indicated that the posterior-competent domain in the mature limb is wider than the embryonic *Shh* lineage that we have labelled (see e.g. PMID:7003051, PMID:751844, PMID:7371977)

Digit reduction following amputation is only interesting with respect to this subject (Figure 3h-l, extended data 6i-j) if the authors show controls have no digit reduction. Controls are essential for making the conclusions they make in the paper. Yet, the number of controls is very meager, and it is unclear if some of the experimental groups having deviations from the normal number of digits is significant given the number of samples in the experimental groups as well.

- Below are the *n* numbers for the experiments relating to digit reduction, which we believe are sufficient. Control animals never deviated from the normal, 4 digits.

Experiment	Figure panel	n number of controls	n number of test
Hand2 CRISPrant has fewer digits after amputation	3d	46 limbs	112 limbs
Prrx1-Hand2 misexpression has fewer digits	4e (ZRS>TFP genetic background)	24 limbs	46 limbs
	Extended Data 7i (Hand2 :EGFP genetic background)	24 limbs	68 limbs

For example, is N=6 for the ALM model enough to conclude that if accessory limbs fail to form in the experimental group, this failure is attributable to genotype of graft rather than experimental noise? What is the standard ALM success rate in the authors’ hands and in the hands of others (published accounts) with and without epidermal tissue grafted? These authors appear to only have a 50% success rate with wild-type grafts, as they point out, and that is only N=3/6.

- We agree that the *n* number for this experiment was low and have repeated the assay after some practice to improve the efficiency of control ALM formation.

- We reached the same conclusion that ALM fails when *Hand2* is mutated.
 - We performed 6 more control ALMs and 6 more *Hand2* CRISPAnt ALMs. The *Hand2* CRISPAnts were generated using the new sgRNAs targeting *Hand2* M146 (please see later for explanation). Each host animal received one control ALM and one CRISPAnt ALM to opposite arms.
 - 5/6 control ALM surgeries induced accessory limbs (83%, **panel a**) while *Hand2* CRISPAnt grafts did not (0/6, 0%, **panel b**).
 - We tested if this difference is statistically significant using a Wilcoxon signed rank test. We tested if the number of ALM digits is significantly different from the assumed 4 digits of wild type limbs (no ALM = 0 digits). The lack of ALMs in the CRISPAnt condition was statistically significant ($p = 0.03$, $W = -21.00$, $n = 6$).
 - We also performed the same statistical analyses on our original ALM datasets, and found that the difference between control and CRISPAnt grafts was statistically significant also in these cases.
- For reference, the success rate of control ALM formation was 74% (11/15) in the original ALM methods paper (PMID: 15136146). Thus, our control ALM efficiency is comparable to that in the original publication.

- We have added these data to **Extended Data Fig. 6n**.

Lines 285-301: Is this the right experiment? Why ectopically express *Hand2* everywhere rather than only in the anterior domain? The interpretation of the results is hairy given how the experiment was performed. Please discuss.

- We agree that anterior-restricted misexpression would be preferable but currently lack a transgenic animal to perform this experiment.
 - Instead, we have re-analysed our misexpression data in which we misexpressed mCherry-*Hand2* using the *Prrx1* enhancer (F0, mosaic). We found three limb buds in which misexpression was predominantly restricted to the anterior side. ZRS>TFP was expressed anteriorly in all three limb buds (black arrowheads, **panel a**), similar to limb buds in which mCherry-*Hand2* was misexpressed throughout the limb field. This upholds our conclusion that *Hand2* is sufficient for *Shh* expression.

a 3 examples of predominantly anterior *Hand2* misexpression

mCherry-Hand2 ZRS > TFP

- We have adjusted the wording to discuss this:
 - *“Ideally, we wished to misexpress Hand2 only in the anterior domain of the limb field; however, we currently lack a transgenic tool that can achieve this reliably. We therefore injected fertilised eggs with an integrating construct harbouring the mouse Prrx1 limb enhancer...”*
 - *“In F0 animals, we detected mosaic expression of mCherry-Hand2 at different expression levels and in different spatial domains, presumably depending on the number and locations of transgene copies integrated into the genome.”*
 - *“We found that Hand2-misexpressing limb buds expressed ectopic ZRS>TFP (n = 7/9 limb buds carrying the TFP transgene) and developed polydactyly (n = 7/16 limbs total), including in three cases that displayed predominantly anterior misexpression”.*

Some claims are difficult to evaluate because incomplete info is provided; particularly lacking are low-magnification views that would enable reader to assess how specific some of the markers or genes of interest are for the areas of tissue in question. For example, how specific is the probe shown in figure 5h? We can't really evaluate that question because all they present is a very high-magnification view. At the least, some of these need to be shown in full limbs, probably in supplemental material so they can be displayed with high resolution and large, and those images need to be at a reasonable enough exposure time to believe that if positive cells were outside the reported expression domain, they would be detected.

- We now demonstrate the specificity of the *Shh* probe by providing low-magnification views of:
 - An entire stained limb bud, with 3D views (**panel a**)
 - The full blastema depicted in figure 5h (**panel b**).

- We have added these data to **Extended Data Fig. 2c** and **Extended Data Fig. 9c**.

Another spot in question is the high-magnification view of the blastema in figure 2k. It is important to also present the low-magnification view featuring the full limb with the overlay of the two signals?

- This image already depicts the full blastema with the overlay of the two signals (*Shh*^{HCR} and mCherry).

- However, to facilitate interpretation of this result, we now provide single channel grayscale images in **Extended Data Fig. 5i**.

Claims that a limb has been “completely posteriorized” are weakly supported. The authors have the kinds of data in hand that would guide a reasonable amount of experimentation to more responsibly make this statement—they have the expression data from their sequencing study. They need to do more to characterize these markers to make this claim.

- We believe that our functional test (ALM using *Hand2*-misexpressing anterior skin) demonstrates posteriorisation of function, beyond just changes in marker gene expression (**Figures 4h-i**).

- However, we have now assessed gene expression changes induced by *Hand2*-misexpression using RNA-sequencing.
 - We physically isolated the anterior blastema from *Prrx1>mCherry* (control) or *Prrx1>mCherry-Hand2* (test) animals and purified mCherry-positive cells by FACS. For additional comparison, we sequenced anterior blastema cells isolated from our *Alx4:mCherry* transgenic. Although *Alx4:mCherry* labels both anterior blastema cells and a subset of posterior cells (**Extended Data Fig. 8b**), we could selectively purify anterior cells by simultaneously sorting against the posterior *Hand2:EGFP* reporter present in the genetic background.
 - We found that *Hand2*-misexpressing cells acquired posterior cell signatures:
 - Upregulated posterior transcription factors *Klf8*, *Hoxd13* and endogenous *Hand2*.
 - Downregulated anterior transcription factors *Lhx9*, *Barx1*, *Zfhx4*, *Hoxc10*, *Lhx2*.
 - We conclude that *Hand2* misexpressing cells acquire some gene expression features of posterior cells. The interpretations are a bit limited due to the necessity of using hypomorphic *Hand2*-misexpression animals (discussed in the manuscript).
- We have added these data to **Extended Data Tables 3-4** and **Supplementary Tables 3-6**. The manuscript now has a new results section:
 - *“To identify the gene expression changes underlying Hand2-induced posteriorisation, we performed RNA-sequencing. We amputated Prrx1>mCherry-Hand2 misexpression limbs and Prrx1>mCherry control limbs. We were limited to analysing Prrx1>mCherry-Hand2 limbs with hypomorphic phenotypes, since strong misexpression blocks limb outgrowth. At 14 dpa, we harvested the anterior blastema from these animals and FACS isolated mCherry-labelled cells to identify Hand2-induced genes. As a second comparison, we isolated bona fide anterior blastema cells labelled with Alx4:mCherry (see below, and Methods). As the Hand2 misexpression cassette is codon-altered, we could discriminate endogenous Hand2 transcripts. We found that Hand2-misexpressing cells acquired features of posterior cells: they upregulated posterior transcription factors (Hand2, Hoxd13, Klf8) and downregulated anterior factors (Lhx2, Lhx9, Barx1, Zfhx4, Hoxc10) (Extended Data Tables 3-4). Therefore, Hand2 induces transcriptional changes towards a posterior identity beyond Shh expression. The full lists of differentially expressed genes in Hand2-misexpressing cells are available in Supplementary Tables 3-6.”*

Are the ectopic digits produced really posterior digits in figures 4b and 4c? How do we know? This has been a persistent issue/debate in limb development field over the years, and there are ways that these authors can shore up their claims to support, for example, at a molecular level, that these digits actually are posterior digits.

- We do not claim that the ectopic digits are posterior. Indeed, based on previous *Shh* misexpression in axolotl limbs using Vaccinia virus, we also do not necessarily expect this (PMID 10656763, as pointed out by Reviewer 1).
 - Pasted from this reference: *“Rather than the mirror-image duplicated patterns seen in developing chick limbs (Riddle et al., 1993), these limbs [Shh misexpression] formed expanded hands. In expanded hands, as defined by Iten and Bryant (1975), the most anterior and posterior digits and carpals/tarsals appear normally organized, but there are a variable number of extra digits (and*

carpals/tarsals) in between. These extra digits are difficult to identify with certainty, but the most posterior of the internal set in any of these limbs, based on criteria described in Iten and Bryant (1975), is either digit 3 or digit 2".

Is there precedent for CRISPR mutagenesis and genotyping using alterations to codon (making base pair diffs that do not change amino acid sequence)? If so, please cite.

- We are not sure why this is relevant to our mutant.

Important question—one of the biggest technological issues with the paper: The authors do not use established, homozygous lines for their claims about loss-of-function. They use F0 animals. Axolotl generation time is about one year, so clearly doing the full-on genetics will take significantly more time. However, it is critical that they make clear that they are using animals that are not clean lines and the caveats that implies. Because they only use this one mutagenesis approach to support their claims about Hand2, and no parallel methods, there are important shortcomings that should be addressed. These are crispants. The cells will be mixture of different alleles, and some will be differentially edited at the two different alleles. Therefore, typical validations such as characterizing RNA expression, or even the sequence of the RNA expressed, cannot be performed. There will be too many different alleles/RNA species with potentially different effects. If they had a stable line, they could confirm the loss of the mRNA expression, or at least that the resultant sequence was untranslatable. This issue is critical for this work, but these particular authors addressing this exact issue is something that is highly critical for the field. The reason is because the Tanaka lab has pioneered these techniques in axolotl, and the work it produces will be taken as a sort of “industry standard” down the line when other papers are considered and peer reviewed. It sets a precedent. Most people working on axolotl would benefit, publication-wise, if F0 data is published as loss-of-function proof in axolotl in Nature. Use of F0 will be more commonly accepted in peer-reviewed publications, which can be good to accelerate sharing results, but it always needs to be coupled with clear explanations of the potential downfalls of this strategy/short-cut. As a counterpoint to what is presented in this work, for their 2022 paper published in Development, Takeuchi et al. went further than crispant-based data in a newt study. They intercrossed F1 animals with their Hoxa13 CRISPR-induced alleles to generate F2 animals, a real allelic series in a stable genetic setting. This appears to have been required to make the loss-of-function claims with respect to patterning along the proximodistal axis. So it is possible. It is also possible to make cohorts of F1 animals that at least all have the same genotype by mating two F0 and selecting those offspring with equivalent alleles for study.

- We thank the reviewer for this comment. We had expected embryonic lethality of axolotl *Hand2* mutants based on mouse and zebrafish mutants, as well as the high lethality among F0 *Hand2* CRISPR animals (Figure 3). We now present data demonstrating that germline-transmitted *Hand2* mutant axolotls are lethal prior to 2 months (limb completion), justifying our use of CRISPRants.
 - We inter-crossed a stable F1 heterozygous *Hand2* mutant to an F0 *Hand2* CRISPRant (**panel a**).
 - We were unable to inter-cross two F1 heterozygotes due to animal availability.
 - The mutant allele of the F1 parent is a 64 bp deletion that removes the *Hand2* translational start. We refer to this deletion allele as *Hand2*^{D64}.
 - The F0 parent contributes various mutant alleles, including *Hand2*^{D64}.
 - We identified progeny homozygous mutant for *Hand2*^{D64} using a 3-primer PCR strategy (**panels b-c**). Lethality among these animals was **96.7%** ($n = 29/30$).

- For comparison, the lethality among sibling progeny was 19.4% ($n = 12/62$). This includes trans-heterozygous animals carrying *Hand2*-inactivating mutations that cannot be detected by PCR (small indels/frameshifts).

- We have included this result in **Extended Data Figs. 6r-s** and added the following text:
 - “As part of our *Hand2* characterisations, we attempted to derive F2 knockout animals that were homozygous mutant for the 64 bp deletion around the *Hand2* translational start site (*Hand2*^{Δ64}, **Extended Data Fig. 6q-s**). However, we found that homozygous *Hand2* mutant axolotls were 96.7% lethal before the end of limb development ($n = 29/30$ animals), requiring us to use CRISPAnt animals in these assays.
- We view the ability to make chimeric F0 CRISPAnts as an interesting experimental opportunity that gave us access to functional dissection of *Hand2* function postembryonically. As suggested by the reviewer, we have ultimately targeted two distinct locations of the *Hand2* coding sequence (using 5 gRNAs), with comparable results (see below), to demonstrate specificity of the phenotype, negating off-target effects.

The alleles themselves should be further analyzed with experiments

- Please see our characterisation of the *Hand2*^{Δ64} allele, above.

The targeted removal of the start codon may be an odd choice. The authors mention they did this and it is schematized in the extended data. However, they never explain why they designed the targeting strategy this way, nor do they provide a citation of anyone else doing this before. There are good reasons why this could be problematic. *Hand2* is a 2-exon gene with a single intron. There is a second methionine at amino acid position 146, still within the first exon, but comfortably far away from their double gRNA strategy. Thus, it is quite possible this locus could be translated into an abbreviated version of *Hand2* in their mutants (aka crispants). If this other methionine is being used in the animals to make a version of *Hand2* protein, it would be an N-terminal truncated version, and all bets are off about what kind of function that truncated protein might have. It could be dominant negative or geomorphic, or it could have wild-type function (or it could be a null, which is probably what they hope, but they have not evaluated nor even discussed these important issues). For context, please see Tuladhar et al., 2019, Nature Communications, “CRISPR-Cas9-based mutagenesis frequently provokes on-target mRNA misregulation” (PMID 31492834). They addressed these issues using a variety of genes with well-established biological roles, and they found that ~50% of the induced alleles (across intended to be loss-of-function null alleles) indeed produce some type of protein and that these can have a variety of functions with respect to what the wild-type protein does and what the true

null phenotype would be.

Why not use a single guide right after the M146-encoding codon, but before the intron? This would likely be sufficient. Alternatively, they could use two guides to cut the whole gene out or the whole first exon, or they could take out the transcriptional start site. The results of this study, as presented, may turn out to match the results of these more rigorous methods for creating null alleles, but we don't know. What is presented cannot be claimed to be a true loss-of-function phenotype without more defense. It currently reads as, "We found the phenotype we predicted, so it must have worked as imagined."

- Thank you for this suggestion. We designed 5 new gRNAs that target sequences at (or after) M146 and before the intron, as suggested by the reviewer (**panel a**).
- We injected these gRNAs individually into 1-cell stage eggs and assessed the limb phenotypes of these new CRISPs at 8 cm animal size.
 - 3 gRNAs (#2, #3, #4) induced limb patterning phenotypes paralleling those observed with the original 2-gRNA strategy (and at comparable frequencies) - **panel b**.
 - gRNA #1 resulted in full penetrance lethality prior to limb formation.
 - gRNA #5 did not induce a phenotype but had a low genome editing score as assessed by ICE Synthego (see **table**). We infer that it is an inefficient gRNA.
- In total, we obtained similar phenotypes from 4 sets of sgRNAs targeting *Hand2*.

	Guide sequence	Animals with limb defect	Editing efficiency (from tail clip)
Control	-	0/8 (0%)	0%
Hand2 gRNA new#1	GCTACATCGCCTACCTCA	Lethal (n/a)	60%
Hand2 gRNA new#2	CCTACCTCATGGACCTGC	9/13 (69%)	48%
Hand2 gRNA new#3	TCATGGACCTGCTGGCCA	4/9 (44%)	36%
Hand2 gRNA new#4	CCAAGGACGAGCAGAGCG	5/12 (42%)	89%
Hand2 gRNA new#5	TGCTCGTCCTTGCCAGC	0/10 (0%)	6%

- We have added the data for gRNAs #2-4 to **Extended Data Figs. 6j-k**. We excluded gRNA#1 as it could not be analysed due to lethality and gRNA#5 as it did not induce efficient editing. In addition, we now explain in the text:
 - “*Hand2* harbours a second Methionine codon within exon 1 (M146). We individually injected 3 sgRNAs targeting sequences at, or just downstream of, M146 to generate mosaically mutated animals that we refer to as *Hand2*^{M146} CRISPs (Extended Data Fig. 6j). We observed similar limb phenotypes in the *Hand2*^{M146} CRISPs as in our original *Hand2* CRISPs, and at comparable frequencies (Extended Data Fig. 6k).”

Another approach could be to use the start-codon guide they already have to take out the fluorescence in their *hand2* knock-ins. If the protein is silenced, the GFP should also be silenced; if the protein is truncated, then the GFP should persist. However, it’s still difficult to interpret this one because there are many possible outcomes. The point stands that if they are going to be permitted to make the claims they want from F0 data, they should at minimum have more validation, and they also need to note the shortcomings of the approach in the text.

- We have now performed this experiment.
- We observed loss of *Hand2*:EGFP signal, supporting silencing of *Hand2* expression by the sgRNAs rather than truncation.
 - We co-injected the 2 sgRNAs targeting the *Hand2* start codon into the left or right blastomeres of 4-cell stage *Hand2*:EGFP eggs (**panel a**). We injected 4-cell stage eggs due to incomplete cell cleavage at the 2-cell stage.
 - This should mutate *Hand2* in one half of the animal, leaving the opposite half as an internal control. Based on experiments in our manuscript, we expected ~45% of injected animals to have sufficient levels of *Hand2* mutation to result in limb bud phenotypes (Figure 3b).
 - As external controls, we injected sibling eggs with injection mix minus sgRNAs.
 - At limb bud stage, 46% of sgRNA-injected larvae ($n = 13/28$ larvae) had reduced limb buds on one side compared to the other, a proportion consistent with our previous experiments. The reduced limb bud invariably had depleted *Hand2*:EGFP signal compared to the opposite limb bud (**panel b**).
 - We calculated the ratio of EGFP signal between the two limb buds, always dividing the brighter limb bud by the dimmer limb bud (**panel c**). (Note: we cannot control which limb bud (left/right) becomes mutated in this assay).
 - Control animals had ratios close to the expected 1 (mean 1.34 ± 0.30 , $n = 11$).
 - sgRNA-injected larvae with limb phenotypes exhibited strong deviations from 1 (mean 13.31 ± 16.50 , $n = 13$ larvae), indicating loss of *Hand2*:EGFP on one side.
 - sgRNA-injected larvae without obvious limb bud phenotypes had ratios close to that of controls (mean 1.61 ± 0.51 , $n = 15$ larvae).

- We have added these data to **Extended Data Figs. 6g-i** and included the following text:
 - “In the first validation, we generated *Hand2* CRISPR/Cas9 mutants in a *Hand2:EGFP* reporter background. *Hand2* mutant cells should not express EGFP. We injected *Hand2* sgRNAs into one side of cleaved *Hand2:EGFP* eggs, allowed these to develop and compared EGFP fluorescence in the left and right limb buds (**Extended Data Fig. 6g**). We found that 46% of sgRNA-injected axolotls displayed a mutant phenotype ($n = 13/28$), a similar frequency as in our original experiments (**Extended Data Fig. 6d**). These animals had significantly less EGFP in one limb bud compared to the other, supporting that *Hand2* had been deleted (**Extended Data Fig. 6h-i**).”

Adding a very clear table about the different lines produced, whether they are F0 mosaic transgenics (random insertion), F0 mosaic knock-ins, F0 mosaic crispants, F1 knock-ins, established reporters, etc. would be very helpful. Extended data figure 1 is still confusing.

- We have added the following table to **Extended Data Fig. 1a**.

a	Short name	Transgenic type	Genetic cassette	Generation	Figures
	Alx4 :mCherry	Knock-in at Alx4 C-terminus	T2A-mCherry	F1+ (germline transmitted)	Fig. 5. Extended Data Fig. 8,9.
	Hand2:EGFP	Knock-in at Hand2 C-terminus	T2A-EGFP	F1+ (germline transmitted)	Fig. 2,5,6. Extended Data Fig. 5-10.
	Hand2 lineage tracer	Knock-in at Hand2 C-terminus	P2A-EGFP-T2A-ER ^{T2} -Cre-ER ^{T2}	F1+ (germline transmitted)	Fig. 2,6. Extended Data Fig. 5,10.
	Hand2 CRISPR/Cas9 mutant targeting translational start of Hand2	CRISPR/Cas9 mutant targeting translational start of Hand2	-	F0 (mosaic)	Fig. 3. Extended Data Fig. 6.
	Hand2 ^{M146} CRISPR/Cas9 mutant targeting M146 of Hand2	CRISPR/Cas9 mutant targeting M146 of Hand2	-	F0 (mosaic)	Extended Data Fig. 6.
	Prrx1 > mCherry	I-SceI (Random integration)	Prrx1 > mCherry	F0 (mosaic) F1+ (germline transmitted)	Fig. 4. Extended Data Fig. 7. Fig. 2,4,6. Extended Data Fig. 7,9.
	Prrx1 > mCherry- Hand2	I-SceI (Random integration)	Prrx1 > mCherry- Hand2	F0 (mosaic) F1+ (germline transmitted)	Fig. 4. Extended Data Fig. 7. Fig. 4. Extended Data Fig. 7.
	ZRS axolotl	I-SceI (Random integration)	ZRS > TFP _{nl5} -T2A-ER ^{T2} -Cre-ER ^{T2}	F1+ (germline transmitted)	Fig. 1,2,3,4. Extended Data Fig. 2,3,6,7.

It may be possible for the authors to use a complementary method to prove their points without having to derive new alleles or breed these to true homozygosity. It is likely that other reviewers outside the axolotl field will be more insistent on true F2 homozygotes versus wt sibs for these studies (for example, zebrafish).

- We hope that our additional validations, including generating new alleles, sufficiently support our points.

Other issues to address/consider in contextualizing this work:

There are no references to the two key papers by Stéphane Roy, Susan Bryant, David Gardiner. These papers are foundational for this research question and must be presented. In one published in 1999 (PMID 10656763), it was demonstrated (through ectopic expression using Vaccinia viruses) that ectopic Shh expression in axolotl blastemas causes mirror-image duplications of digits along the A/P axis, which resembles what is seen in mouse and chick. In the other (PMID 12115913, published in 2002), it was demonstrated that inhibiting Shh activity (pharmacologically, using cyclopamine) reduces axolotl digits in the manner predicted based on mouse and chick. Both of these papers together underscore that cell behaviors downstream of Shh activity are similar in the contexts of development and regeneration. Overall, they established this to be true using the axolotl model. This work presented in this paper clearly stands on that foundational work, yet it is never mentioned in the text.

- Thank you for these suggestions; we have now added these:
 - *“Inhibition or misexpression of Shh in axolotls yields comparable digit reduction or expansion phenotypes during axolotl limb regeneration as during chick (*Gallus gallus*) and mouse (*Mus musculus*) limb development, suggesting broad similarities in the cellular pathways that operate downstream (Roy 2002, Roy 2000).”*
 - *“The polydactylous phenotype seen after mosaic Hand2 misexpression is similar to that seen after viral misexpression of Shh in axolotl limbs (Roy 2000).”*
- Just a small correction: in contrast to mouse and chick, Shh misexpression gave rise to expanded hands with extra digits in axolotls, but never to mirror-image duplications. This is relevant as we obtained a similar result in our study.

Lines 166-167 reference a study in zebrafish that came to similar conclusions as this paper with respect to Shh. Additional context for why the axolotl study is so novel is necessary, which should be beyond the idea that now the appendage under study is a tetrapod limb.

- The following sentence in the discussion addresses this point directly:
 - *“Interestingly, Hand2 was previously implicated in posterior identity in the regenerative zebrafish pectoral fin, although in this system, Hand2 was not coupled to Shh and memory effects were not assayed²⁹.”*

Referee #2 (Remarks to the Author):

Otsuki and colleagues investigated how anterior-posterior CT cells maintain their positional features in the axolotl limb. Their research began by showing that during limb regeneration, posterior Shh expression doesn't necessarily originate from cells that once expressed Shh. The authors revealed the dynamics and lineage of posterior CT-associated Hand2+ cell expression. Through a series of thorough gain- and loss-of-function experiments, a solid link between Hand2 and Shh was established. Using a double transgenic model, the work revealed that anterior to posterior transplants could activate Hand2 upon amputation. Conversely, posterior-to-anterior transplants were unable to stimulate anterior-associated Alx4 expression upon amputation. Instead, these cells maintained Hand2 expression, activated Shh, and resulted in abnormal patterning. Further experiments showed that some cells that once expressed Alx4 in

anterior-posterior grafts were able to express some Shh. Shh was shown to mediate the activation of Hand2 in anterior to posterior transplants. The team also used a memory assay, which showed that even brief exposure to Shh can prime anterior cells to express Hand2. Based on these, the authors proposed a model to explain how AP axis info is retained during regeneration, emphasizing the dominant role of Hand2 expression and posterior cells.

The study is methodically planned and excellently implemented by combining sophisticated axolotl transgenic lines with traditional grafting studies, effectively utilizing the axolotl's distinctive features. The high standard of presentation of the research deserves special mention. This study is a commendable extension of the Tanaka Lab's prior work, Nacu et al (2016), which initially identified the Shh-Fgf8 loop as the mechanism controlling anterior-posterior positional information. This manuscript is an important work and would be of interest to developmental and regeneration biologists. I am unsure if its reach can hit the broader-audience, as the topic is rather niche and their earlier work (Nacu et al 2016) has conceptually proposed the molecular basis for the anterior-posterior features, and this work elegantly adds more components. The most innovative experiments are shown in Figures 5 and 6. The established link between Shh and Hand2 (shown in Figures 2, 3, and 4) mirrors the known link in other species, as the authors also mentioned. The post-embryonic posterior expression of Hand2 is another novel finding. The experiments and results are sufficiently detailed in the figures/methods, and the figures are designed to make even the complicated experiments easy to understand. Although it is a very high-quality work, my biggest frustration with this manuscript is that, the text and interpretations contain many terminologies that oversophisticate/complicate findings. Claims of “memory”, “reprogramming”, “positional identity” require substantial revisions via editing the text and performing new experiments. The proposed model may not be able to explain some performed experiments or findings in the literature. Also, the results are interesting, but their broader implications are not substantiated, giving the impression of overstating the importance of the findings. Without overly sophisticated terminologies and resulting complications, this is an interesting and solid work.

Comments

1. One of the central issues of the manuscript revolves around the ambiguous description of anterior/posterior "identity," demarcated by Alx4 and Hand2/Shh expression. If a genuine identity component exists, one would anticipate a pronounced transcriptional difference between these populations, especially considering that Alx4 and Hand2 are transcription factors. It would be necessary to explore available scRNA-seq datasets to confirm if Alx4 and Hand2 cells manifest significant differences and subsequently assay additional marker gene expressions to bolster the "identity" element beyond a singular gene observation. However, I remain dubious about the presence of a major transcriptional shift. Previously published limb scRNA-seq datasets, including those from Tanaka Lab, have failed to distinguish between anterior and posterior CT. This suggests that positional information might not represent distinct identities. Instead, CT cells may possess certain plasticity, with Hand2 inducing stable dominant alterations within this framework.

- We would like to refer the reviewer to our bulk RNA-seq dataset (**Figure 2b and Extended Data Fig. 4a-h**), which reveals a pronounced transcriptional difference between anterior and posterior cells: 146 genes are upregulated anteriorly and 160 genes are upregulated posteriorly. In newly added RNA-seq data, we also show that anterior-to-posterior cell transplantation shifts overall transcriptome identity (PCA, please see below, **Extended Fig. 9d**). Both datasets demonstrate that anterior and posterior cells have genuine molecular differences beyond just *Alx4* and *Hand2*.

- It is likely that scRNA-seq lacks the sensitivity to detect these differences--dropouts, especially for lowly expressed genes such as for transcription factors, is a known phenomenon in single cell data. We checked the dataset that the reviewer mentioned (PMID: 30262634). Table S5 shows that, of 2,375 uninjured upper arm connective tissue cells profiled with 10X scRNA-seq, *Hand2* was detected in only 250 cells (10.5%). For *Alx4*, the number is only 4 cells (0.2%). These are clear under-representations based on our *in vivo* characterisations. Thus, it is unsurprising that anterior-posterior differences have not been detected in existing scRNA-seq data for uninjured cells.
- We showed that *Hand2* is upregulated ~6-fold from steady state to regeneration (**Figure 2f**). Although 10X scRNA-seq indeed detects more *Hand2*+ cells during regeneration (4,150 *Hand2*+ cells detected out of 21,819 cells (19.0%), Table S9 of PMID: 30262634), even this proportion is an under-representation. It is likely that the expression levels of transcription factors relevant to our manuscript are generally at the lower limit that can be detected with scRNA-seq. This supports our use of bulk RNA-seq to identify lowly expressed genes.

2. The authors employ the term "reprogramming" based on a transition from *Alx4* to *Hand2* expression. However, this alteration might not necessarily denote reprogramming, as the term "reprogramming" is used to describe perturbations that induce stable cell identity changes. Recent research further shows reprogramming via single-cell transcriptomics-based clustering analysis for validation. The authors could undertake scRNA-Seq on sorted *Alx4*+ and *Hand2*+ cells after their grafting experiments depicted in Fig 5 with appropriate controls to validate their reprogramming assertion. Yet, I remain uncertain if the authors would genuinely identify a cell fate alteration qualifying as reprogramming. Based on extensive publicly accessible scRNA-seq data, even cells expressing *Shh* (potentially inclusive of *hand2*+) don't segregate into distinct clusters indicative of a new, stable cell type. Similarly, anterior and posterior CT cells lack discernible transcriptome distinctions in such datasets. The manuscript's bulk RNA-seq results further underscore the idea that CT cells are malleable, reacting to external cues, with *Shh* acting as a potent inducer of stable *Hand2*. Given these factors, it remains ambiguous whether the observed phenotypes in this study genuinely epitomize positional identity, memory, or reprogramming.

- Regarding positional identity, please see our response above. We argue that scRNA-seq does not have the sensitivity to detect the lowly expressed gene differences between steady state cells. We find that cells in the uninjured limb have very low mRNA content, exacerbating sensitivity issues in scRNA approaches. More broadly, we argue that cell identity refers to cells expressing a different transcriptional profile under the influence of extracellular cues. Our anterior vs posterior dermal RNA-seq dataset demonstrate this abundantly.
- Regarding positional memory: positional memory has been defined functionally in the context of regeneration as a transplantable and heritable property irrespective of molecular mechanism. In this manuscript, we have conferred this memory property to cells by transplanting them posteriorly or by treating them with SAG. Cells treated in this manner can express *Shh* following a second amputation several months after first treatment, an ability that they do not normally have. This prolonged time frame far exceeds the duration of a single cell cycle and likely reflects a stable, epigenetically altered state consistent with cellular memory.
 - However, we agree that positional memory could be explained better in the manuscript and have added some clarifications to the text:

- “Given that this phenotype *manifested in the following round of regeneration*, we infer that the positional identity of the anterior cell was stably posteriorized in the transplantation assay and deem this to be an alteration to positional memory.”
- “Importantly, the positional memory of these anterior cells became stably posteriorized, as their *descendants expressed Hand2:EGFP when we allowed the limb to fully regenerate and then re-amputated it (Fig. 6e, Extended Data Fig. 10f).*”
- Regarding reprogramming: in addition to our comment about stable positional memory (above), we have newly performed RNA sequencing to assess the extent of gene expression change in posteriorized anterior cells. We found that posteriorized cells downregulated many anterior transcription factors and approached a posterior cell identity as assessed by PCA (**Extended Data Figs. 9d-g**). Please see the following response.

3. In connection with Fig 5, when grafting from anterior to posterior limbs, it's possible that anterior genes, aside from *alx4*, remain active in the grafts. The authors should perform staining for anterior and posterior markers in these experiments. Ideally, conducting scRNA-seq on these grafted samples would offer definitive data regarding the extent to which anterior or posterior genes have altered, providing support about positional characteristics as distinct identities.

- To obtain a more comprehensive readout, we chose to compare transplanted cells *versus* ‘normal’ anterior and posterior blastema cells using RNA-sequencing.
 - Principal component analysis (PCA) revealed that anterior cells transplanted to the posterior limb acquired an overall transcriptome identity closer to posterior blastema cells compared to both anterior blastema cells and anterior cells transplanted to the anterior (**panel a**).
 - We analysed the gene loadings for PC2, a principal component that correlated strongly with this transition (explaining 25% of variance). PC2 was driven heavily by posterior genes (e.g. *Hand2*, *HoxD13*) versus anterior genes (e.g. *Lhx9*) (**panel b**). This supports an identity switch.
 - We performed differential expression analyses. We found that posteriorly grafted cells both downregulate anterior transcription factors (*Lhx9*, *Dmrt2*, *Pbx3*, *HoxC10*, *Tbx22*, *Alx1*, *Zfhx4*) and upregulate posterior transcription factors (*Hand2*, *HoxD13*) (**panels c-d**), while noting that bulk RNA-seq cannot distinguish heterogeneities between individual cells.
- We conclude that posteriorly transplanted cells downregulate anterior genes, supporting an identity change (whether requiring extracellular signaling or not). Moreover, PCA supports a major identity transition towards a posterior cell identity state.

- We have added these data to **Extended Data Figs. 9d-g**, **Extended Data Tables 5-6** and **Supplementary Tables 7-11**.
- We have added a new results section to the manuscript to explain these results:
 - **“Anterior cells acquire posterior gene expression and lose anterior gene expression during reprogramming**

To reveal the extent to which anterior cells become posteriorized in our transplantation assay, we performed RNA-sequencing on anterior cells transplanted posteriorly or anteriorly (2 replicates each), along with non-transplanted controls (4 replicates each). In more detail, we transplanted double reporter cells taken from anterior limb to the posterior (A->P, test) or to the anterior (A->A, control) region of unlabelled host limbs, then performed amputations. All donor cells were initially *Alx4*:mCherry-positive and *Hand2*:EGFP-negative. As before, A->P cells became *Hand2*:EGFP-positive in the blastema (**Fig. 5d**), while A->A cells remained EGFP-negative. After these limbs had regenerated, we performed a second amputation as before. We FACS purified A->P descendant cells (*Hand2*:EGFP-positive) and A->A descendant cells (*Alx4*:mCherry-positive) from these second blastemas for RNA sequencing.

We performed principal component analysis (PCA) to compare gene expression across the four sample types (**Extended Data Fig. 9d**). We found that the first principal component, PC1, discriminated samples based on the presence/absence of transplantation. More interestingly, PC2 discriminated A and A->A cells from P and A->P cells, explaining 25% of total variance. We analysed the gene loadings for PC2 and found that this principal component was driven by anterior-posterior identity factors (**Extended Data Fig. 9e**, **Supplementary Table 7**). *Hand2* and EGFP were among the top contributors to PC2, which is expected given that A->P and P cells were isolated based on *Hand2*:EGFP expression. However, we also found other anterior-posterior transcription factors contributing to PC2, including *Hoxd13* and *Lhx9* (**Extended Data Fig. 9e**). Crucially, the PCA plot for PC1 and PC2 revealed that A->P cells

were more similar to P cells than to either A cells or A->A cells (**Extended Data Fig. 9d**). This supports a widespread transcriptional shift of transplanted anterior cells towards a posterior identity.

Next, we analysed the expression of anterior and posterior transcription factors in these samples (**Extended Data Fig. 4a-b**). We were particularly interested to see if anterior factors remain on, or become downregulated, during this identity change. By comparing A->P transplanted samples against normal A (anterior blastema) cells, we discovered that A->P cells not only upregulate posterior transcription factors (*Hand2*, *Hoxd13*) (**Extended Data Fig. 9f, Extended Data Table 5**) but additionally downregulate anterior transcription factors (*Alx1*, *Lhx9*, *Dmrt2*, *Pbx3*, *Hoxc10*, *Tbx22*, *Zfhx4*) (**Extended Data Fig. 9g, Extended Data Table 5**). We also detected *Shh* transcripts in the A->P transplanted datasets, validating our conclusion from *in situ* hybridisation that A->P cells can newly express *Shh* (**Figs. 5e-f, Extended Data Fig. 9f**). Thus, transplanted cells not only gain posterior identity markers but leave their original, anterior-identity state. Importantly, the loss of anterior markers is not an artefact of the transplantation assay. Comparison of A->P transplanted cells with A->A transplanted controls demonstrated that loss of anterior genes (*Pbx3*, *Dmrt2* and *Hoxc10*) only occurred in A->P transplantations (**Extended Data Table 6**). The full lists of differentially expressed genes in A->P cells are available in **Supplementary Tables 8-11**.

Our analysis revealed that anterior-to-posterior cell transplantation caused substantial shifts in mRNA expression, resulting in a posterior-like expression signature. These data support a reprogramming event that induces lasting changes to gene expression and signalling potential (Shh expression)."

4. In several sections of the text and figures, there's ambiguity regarding how the authors differentiate between "memory" and "positional identity". For instance, in the titles of Fig 5 and 6, substituting "memory" with "positional identity" wouldn't seem to alter the meaning. Could the authors elucidate their criteria for distinguishing between memory and positional identity, aside from the experiment in Fig 6d and e (which appears to be the sole definitive assay for memory)?

- Thank you for this comment. We defined 'memory' as a heritable effect that can be detected in the subsequent regeneration round. In this sense, we demonstrated heritable effects in Main Figures 5e/h and 6d-e so the titles of these figures seem appropriate to us. However, we have re-checked the manuscript carefully and taken care only to use 'memory' to describe the appropriate experiments in our revised submission.
- We have added clarifications throughout the text for results pertaining to memory:
 - *"Given that this phenotype manifested in the following round of regeneration, we infer that the positional identity of the anterior cell was stably posteriorized in the transplantation assay and deem this to be an alteration to positional memory."*
 - *"Importantly, the positional memory of these anterior cells became stably posteriorized, as their descendants expressed Hand2:EGFP when we allowed the limb to fully regenerate and then re-amputated it (Fig. 6e, Extended Data Fig. 10f)."*

5. In Extended Data 4, while RNAseq results consistently highlight *Hand2* and a few others as being enriched in the posterior, there seems to be inconsistency in the replicates for genes associated with signaling pathways. It might be prudent to exclude conclusions related to these pathways, given that the findings aren't stemming from reproducible data. Moreover, such information detracts from the central narrative and doesn't offer significant insight.

- We have removed the data and conclusions for these signalling pathway genes from **Extended Data Fig. 4**.

6. The conclusion made by the authors on lines 230/510, stating that "*hand2* encodes posterior identity," lacks adequate support. If *Hand2* indeed encoded posterior identity, its overexpression would result in double-posterior limbs, evident through mirrored digit morphologies (PMID: 15192229, 2226209). However, this phenotype isn't observed in the present work. The conclusion on lines 230/510 needs revisiting. Interestingly, other portions of the text seem to address this topic with greater precision.

- Thank you for this comment regarding the double-posterior limb phenotype.
- As explained in the text, we expect a full penetrance double posterior limb field not to grow out, and this is indeed the phenotype that we observed (**Figs. 4e,g, Extended Data Figs. 7i-j**). In Stocum's past P-P transplantation work we infer that he transplanted posterior plus some anterior tissue in those transplants, as he had no marker for the posteriorly determined cells (PMID: 644323). In contrast, by *Hand2* overexpression, we can make all cells posterior, so that there is no longer anterior and posterior cells in the limb bud. Based on theoretical and practical work, we expect this posterior-only situation to yield no outgrowth or regeneration.
- Regarding our mosaic misexpressions of *Hand2*, we expected expanded hands and never mirrored digits (as reported in the *Shh* overexpression experiments by PMID 10656763). This is unique to axolotls and different to mouse and chick. This is indeed the phenotype that we observed ($n = 7/16$ limbs, **Fig. 4b**). We now mention in the text:
 - *"The polydactylous phenotype seen after mosaic Hand2 misexpression is similar to that seen after viral misexpression of Shh in axolotl limbs (Roy 2000)."*

7. Figure 6d-e shows that brief activation of *Shh* can prime anterior cells to express *Hand2*, so one would expect that cells that have received *Shh* once would be able to readily re-activate *Hand2/Shh*. However, Figure 1 shows that the *Shh* lineage does not contribute to the new regenerative *Shh* cells, which contradicts the proposed priming data and suggests that there may be no so-called "memory". *Shh* treatment can also prime anterior cells, but, surprisingly, *Shh* progeny do not necessarily activate *Shh*. Based on their model, this finding is a bit contradictory. Do the authors have an explanation for this in terms of their model?

- Thank you for this discussion of the model. We have demonstrated that *Hand2* defines a posterior memory domain larger than the endogenous *Shh* lineage, and that this memory domain is competent to newly express *Shh* during regeneration (*Shh* is not expressed at steady state, while *Hand2* is expressed continuously). When we removed *Shh* lineage cells surgically, *Hand2* memory cells that remained migrated into the location where *Shh* can be expressed. There, they express *Shh*, substituting the function of the original *Shh* lineage cells. Thus, our manuscript demonstrates a memory effect operating through *Hand2*. As the entire *Hand2* domain is competent to express *Shh*, the data in **Fig 1** (that *Shh* lineage contributes little to regenerative *Shh* cells) are not contradictory to the model.

- Importantly, our conclusion is not that original *Shh* cells do not contribute to the regenerative *Shh* centre, but rather that cells from outside of the lineage (i.e. *Hand2* cells) can.

8. Fig 6d-e shows that there is a priming induced by Shh activation in anterior tissues. Based on this, one would expect that repeated amputations would result in axolotls with limb growth defects as they would activate Hand2/Shh more broadly. However, the axolotl is known to regrow its limb perfectly even after repeat amputations, although one study suggests that they may even lose their regenerative capacity (PMID: 29302364). So how do the authors resolve this with their proposed model?

- Thank you for this discussion. It is clear that, during normal regeneration, the *Shh* signalling domain defines a *Hand2* domain size that corresponds to flawless regeneration. The mechanisms enabling this precise control are an important topic of future study but could be achieved, for example, if a set percentage of *Hand2* memory cells initiates *Shh* expression. Our experiments merely point out that expanding the *Shh* signalling domain via SAG treatment leads to a larger *Hand2* domain, unbalanced regeneration and heritable limb patterning phenotypes. Indeed, we found that re-amputating SAG-treated animals and allowing them to undergo a second round of regeneration resulted in an abnormal number of digits in 4 out of 6 cases.

9. ZRS is used as a Shh enhancer to label the Shh lineage, which marks distal posterior cells. However, there is another study (PMID: 31538936) that shows an additional proximal anterior Shh expression in developing axolotl limbs during limb bud stages. Some of the images in this manuscript (Extended data 2) also show some fluorescent signal for the proximal anterior region (although the signal in these data could be autofluorescence?). However, their lineage tracing data does not show this anterior proximal signal. Could you clarify this point and check if you see the proximal anterior signal by other means such as HCR? Because if they are seeing Shh in the proximal anterior region, they may need to rethink their interpretations because these populations would be primed for Shh? This situation is confusing within this manuscript and inconsistent with another published paper.

- Based on several years of HCR-based characterisations in the lab, we have not observed this anterior *Shh* domain either during development or regeneration (see e.g. **panel a**, below). We have also discussed this issue with colleagues, who similarly did not observe this anterior domain using HCR. We hypothesize that the anterior signal observed in the other study represents probe trapping in whole mount samples.
- As the reviewer mentions, ‘anterior’-looking signal in our study is autofluorescence (visible also in non-stained imaging channels). We have added yellow asterisks throughout the figures to indicate autofluorescence.

10. The broader implications for the tissue engineering component are very vague and a bit stretched. The mechanism described here reads very specific to the axolotl, even for limbs, as the axolotl has its unique anterior-posterior mesenchymal *fgf8-shh* loop. Is there a concrete example and experimental evidence that the authors can provide for their proposed tissue engineering idea? Softening or modifying such broad implications would be a better presentation of the implications.

- We have softened the discussion by removing the following sentence:
 - “An exciting prospect would be to programme patient cells with synthetic cellular memories so that, in response to diverse injury or disease stimuli, these cells express pre-determined signalling outputs in spatial domains conducive to repair or regeneration.”

11. Fig 2 legend indicates “posterior memory cells”. It is not clear how this figure has data related to so-called “memory”. Deleting memory and just mentioning “posterior cells increase Hand2 expression...” represents the data accurately.

- Thank you – we have re-written this figure title as suggested.

12. The authors used *Alx4* to label anterior cells based on an earlier study. The same study and other work in the field found *Gli3R* as another anterior marker (at least at limb bud stages), which is also more relevant to the *Shh* pathway. Is there a reason why the authors did not follow up on *Gli3R*? Do they see its expression as anterior specific in the axolotl and still expressed at post-embryonic stages as *Hand2*? If the transplanted anterior cells contain *Gli3R*, one would expect them to turn on *Hand2* and *Shh* during the regeneration assay, as shown in Fig. 5. This would also support the idea that posterior to anterior transplants induce patterning defects and extra limbs because the cells would have receptors to easily receive *Shh*.

- We chose not to investigate *Gli3R* because its anterior localization is post-translationally regulated and therefore, simply driving reporter gene expression from the *Gli3* locus would not result in anterior expression. The *Gli3* gene itself is not differentially transcribed between anterior and posterior cells at steady state, which precludes its use as a transcriptional reporter for anterior cells. We attempted antibody staining against *Gli3R* but without success. *Alx4* was better matched to our needs for a live anterior cell label.

13. Do the authors have an explanation for why *Hand2* crispants regenerate with fewer digits? If

these crispants have mosaic Hand2 expression or clones with higher Hand2 expression, these Hand2+ cells would be selected, eventually leading to more digit formation during regeneration.

- According to ICE Synthego genotyping of original and regenerated limbs, *Hand2* mutant cells are not outcompeted during regeneration. In regenerating limbs (which are larger than original limbs during development), it might be easier for *Hand2* mutant cells to disrupt the threshold level of *Shh* activation required for successful patterning.

Likewise, if there were a memory or Shh-based priming, one would expect to see more digits after amputations because there would be more Shh? I am not sure that the current model explains these data.

- This would be the expectation if every primed cell expressed *Shh* after amputation. However, our proposed model only relates to the competence to express *Shh*, not actual expression. This is evident from the fact that many posterior cells have the competence to express *Shh* during regeneration, but only a minority of posterior cells actually does (**Extended Data Fig. 3**). It is likely that at least one other factor is required to trigger *Shh* expression. The exact combination of factors remains undefined, but includes e.g. the dorsal-ventral ectoderm, *Tbx2* (PMID: 17300775) and both positive and negative transcription factor inputs (PMID: 28205287).

14. Ext data fig 5 shows that hand2 is expressed in dermal and interstitial fibroblasts, but the images in the main figures look like hand2 is expressed in perichondral fibroblasts, and joints. Could the authors provide additional confocal sections to map where Hand2 is expressed precisely?

- Yes, we have now analysed *Hand2* reporter expression in more depth using confocal microscopy and found expression in all of the connective tissue cell populations mentioned by the reviewer.
 - Please see point 15, below, for detailed characterisation.

15. It is not shown at the cellular level where *Alx4* is expressed. This is important information because the grafting experiments between the AP axis in fig5 assume that *Alx4* and *Hand2* are similar cell types with different positions, but no data on this is shown.

- Yes, we have now analysed *Alx4* reporter expression using confocal microscopy and compared its expression domain with that of *Hand2:EGFP*.
 - We sectioned limbs from large (15 cm) *Alx4:mCherry_Hand2:EGFP* double reporter animals (**panels a-b**) to be certain that we were profiling mature expression and not developmental expression. We profiled longitudinal sections of the entire limb (**panels c-d**).
- In addition, we performed detailed characterisations of connective tissue expression in cross-sections of the lower arm (the area from which grafted cells were isolated). We confirm that *Alx4:mCherry* and *Hand2:EGFP* label similar cell populations in different positions (**panel e**):
 - Peri-skeletal connective tissue, muscle connective tissue, interstitial connective tissue and dermal connective tissue cells.

- We have added all of these data to **Extended Data Figs. 8a-f**, with the following text:
 - “We characterised *Alx4:mCherry* and *Hand2:EGFP* reporter expression throughout the steady state limb using confocal microscopy, finding predominant expression domains across loose connective tissue, joints, skeletal and peri-skeletal elements (**Extended Data Fig. 8c-e**). Importantly, *Alx4:mCherry* and *Hand2:EGFP* labelled a similar range of loose connective tissue cells in the lower arm (dermal, interstitial, muscle, peri-skeletal, **Extended Data Fig. 8f**), allowing us to use these reporters to sort equivalent anterior or posterior cells for transplantation.”

16. The results suggest that hand2 cells are responsible for the effect of the posterior skin graft on the anterior regions, possibly due to their priming for shh expression. Do the authors still see the ectopic patterning and shh expression after amputations when they sort and inject hand2-cells (from the hand2+ posterior areas) (perhaps using prrx1: mcherry background can help with this experiment)? If they don't, then it would further solidify that the positional information refers to this Hand2 gene expression, but not to any other gene expression associated with posterior cells.

- Thank you for this suggestion. We attempted to locate these cells and found that only $3.95 \pm 1.3\%$ of loose *Prrx1*>mCherry+ cells in the posterior limb are *Hand2*:EGFP-negative (**panel a**, arrowheads. $n = 8$ zeugopod cross sections; cells were counted to the level of the ulna, excluding peri-skeletal and skeletal cells).
- This makes it impractical to sort sufficient cells for a transplantation experiment.

17. Hand2 was reported to be posterior specific by the same group, Nacu et al. (ext data fig9. Fig 2 tries to provide the same information as novel. Please provide accurate information on novelty.

- We have added the following clarification:
 - “Expression of *Hand2* in the posterior blastema has been described previously (Nacu 2016).”

18. In Nacu et al - the authors showed that cyclopamine treatment blocks regeneration when given on day 6. Here they use a different inhibitor for the same purpose (which is also suggested to block regeneration in another study PMID: 35433673), but they see some form of growth/regeneration happening in Fig. 6c. Could the authors comment on this discrepancy?

- Nacu et al. blocked regeneration by continuously applying cyclopamine from day 6 through to day 30 post-amputation. In our experiments, we applied BMS inhibitor only once on day 6 post-amputation. We deliberately delayed drug administration until after blastema cell recruitment and initial outgrowth, to test if we can block blastema cell identity from switching. Completely blocking regeneration would prevent us from testing this hypothesis (no outgrowth = no cells to assay).

19. It would be great to indicate autofluorescence. ext Fig. 3 f, Shh could be perceived as it is expressed in the middle bones. A similar situation can be seen in other figures.

- Yes, we have better highlighted autofluorescence throughout the figures with yellow asterisks.

20. Skin grafting with AP juxtaposition can induce additional limbs (ALM model). Meanwhile, in this study, anterior or posterior cell "injection" does not. The authors' figures show that the ALM model requires additional wounding and innervation. It might be useful to point out this difference in the text, as it might be perceived as somewhat contradictory to their earlier study.

Thank you for this suggestion. In contrast to the ALM, we do not deviate a nerve to the site of cell injection. We inject cells and amputate after a recovery period. Thus, we do not expect an additional limb field to form in the cell injection assay.

21. The experiment in Figure 5g is labeled "regenerated limb," but the limb looks abnormal and is noted as ectopically patterned in the text. It might be more accurate to mention that this does not fully suggest a regeneration scenario in the figure.

- Yes, the Figure legend 5g clarifies this:
 - *"These limbs regenerated with aberrant pattern."*

Referee #3 (Remarks to the Author):

The manuscript by Otsuki et al reports on the molecular nature of cellular memory in axolotl limbs and limb blastemas during regeneration. The authors develop an extensive set of new cell tracing and functional genetic models in the axolotl and combine this with drug treatments and tissue transplantation/amputation to demonstrate that persistent Hand2 expression is responsible for posterior identity during limb regeneration. They also demonstrate that the signalling environment can switch anterior to posterior cell identity but not vice-versa. Furthermore, they uncover a positive feedback loop between Shh and Hand2 required for regeneration and able to reset anterior cell identity to a posterior identity/memory. This work is a major breakthrough in understanding positional memory and its role in regeneration, with implications for tissue engineering. The manuscript is rather mature for publication and I only have minor comments/suggestions:

-In Figure 2e, Hand2 reporting in the mature limb appears evident only in skeletal elements, however in the sections shown in the extended data Figure 5a, the reporter appears exclusively in connective tissue. The authors may consider explaining this apparent discrepancy

- Yes, we have now analysed in more depth the *Hand2* reporter expression domain and added these data to to **Extended Data Figs. 8a-f**.
 - We sectioned limbs from large (15 cm) *Alx4:mCherry_Hand2:EGFP* double reporter animals (**panels a-b**) to be certain that we were profiling mature expression and not developmental expression. We profiled longitudinal sections of the entire limb (**panels c-d**).
- We also performed detailed characterisations of connective tissue expression in cross-sections of the lower arm (the area from which grafted cells were isolated). We confirm that *Alx4:mCherry* and *Hand2:EGFP* label similar cell populations in different positions (**panel e**):
 - Peri-skeletal connective tissue, muscle connective tissue, interstitial connective tissue and dermal connective tissue cells.

- We have added all of these data to **Extended Data Fig. 8**, with the following text:
 - “We characterised *Alx4:mCherry* and *Hand2:EGFP* reporter expression throughout the steady state limb using confocal microscopy, finding predominant expression domains across loose connective tissue, joints, skeletal and peri-skeletal elements (**Extended Data Fig. 8c-e**). Importantly, *Alx4:mCherry* and *Hand2:EGFP* labelled a similar range of loose connective tissue cells in the lower arm (dermal, interstitial, muscle, peri-skeletal, **Extended Data Fig. 8f**), allowing us to use these reporters to sort equivalent anterior or posterior cells for transplantation.”

- Do the authors have a notion of the proportion of Hand2-deleted cells required to produce the diversity of phenotypes shown in Figure 3 and extended data Figure 6? Given that in some of these mosaic animals Shh is not detected, does this suggest non-autonomous cell effects of Hand2 deletion?

- Thank you for this interesting question. We did attempt to correlate limb phenotype with indel frequency (determined using ICE Synthego genotyping). However, we found this analysis to be unsatisfactory without spatial information for each animal (e.g. *Hand2* deletion in an anterior cell would not be equivalent to *Hand2* mutation in a posterior cell). Directly comparing the positions and frequencies of mutant cells with final limb phenotype is an interesting (but challenging) question for future study.
- Without knowing where the mutant cells are located, it is difficult to comment on autonomous vs non-autonomous effects on *Shh* expression.

- The authors may think on revising this statement: "Taken together, these results show that Hand2 is necessary for posterior identity and suggest that cells that fail to acquire a posterior identity do not default to an anterior identity".

Another view is that cells do not fail to acquire posterior identity (which could only be determined by a wider expression analysis) but rather that they lost their ability to support regeneration through losing their ability to activate Shh.

- Thank you for this suggestion. We have revised this statement to reflect this:
 - *"Taken together, these results show that Hand2 is necessary for posterior identity, at least insofar as inducing Shh expression, and suggest that cells that fail to acquire a posterior identity do not default to an anterior identity."*

Referee #4 (Remarks to the Author):

The study by Otsuki et al. identifies the molecular mechanism that provides posterior cells in adult axolotl limbs with a memory of their posterior origin by their low level expression of the Hand2 transcription factor. This primes for regeneration after limb amputation as the observed Hand2 up-regulation reinitiates Shh expression in the blastema and establishes a Hand2-Shh positive feedback loop. After limb regeneration is complete, this posterior memory is re-established by sustained low level Hand2 expression in posterior cells. The authors show that this memory is stable and that Hand2-expressing cells cannot be "anteriorized". Instead they keep their ability to trigger the Hand2-Shh feedback loop repeatedly following re-amputation. Furthermore, anterior cells can be reprogrammed as Hand2 expressing memory cells by exposure to Shh.

This study provides fundamental novel insights into how a long-term molecular memory of posterior identity is maintained and able to trigger complete regeneration by initiating a positive feedback loop between Hand2 and Shh. These results are of very high general importance as they provide insights into processes that to date have remained elusive. In addition, the approaches are state-of-the-art and showcase novel genetic and molecular tools that make axolotl the prime model to study the molecular basis of cellular memory for limb regeneration.

Comments

1. Hand2: based on their transcriptome analysis (Fig. 2a-c), the authors select the transcription

factor Hand2 as the prime transcription factor candidate for molecular memory. This is valid but one could have equally well chosen Hoxd13/a13 as these 5' Hox transcription factors are also required to activate Shh and function as part of a positive feedback with Shh during limb development in vertebrates. However, this reviewer agrees that Hand2 is a good choice and is not suggesting analysis of 5' Hox genes, but would like the authors to discuss the potential involvement/requirement of 5' Hox and other posterior transcription factors. It is almost a given that for long-term and robust molecular memory, more than a two-factor feedback system is needed.

- In our new transcriptional profiling of *Hand2*-misexpressing cells, we found that *Hoxd13* expression is strongly induced by *Hand2* (**Extended Data Table 3**). This supports the reviewer's view that 5' Hox is involved in the molecular memory mechanism and demonstrates that *Hoxd13* is at least downstream of *Hand2*. It may additionally act upstream of *Hand2*. We now note this in the text.
- We have now clarified in the discussion that additional molecular factors such as 5' Hox genes are likely to play roles in the robust memory mechanism:
 - *"It is possible that 5' Hox transcription factors such as Hoxa13 and Hoxd13 interact with Hand2 to maintain robust posterior identity and regulate Shh expression, similar to their roles during limb bud development: we indeed observe that these genes are posteriorly enriched in our RNA-seq data and that Hand2 induces Hoxd13 expression (Fig. 2c, Extended Data Fig. 4b, Supplementary Data Table 3)."*

2. The authors should be cautious about quantitative statements when using EGFP reporter construct as this protein is very stable. For quantitative analysis one would have to use one of the available labile GFP versions.

Can add this to the text.

- Thank you for pointing this out. We now mention this in the manuscript:
 - *"Note that these fluorescence comparisons should be taken as semi-quantitative estimates, as we did not make use of a labile EGFP in these assays."*

3. Page 7: the authors state that in other vertebrates it is the distal epidermis-posterior mesenchyme that controls limb bud outgrowth, which is too simplistic. Many studies have shown that for properly controlled limb bud development, interactions between anterior and posterior transcription factors and signaling pathways are crucial.

- Thank you. We have amended this sentence to reflect this:
 - *"In the context of limb outgrowth, a major source of positional discontinuity in axolotls for both development and regeneration is the anterior mesenchyme-posterior mesenchyme. In other vertebrates, one of the most intensively studied cellular interactions is between the distal epidermis and posterior mesenchyme, although anterior-posterior interactions are similarly important¹⁷."*

4. In this reviewer's view the authors should discuss the striking parallels that emerge thanks to their study between normal limb development (chicken/mouse) and axolotl limb regeneration. This is really stunning – just one example: the posterior dominance during regeneration seems equivalent to posterior prevalence during normal limb development.

- Thank you for the enthusiasm. We have added a sentence highlighting this intriguing parallel.

- *“This posterior dominance bears striking parallels with posterior prevalence of Hox patterning genes during limb development, in which posterior Hox proteins exert a more dominant effect than anterior Hox genes in areas of co-expression. It is possible that 5’ Hox transcription factors such as Hoxa13 and Hoxd13 interact with Hand2 to maintain robust posterior identity and regulate Shh expression, similar to their roles during limb bud development: we indeed observe that these genes are posteriorly enriched in our RNA-seq data and that Hand2 induces Hoxd13 expression (Fig. 2c, Extended Data Fig. 4b, Extended Data Table 3).”*

5. As this manuscript is submitted to a very general journal: can the authors briefly discuss/speculate if such memory cells exist or not in higher vertebrates like mammals and why this trait was lost during evolution of higher vertebrates.

- We have added a sentence to the discussion:
 - *“An important question now is if Hand2 memory cells exist in adult humans and if they could be triggered to induce Shh expression for tissue formation. A memory code based on spatially appropriate expression of Hox genes has been described in adult mammals in the context of the mouse limb (Rux et al. 2016, 2017; Pineault et al. 2019) as well as more generally throughout adult human body tissues (Chang et al. 2002, Rinn et al. 2006).”*

Technical issues

This is a complex and data-rich study of very high quality and excellent design, which fully supports the conclusions by the authors. There is few, but important additional comments:

6. Shh HCRs: there is unfortunately significant background signal -probably from the vascular system. This renders assessment of data difficult.

Fig. 2k: in addition to the overlap, please include the individual panels for the Hand2 lineage and Shh HCR using a greyscale.

- Yes, we have added the following individual panels as greyscales to **Extended Data Fig. 5i**.

Fig. 5e/h: Shh signal and background are difficult to discern.

- We have added a zoomed-out image of 5h in **Extended Data Fig. 9c**, as similarly requested by Reviewer 1, which better demonstrates the specificity of the Shh probe.

Fig. S9g: due to the background the normal and ectopic Shh signaling center are hardly discernible and the scheme is not really helpful either. Can the authors please consider to include high mags and or conventional BMPurple whole mount *in situ* to detect Shh.

- We repeated these experiments but experienced unavoidable background when detecting *Shh* by *in situ* in these treatments and in these sizes of animals.
- However, we now confirm that posteriorized anterior cells can express *Shh* using a different assay, RNA-sequencing.
 - We FACS purified anterior cells transplanted anteriorly (A->A) or posteriorly (A->P) and compared gene expression. In parallel, we also sequenced *bona fide* untransplanted anterior cells (A) and posterior (P) cells. We found that only posterior cells and cells transplanted posteriorly expressed *Shh* among these conditions (**panels a-b**).

- We have added these data to **Extended Data Figs. 9d,f**.

7. Fig. 4l: The “ZRS>TFP” lineage is indicated but not visible. Including a high magnification of the relevant region of the ectopic limb in the left panel seems necessary.

- Thank you for pointing this out. In fact these panels have no TFP signal (they have exceeded the expression window of *Shh*), and neither is this information relevant to the interpretation of the experiment. Therefore, we have removed this image channel and caption from the Figure. The earlier expression of TFP in younger ALMs can be seen in **Extended Data Fig. 7k**.

Minor but still important issues

8. Line 232: “...Hand2 is necessary to activate Shh expression...” (rather than "to express")

- We have edited this sentence as suggested.

9. Over all the readability of the manuscript could be improved somewhat by removing some non-important details and redundant text (not easy to pinpoint what to cut exactly).

- We have taken care in our revisions to provide clear statements.

10. Literature reference 27 (Charite et al. 2000) is problematic as citation for proof that Hand2 is required to activate Shh. The main reason is that Hand2-deficient embryos start to die around the stage (due to the severe heart defects) when Shh is normally activated in early limb buds, which poses a cause and consequence problem. This was circumvented by limb bud specific conditional inactivation of Hand2, which indeed disrupts Shh activation but not early limb bud development (Galli et al. 2010, reference 32).

- Thank you – we agree and have corrected our referencing.

END

Response to reviewer comments, round 2

(comments received on 2025-02-25)

We thank the reviewers for their help in improving our manuscript.

Referee #1 (Remarks to the Author):

General:

The authors have done a very thorough job addressing each of the concerns we raised, many of which were vital for ensuring arguments were well supported, the audience was informed of the considerations and nuances influencing experimental decisions and conclusions, and the work was adequately contextualized with existing literature. Certainly, we can all agree that interesting questions remain for future work that will be inspired by this study, including some questions I could now ask that could be riffs off valid questions the other reviewers raised. However, assuming the other reviewers are satisfied with the authors' efforts and responses to their personal critiques, I do not favor more revision. A second revision would perpetuate the practice of journals asking for even more experimentation and analysis beyond what was reasonably and implicitly agreed upon by all the parties at this table. Scientific progress using the axolotl model has been thwarted by these kinds of practices, resulting in major papers being held up for years.

Thank you.

Specific (minor, but would be an easy addition that I do not need to re-review following inclusion in text):

Their reasoning about the genetics makes sense, and it is important to include these details in the manuscript, which the authors have now done. One final point that I think should be noted in the revised text before it is published is that around lines 246-250, after rationalizing why use the crispants in the cross due to probable homozygous loss-of-function lethality at the Hand2 locus: please add one sentence mentioning that conditional alleles of endogenous genes (for example, floxed alleles of Hand2) is a method not yet available in axolotl. Without mentioning this, I feel readers will next wonder why not knock Hand2 out specifically in limb tissues, in animals that had already fully developed. That strategy is commonplace in mouse, for example, and I think it would benefit both the paper in hand and our field for this information to be made overt. Beyond it not being an established technology in axolotl (likely due to low efficiency of targeted knock-ins in general?), such an approach would also require something like three years to execute rigorously.

We agree and have added the following sentence to the manuscript:

“Conditional deletion of endogenous genes is not readily feasible in axolotls.” (Line 176).

Referee #2 (Remarks to the Author):

The authors have adequately addressed most of my concerns. I remain skeptical about their claims that the observed phenotype represents cellular reprogramming. In the current literature, such claims are demonstrated and well-supported by single-cell sequencing methods or pure isolated cell grafting and tracing experiments. Bulk RNA-seq and heterogeneous population grafts, as performed here, are less rigorous (their methods and conclusions would have been acceptable a decade ago, but this concept and the methods

used to investigate it have evolved). While the technical limitations they discuss are somewhat valid (although more sensitive single-cell methods, like SMART-seq variations, exist, albeit impractical in this context), I still recommend they reconsider their use of the term "reprogramming" to align with current standards and to avoid unnecessary critique of an otherwise high-quality and interesting study.

I have one analysis suggestion to partially address my concern. The authors indicated that they identified 146 genes linked to anterior tissue identity and 160 genes to posterior tissue identity in their bulk RNA-seq data. In their new bulk RNA-seq analysis following grafting between anterior and posterior tissues (Extended Data Fig. 9), they should report how many of the anterior and posterior genes from the 146 and 160 suggested, beyond the six selected (Extended Data Fig. 9f), show statistically significant changes following grafting. While they highlight statistically significant genes in Supplementary Tables 8-11, these documents do not provide this specific information. Reporting in-text how many of these anterior and posterior identity genes (e.g., X/146) show significant shifts following grafting would strengthen their claims, even if it does not fully meet the standards of current literature (and they would still not account for other potential explanations such as tissue heterogeneities influencing results..).

This is an excellently implemented and interesting study. I still find the manuscript verbose, and their responses to questions about their model are somewhat incomplete. However, these issues should not prevent the manuscript's publication. I am supportive of its acceptance.

Thank you.

To address the reviewer's concern, we have de-emphasised reprogramming in our revised manuscript. We replaced almost all instances with "posteriorized". However, we stand by our original conclusion that we observe a significant identity change consistent with reprogramming (as supported by transcriptional profiling, imaging and *in situ* assays).

We agree with the requested analysis and have performed this. Instead of comparing to the 146/160 genes suggested by the reviewer (which were identified in steady state cells), we identified anterior- and posterior-specific genes from our blastema RNA-seq data. We overlapped these genes with the genes up- and down-regulated in transplanted cells. We added the following result to the manuscript:

"Overall, A->P transplants downregulated 60.1% (578/961) of anterior blastema-specific genes and upregulated 22.5% (78/346) of posterior blastema-specific genes (Supplementary Table 13)." (Lines 315-316).

We added a new table listing these genes (Supplementary Table 13).

Referee #3 (Remarks to the Author):

The authors have answered all my concerns. The manuscript reveals a fundamental aspect of regenerative biology through extensive and rigorous experiments. It also opens several future questions related to the topic of positional memory: how is hand2 expression maintained? is it functional in mature cells or just sitting there as positional memory? is hand2 alone or just a part of a TF network with various alternative entry points?

Thank you – yes, we agree with these fascinating future questions.

Referee #4 (Remarks to the Author):

Otsuki and colleagues have conducted a large number of additional analyses and revised the text extensively in response to the comments of all four reviewers. This has significantly improved the study and added important support to the data and conclusions.

There is one issue that this reviewer already pointed out in the initial review, namely that a 2-component feedback system involving HAND2 and SHH as proposed by the authors is unlikely sufficient for establishing and maintaining a robust but yet reprogrammable memory mechanism. Rather, this study appears to identify two of the (essential) components of a likely larger and more complex molecular mechanism that underlies positional memory and its re-programmability, which as such is a very important discovery. The comment is not focusing on a detail, as data-based simulations of feedback systems have shown that systems consisting of several interacting feedback loops generate both robustness and adaptability - as seems the case for positional memory. Furthermore, HAND2 regulates the spatio-temporal kinetics of Shh expression in concert with several other transcriptional regulators. Genetic analysis in the mouse shows that the essential transcription factors include Hand2, 5'Hox TFs, Pbx, Gli3 and others that function in Shh activation, up-regulation and restriction within the posterior limb mesenchyme. The authors should nuance their discussion and put their findings better into this broader context in light of the general and fundamental importance of studies published by Nature.

Thank you for this important point.

We have extended the discussion to reflect these comments:

“Shh activation during limb development involves multiple transcriptional inputs. Genetic analyses in mouse and chick uncovered numerous transcription factors (including Pbx1/2, Gli3, Tbx3) acting with Hand2 and 5' Hox required for posterior Shh expression^{42,44-46}, although precisely how such factors place the Shh domain remains elusive⁴⁷. Moreover, the BMP inhibitor Grem1 acts as a relay to balance posterior Shh and BMP activity with Fgf expression, outgrowth and termination of limb development⁴⁸⁻⁵¹. Understanding how robust positional memory is achieved will require analysis of anterior-posterior chromatin accessibility, transcription factor binding, and modelling of multiple feedback loops between signalling pathways beyond Hand2 and Shh.” (Lines 374-382).

END